# Climate indices in historical climate reconstructions: A global state-of-the-art

David J. Nash[a,b], George C.D. Adamson[c], Linden Ashcroft[d,e], Martin Bauch[f], Chantal Camenisch[g,h], Dagomar Degroot[i], Joelle Gergis[j,k], Adrian Jusopović[l], Thomas Labbé[f,m], Kuan-Hui Elaine Lin[n,o], Sharon D. Nicholson[p], Qing Pei[q], María del Rosario Prieto[r†], Ursula Rack[s], Facundo Rojas[r] and Sam White[t]

[a]  School of Environment and Technology, University of Brighton, Brighton, United Kingdom

[b]  School of Geography, Archaeology and Environmental Studies, University of the Witwatersrand, Johannesburg, South Africa

[c]  Department of Geography, King's College London, London, United Kingdom

[d]  School of Earth Sciences, University of Melbourne, Melbourne, Australia

[e]  ARC Centre of Excellence for Climate Extremes, University of Melbourne, Melbourne, Australia

[f]  Leibniz Institute for the History and Culture of Eastern Europe, University of Leipzig, Leipzig, Germany

[g]  Oeschger Centre for Climate Change Research, University of Bern, Bern, Switzerland

[h]  Institute of History, University of Bern, Bern, Switzerland

[i]  Department of History, Georgetown University, Washington DC, USA

[j]  Fenner School of Environment & Society, Australian National University, Canberra, Australia

[k]  ARC Centre of Excellence for Climate Extremes, Australian National University, Canberra, Australia

[l]  Institute of History, Polish Academy of Sciences, Warsaw, Poland

[m]  Maison des Sciences de l'Homme de Dijon, University of Burgundy, Dijon, France

[n]  Research Center for Environmental Changes, Academia Sinica, Taipei, Taiwan

[o]  Graduate Institute of Environmental Education, National Taiwan Normal University, Taipei, Taiwan

[p]  Department of Earth, Ocean, and Atmospheric Science, Florida State University, Tallahassee, Florida, USA

[q]  Department of Social Sciences, Education University of Hong Kong, Hong Kong, Peoples Republic of China

[r]  Argentine Institute of Nivology, Glaciology and Environmental Sciences (IANIGLA-CONICET), Mendoza, Argentina

[s]  Gateway Antarctica, University of Canterbury, Christchurch, New Zealand

[t]  Department of History, Ohio State University, Columbus, Ohio, USA

[†]  Deceased

*Correspondence to*: David J. Nash (d.j.nash@brighton.ac.uk). ORCID: 0000-0002-7641-5857

**Abstract**. Narrative evidence contained within historical documents and inscriptions provides an important record of climate variability for periods prior to the onset of systematic meteorological data collection. A common approach used by historical climatologists to convert such qualitative information into continuous quantitative proxy data is through the generation of ordinal-scale climate indices. There is, however, considerable variability in the types of phenomena reconstructed using an index approach and the practice of index development in different parts of the world. This review, written by members of the PAGES CRIAS Working Group – a collective of climate historians and historical climatologists researching Climate Reconstructions and Impacts from the Archives of Societies – provides the first global synthesis of the use of the index approach in climate reconstruction. We begin by summarising the range of studies that have used indices for climate reconstruction across six continents (Europe, Asia, Africa, the Americas, Australia) plus the world's oceans. We then outline the different methods by which indices are developed in each of these regions, including a discussion of the processes adopted to verify and calibrate index series, and the measures used to express confidence and uncertainty. We conclude with a series of recommendations to guide the development of future index-based climate reconstructions to maximise their effectiveness for use by climate modellers and in multiproxy climate reconstructions.

**Keywords**. Climate reconstruction; temperature reconstruction; precipitation reconstruction; historical climatology; climate history; documentary evidence

## 1. Introduction

Much of the effort of the palaeoclimatological community in recent decades has focussed on understanding long-term changes in climate, typically at millennial, centennial, or at best (in the case of dendroclimatology and palaeolimnology) sub-decadal to annual resolution. The results of this research have revolutionised our knowledge both of how climates have varied in the past and the potential drivers of such variability. However, as Pfister et al. (2018) identify, the results of palaeoclimate research are often at a temporal and spatial scale that is not suitable for understanding the short-term and local impacts of climate variability upon economies and societies. To this end, historical climatologists work to reconstruct high-resolution – annual, seasonal, monthly and in some cases daily – series of past temperature and precipitation variability from the archives of societies, as these are the scales at which weather impacts upon individuals and communities (e.g. Allan et al., 2016; Brönnimann et al., 2019).

The archives of societies, used here in a broad sense to refer to both written records and evidence preserved in the built environment (e.g. historic flood markers, inscriptions), contain extensive information about past local weather and its repercussions for the natural environment and on daily lives. Information sources include, but are not limited to, annals, chronicles, inscriptions, letters, diaries/journals (including weather diaries), newspapers, financial, legal and administrative documents, ships' logbooks, literature, poems, songs, paintings and pictographic and epigraphic records (Brázdil et al., 2005; Brázdil et al., 2010; Brázdil et al., 2018; Pfister, 2018; Rohr et al., 2018). Three main categories of information appear in these sources that can be used independently or in combination for climate reconstruction: (i) early instrumental meteorological data; (ii) records of recurring physical and biological processes (e.g. dates of plant flowering, grape ripening, the freezing of lakes and rivers); and (iii) narrative descriptions of short-term atmospheric processes and their impacts on environments and societies (Brönnimann et al., 2018).

The heterogeneity of the archives of societies – in time, space and in the types of information included in individual sources – raises conceptual and methodological challenges for climate reconstruction. Historical meteorological data can be quality-checked and analysed using standard climatological methods, while records of recurrent physical and biological phenomena provide proxy information that may be assessed using a variety of palaeoclimatological approaches (cf. Brönnimann et al., 2018). Narrative descriptions, however, require different treatment to make local observations of weather and its impacts compatible with the statistical requirements of climatological research.

A common approach used in historical climatology for the analysis of descriptive (or narrative) evidence is the generation of ordinal-scale indices as a bridge between raw weather descriptions and climate reconstructions. A simple index might, for example, employ a three-point classification, with months classed as −1 (cold or dry), 0 (normal) and 1 (warm or wet) depending upon the prevailing conditions described within historical sources. As Pfister et al. (2018) note, this "index" approach provides a means of converting "disparate documentary evidence into continuous quantitative proxy data… but without losing the ability to get back to the short-term local information for critical inspection and analysis" (p.116). Brázdil et al. (2010) provide a detailed account of the issues associated with the generation of indices.

The index approach to historical climate reconstruction over much of the world – an exception being China – has its roots in European scholarship. There is, however, considerable variability in the types of phenomena reconstructed using an index approach in different areas. There is also variability in practice, both in the way that historical evidence is treated to generate indices and in the number of ordinal categories in individual index series. Variability in the treatment of evidence arises, in part, from the extent to which analytical approaches have developed independently. In terms of categorisation, three-, five- and seven-point index series are most widely used but greater granularity (i.e. a greater number of index classes) may be achieved in different regions and for different climate phenomena depending upon the quantity, resolution and/or richness of the original historical evidence.

This study arises from the work of the PAGES (Past Global Changes) CRIAS Working Group, a cooperative of climate historians and historical climatologists researching Climate Reconstructions and Impacts from the Archives of Societies. The first meeting of the Working Group in Bern, Switzerland, in September 2018 identified the need to understand variability and – ideally – harmonise practice in the use of indices to maximise the utility of historical climate reconstructions for climate change investigations. This study, written by regional experts in historical climatology with contributions from other CRIAS members, is intended to address this need.

The main aims of this paper are to: (i) provide a global state-of-the-art review of the development and use of the index approach as applied to descriptive evidence in historical climate reconstruction; and (ii) identify best practice for future investigations. It does so through a continent-by-continent overview of practice, followed by a review of the use of indices in the reconstruction of climate variability over the oceans. Studies from northern polar regions are reviewed within sections 5 (the Americas) and 7 (the Oceans), as appropriate. To the knowledge of the authors, no studies of the climate history of Antarctica use an index approach.

Three caveats are necessary to frame the coverage of the review. First, the nature of documentary sources is well discussed in the climate history literature for most parts of the world. As such, we provide only limited commentary on sources for each continent, except for selected regions. These include China, where only a few overviews of documentary sources have been published (e.g. Wang, 1979; Wang and Zhang, 1988; Zhang and Crowley, 1989; Ge et al., 2018), and Japan and Russia where, to our knowledge, no detailed descriptions are available for Anglophone audiences. Second, there are instances in the literature where quantifiable data in documentary sources (e.g. sea-ice cover, phenological phenomena) and even instrumental meteorological data are converted to indices for climate reconstruction purposes. This occurs mainly in studies where such data are integrated with narrative evidence to generate longer, more continuous and homogenous series with a consistent (monthly or seasonal) resolution. We do not describe the generation of such index series in detail, but do provide examples in sections 2 to 7, as appropriate. Third, the emphasis of the article is on the documentation of studies that have used an index approach to climate reconstruction, with critical review and comparison where appropriate. The number of instances where comparative analysis is possible is necessarily restricted by the limited number of studies that have undertaken either different approaches to index development for the same location or identical approaches for different regions.

## 2. Climate indices in Europe

### 2.1. Origins of documentary-based indices in Europe

The use of climate indices has a long tradition in Europe, with the earliest studies published during the 1920s CE. As in any area, the start date for meaningful index-based reconstructions is determined by the availability of source material. In Central, Western and Mediterranean Europe, for example, sources containing narrative evidence are sufficiently dense from the 15th century CE onwards to enable seasonal index reconstruction for more than half of all covered years. Exceptionally, indices can be generated from the 12th century onwards, but with greatest confidence from the 14th century when serial sources join the available historiographic information (Wozniak, 2020). The number of index-based climate reconstructions for Europe is large; as such, this section of the review focusses mainly upon studies that include original published series based on primary sources and that reconstruct meteorological entities. This excludes climate modelling and other studies that synthesise or reanalyse previously published historical index series.

Due to the dominance of references to winter conditions in European documentary sources, early investigations centred primarily on winter severity (Pfister et al., 2018). The first use of the index approach was by the Dutch journalist, astronomer and later climatologist Cornelis Easton, who published his oeuvre on historical European winter severity in 1928 (Easton,

1928). In this monograph, Easton presented early instrumental data but also a catalogue of descriptions of winter conditions dating back to the 3rd century BCE derived from narrative evidence. For the period prior to 1205 CE, this catalogue lists only remarkable winter seasons; however, after this date every winter up to 1916 is attributed to a ten-point classification, including a quantifiable coefficient and a descriptive category. Easton's classification appears as an adapted graph in the second edition of Charles E. P. Brooks (1949) book on *Climate Through the Ages* (Pfister et al., 2018).

An isolated attempt to quantify the evaluation of weather diaries (spanning 1182-1780 CE) was proposed by the German meteorologist Fritz Klemm (1970), with a two-point scale for winter and summer temperature (cold/mild and mild/warm respectively) and precipitation (dry/wet). The Dutch meteorologist Folkert IJnsen also developed winter severity indices for the Netherlands (1200-1916 CE) but following a slightly different approach (IJnsen and Schmidt, 1974). However, one of the most important advances came in the late 1970s when British climatologist Hubert Horace Lamb published three-point indices of winter severity and summer wetness for Western Europe (1100-1969 CE) in his seminal book *Climate: Past, Present and Future* (Lamb, 1977). Lamb's methodology was more easily applicable compared to Easton's – a likely reason why successive studies refer to Lamb's method and why, in the aftermath of his publication, the index approach was applied in many different European regions.

In 1984, the Swiss historian Christian Pfister published his first temperature and precipitation indices for Switzerland in the volume *Das Klima der Schweiz von 1525-1860*, expanding his climate indices to cover all months and seasons of the year (Pfister, 1984). Pfister's work adapted Lamb's methods, extending Lamb's three-point scale into monthly seven-point ordinal-scale temperature and precipitation indices (Figure 1). Shortly after Pfister's initial study, Pierre Alexandre (1987) developed a comprehensive overview of the climate of the European Middle Ages (1000-1425 CE), also using indices. Over a decade later, Van Engelen et al. (2001) published a nine-point index-based temperature reconstruction for the Netherlands and Belgium (764-1998 CE). Most research groups investigating European climate history – including those led by Rüdiger Glaser (Freiburg, Germany) and Rudolf Brázdil (Brno, Czech Republic) – now adopt Pfister's approach as the standard method for index development, at least for temperature and precipitation reconstructions. This is described in more detail in section 8 as part of a global overview of approaches to index construction. The opportunity to combine narrative evidence with quantifiable information is one of the great advantages of the index-approach (Pfister et al., 2018). As a result, many index-based series for Europe incorporate some quantitative data. Many series also contain data gaps; the earlier the epoch, the more likely there are to be breaks in series – this is common to almost all index-based series globally.

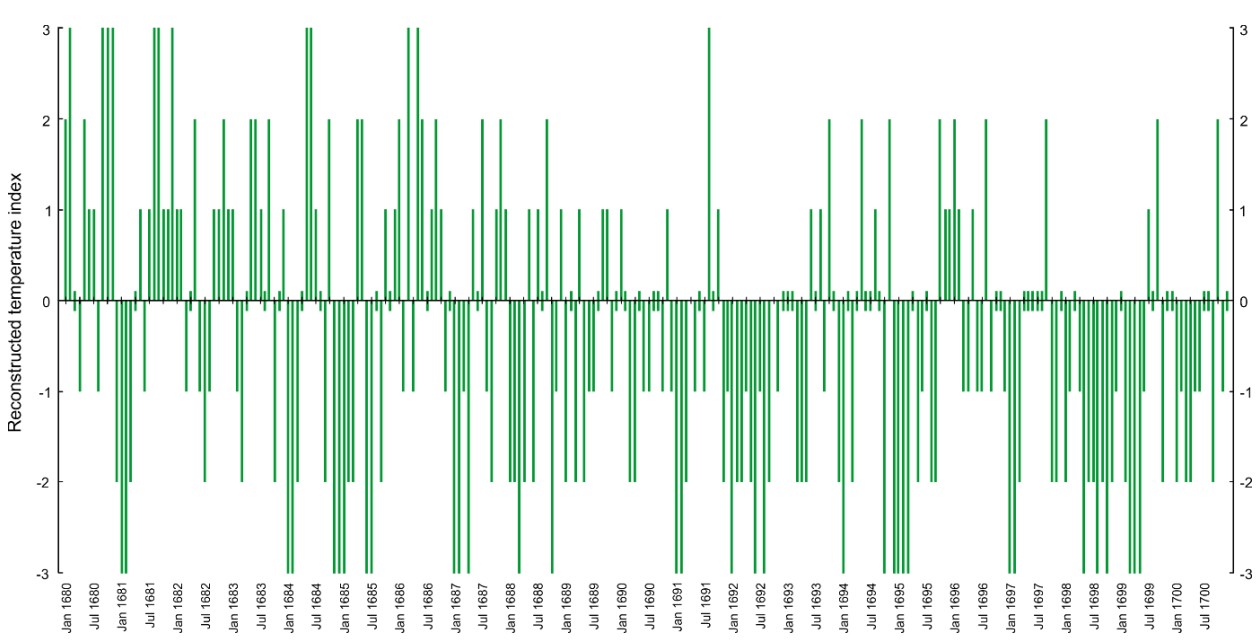

**Figure 1**: Monthly seven-point temperature indices for the Swiss Plateau (1680-1700), reconstructed using the Pfister index approach (data from Pfister, 1998). Zero values for specific months are indicated by a small green bar.

One area of Europe with a different research tradition is Russia (Jusupović and Bauch, 2020). Here, the earliest climate history research was by K.S. Veselovskij (1857), who compared historical information from various source types against early 19th century statistical climate data (for more details of Veselovskiy's work, see Zhogova, 2013). M.A. Bogolepov later analysed climate-related information in published Cyrillic and Latin sources from the 10th century onwards (Bogolepov, 1907, 1908, 1911). Other studies have focused on accounts of anomalous weather in Russian sources (e.g. Borisenkov and Paseckij, 1983, 1988) and on reconstructing historical climate (Burchinskij, 1957; Liakhov, 1984; Borisenkov, 1988; Klimanov et al., 1995; Klimenko et al., 2001; Slepcov and Klimenko, 2005; Klimenko and Solomina, 2010), river flows (Oppokov, 1933) and famine years (Leontovich, 1892; Bozherianov, 1907).

The most important collection of Russian documentary sources is the 43-volume Полное Собрание Русских Летописей ('Complete Collection of Russian Chronicles', abbreviated to ПСРЛ; Borisenkov and Paseckij, 1988). These chronicles document events including infestations of insects, droughts, wet summers, wet autumns, unusual frost events, famine, floods, storms and earthquakes. The records have been used, in conjunction with other European sources, by Borisenkov and Paseckij (1988) to reconstruct a qualitative Russian climate history for the last 1000 years. More recent reconstructions have extended beyond historical sources to include a variety of other climate proxies (e.g. Klimenko and Solomina, 2010). The development of index-based series from narrative evidence has yet to be attempted, although reconstructions of specific meteorological extremes, including wet/dry/warm/cold seasons and floods plus related socio-economic events such as famines, have been published by Shahgedanova (2002) (based on Borisenkov and Paseckij, 1983).

## 2.2. Temperature indices

Temperature is the most common meteorological phenomenon analysed using an index approach over northern and central Europe. Authors who have developed temperature index series include Christian Pfister (1984, 1992, 1999), Pierre Alexandre (1987), Rudolf Brázdil (e.g. Brázdil and Kotyza, 1995, 2000; Brázdil et al., 2013a; spanning periods from 1000-1830 CE), Rüdiger Glaser (e.g. Glaser et al., 1999; Glaser, 2001; Glaser and Riemann, 2009; 1000-2000 CE), Astrid Ogilvie and Graham Farmer (1997; 1200-1439 CE), Gabriela Schwarz-Zanetti (1998; 1000-1524 CE), Lajos Rácz (1999; 16th century onwards), the Dutch working group around Aryan van Engelen (Van Engelen et al., 2001; Shabalova and van Engelen, 2003), Maria-João Alcoforado et al. (2000; 1675-1715 CE), Elena Xoplaki et al. (2001; 1675-1715 and 1780-1830 CE), Anita Bokwa et al. (2001; 16th and 17th centuries), Petr Dobrovolný et al. (2009), Dario Camuffo et al. (2010; 1500-2000 CE), Maria Fernández-Fernández et al. (2014; 2017; 1750-1840 CE), Laurent Litzenburger (2015; 1400-1530 CE) and Chantal Camenisch (2015a; 2015b; 15th century). The basis of these reconstructions is mainly narrative evidence from multiple sources, or in the case of Brázdil and Kotyza (1995, 2000) and Fernández-Fernández et al. (2014), a single narrative source. However, depending on the epoch, evidence may be supplemented by information from early weather diaries, administrative records and legislative sources. The majority of these studies (e.g. Pfister, 1984, 1992; Brázdil and Kotyza, 1995; Glaser et al., 1999; Pfister, 1999; Rácz, 1999; Brázdil and Kotyza, 2000; Glaser, 2001; Van Engelen et al., 2001; Shabalova and van Engelen, 2003; Dobrovolný et al., 2009; Glaser and Riemann, 2009; Camuffo et al., 2010) include an overlap with available instrumental data.

In Europe, different types of index scales have been used. As noted above, Christian Pfister (1984) developed a seven-point scale with a monthly resolution for temperature and precipitation (e.g., for temperature, -3: extremely cold, -2: very cold, -1: cold, 0: normal, 1: warm, 2: very warm, 3: extremely warm). Most historical climatologists follow this approach, though in some cases less granulated versions have had to be applied due to limited source density or quality. For instance, Glaser (2013) followed Pfister's indexing approach but used a three-point scale for the period 1000-1500 as information on weather

appear only occasionally in documentary sources from this time. Schwarz-Zanetti (1998), Litzenburger (2015) and

Camenisch (2015a) have also applied seven-point indices for the late Middle Ages, the latter two series at a seasonal

resolution (Figure 2).

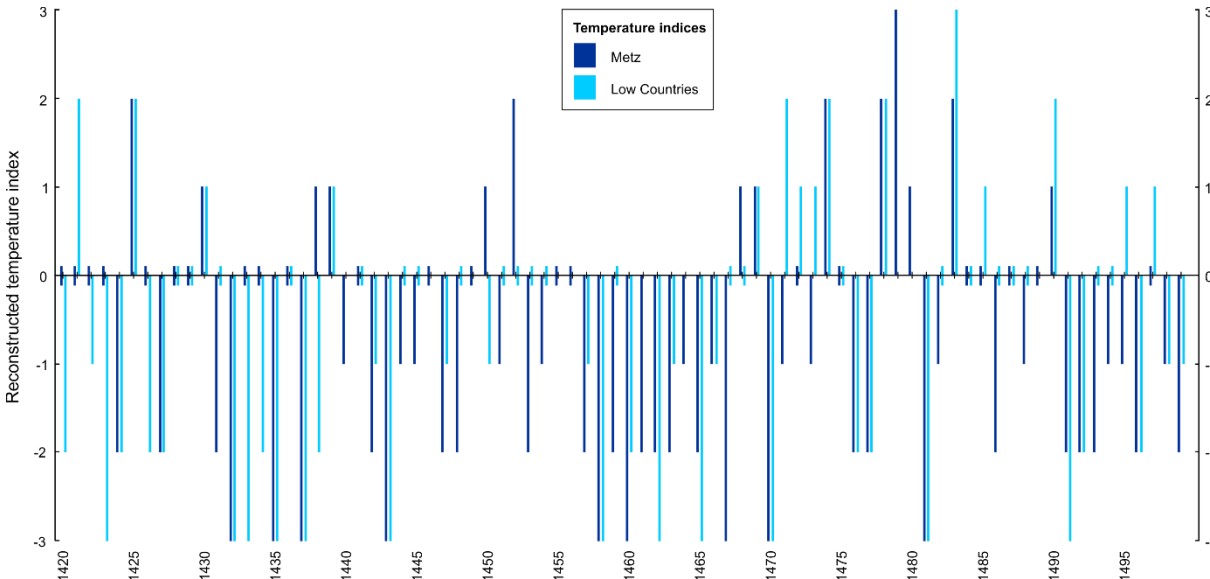

**Figure 2**: Comparison of seven-point winter temperature indices for Metz (Litzenburger, 2015) and the Low Countries

(Belgium, Luxembourg and The Netherlands; Camenisch, 2015a) for the period 1420-1500, reconstructed using the Pfister

index approach. Zero values for specific years are indicated by a small bar.

In addition to these studies, four other approaches exist for Europe: (i) IJnsen's temperature index (IJnsen and Schmidt,

1974) consists of a nine-point scale, which was also adopted by Van Engelen et al. (2001); (ii) Alexandre (1987) used a five-

point scale seasonal index, with categories from -2 (very warm) to + 2 (very cold) and 0 being attributed to non-documented

seasons; (iii) Fernández-Fernández et al. (2014; 2017) used a three-point-scale: (+1: warmer than usual; 0:  normal; -1:

colder than usual) and (iv) Domínguez-Castro et al. (2015) a five-point index (+2: very hot; +1: hot; 0: normal; -1: cold; -2:

very cold). As noted in section 2.1, Klemm (1970) proposed a two-point index (warm/cold) for winter conditions.

### 2.3. Precipitation indices

Many of the authors mentioned in section 2.2 have also published precipitation indices. These reconstructions are usually

based on the same source materials as the temperature indices (an exception being Dobrovolný et al., 2015). However, for

certain regions, very specific source types exist that are more favourable for precipitation reconstructions than temperature –

see, for example, the precipitation series for the Mediterranean based on the analysis of urban annals, religious chronicles

and books of church and city archives (e.g. Rodrigo et al., 1994; Rodrigo et al., 1998; Rodrigo et al., 1999; Rodrigo and

Barriendos, 2008; Fernández-Fernández et al., 2014; Domínguez-Castro et al., 2015; Fernández-Fernández et al., 2015).

These series span various periods of the 16th to 20th centuries and, in some cases, overlap with instrumental data.

Often the same scale is applied for both temperature and precipitation indices; however, in certain regions, precipitation

indices may show more gaps than their temperature counterparts as data may be seasonal or more sporadic. The studies by

Van Engelen et al. (2001), Alexandre (1987), Fernández-Fernández et al. (2014; 2017) and Domínguez-Castro et al. (2015)

are exceptions, in that each adopted a different or more rudimentary scale for precipitation compared to their temperature

reconstructions. Van Engelen et al. (2001) opted for a five-point scale for precipitation compared to a nine-point scale for

temperature, and Alexandre (1987) a three-point rather than five-point index. Alexandre's (1987) precipitation index is also

relatively simple and separates events by their nature (1: Snow; 2: Rain; 3: Dry conditions) rather than intensity. Fernández-

Fernández et al. (2014; 2017) used a two-point scale (0: total absence of rain; 1: occurrence of rain) and Domínguez-Castro et al. (2015) a four-point scale.

Index series based on historical records of religious rogation ceremonies warrant separate discussion. Rogations are liturgical acts conducted to request either rainfall during a drought (termed *pro-pluvia* rogations) or an end to excessive or persistent precipitation (*pro-serenitate* rogations), and were used as an institutional mechanism to address social stress in response to such meteorological extremes (see Martín-Vide and Barriendos, 1995; Barriendos, 2005; Tejedor et al., 2019). Analyses of the occurrence and nature of rogation ceremonies have proven particularly valuable for western Mediterranean regions (most notably the Iberian Peninsula), where they have been used to create precipitation indices spanning the 16th to 19th centuries (e.g. Álvarez Vázquez, 1986; Martín-Vide and Vallvé, 1995; Barriendos, 1997, 2010; Gil-Guirado et al., 2019). In some cases, information about rogation ceremonies has been combined with climate-related narrative evidence to generate precipitation series (e.g. Fragoso et al., 2018). Useful evaluations of different indexing methods are provided by Domínguez-Castro et al. (2008) and Gil-Guirado et al. (2016). For a discussion of the use of rogation ceremonies as a proxy for drought see section 2.5, and for examples of rogation-based reconstructions in Mexico and South America see section 5.

**2.4. Flood indices**

Flood events – the result of short periods of heavy precipitation and/or prolonged rainfall – can also be classified using indices. The basis of European flood indices include descriptive accounts, administrative records such as bridge master's accounts (e.g. those in Wels, Austria, which span the period 1350-1600 CE; Rohr, 2006, 2007, 2013), historic flood marks and river profiles (Wetter et al., 2011; spanning 1268-present and overlapping with instrumental data). In some regions, the availability and characteristics of sources may vary, and certain source types may be more important for flood reconstruction than others. This is, for instance, the case in Hungary, where charters play a particularly important role in flood reconstruction (Kiss, 2019; for the period 1001-1500 CE).

The scales used for flood reconstruction differ slightly from those used for the reconstruction of temperature and precipitation. Drawing on Brázdil et al. (1999; which spans the 16th century), scholars mainly from Central Europe (e.g. Sturm et al., 2001 [for the period 1500 CE-present]; Glaser and Stangl, 2003; 2004 [1000 CE-present]; Kiss, 2019) and France (Litzenburger, 2015) have applied a three-point scale. In contrast, Pfister (1999), Wetter et al. (2011) and Salvisberg (2017; 1550-2000 CE) used a five-point scale for floods of the River Rhine in Basel and the River Gürbe in the vicinity of Bern. The French historian Emmanuel Garnier also developed a five-point scale to reconstruct flood time-series from 1500 to 1850 CE, taking into consideration the spatial extent and economic consequences of each event (Garnier, 2009, 2015). A novel feature of the Garnier index is that it includes a -1 value for events where intensity cannot be estimated through documentary sources. Rohr (2006, 2007, 2013) chose a four-point scale for his flood reconstruction of the river Traun in Wels (Austria). In many cases, the index values express the amount of flood damage and/or the duration of flooding in combination with the geographical extent (e.g. Pfister and Hächler, 1991 [covering the period 1500-1989 CE]; Salvisberg, 2017; Kiss, 2019). Comprehensive overviews of flood reconstruction, including the index method, are given in Glaser et al. (2010), Brázdil et al. (2012) and recent work by the PAGES Floods Working Group synthesised in Wilhelm et al. (2018).

**2.5. Drought indices**

Drought events are closely linked to precipitation variability. As a result, many analyses of historical European droughts use indices adapted from precipitation reconstructions. Evidence of past droughts can be found in administrative sources, diaries, newspapers, religious sources and epigraphic evidence (see Brázdil et al., 2005; Brázdil et al., 2018; Erfurt and Glaser, 2019 [which spans the period 1800 CE-present]). Different approaches exist in historical climatology to express the severity of droughts in index form. Brázdil and collaborators (2013b) proposed a three-point scale (-1: dry; -2: very dry; -3: extremely dry) adapted from the precipitation indices described in section 2.3. Dry periods appear only in the drought index if they last

for at least two successive months. A similar approach is used by Pfister et al. (2006), Camenisch and Salvisberg (2020; covering 1315-1715 CE) and Bauch et al. (2020; 1200-1400 CE). However, Garnier (2018) applies a five-point scale with an additional sixth category for known drought-years with insufficient evidence for a more precise classification.

Drought indices have also been derived for the Western Mediterranean using records of rogation ceremonies, with specific methodologies developed to estimate the length, severity and continuity of drought episodes (see Domínguez-Castro et al., 2008). A number of studies have used evidence of *pro-pluvia* ceremonies (see section 2.3) as a drought proxy (Piervitali and Colacino, 2001; Domínguez-Castro et al., 2008; Domínguez-Castro et al., 2010; Garnier, 2010; Domínguez-Castro et al., 2012b; Tejedor et al., 2019), sometimes in combination with other narrative evidence (e.g. Fragoso et al., 2018; Gil-Guirado et al., 2019). Readers are referred to Brázdil et al. (2018) for a detailed discussion of the different types of drought indices.

**2.6. Other indices**

In Europe, the index method has only rarely been applied in contexts other than for temperature, precipitation, flood and drought reconstruction. Pichard and Roucaute (2009) developed, for example, an index for snowfall in the French Mediterranean region since 1715 CE, including ordinal categories escalating from 1 to 3 depending on the event duration and quantity of snow fallen. This study is based on information from diaries and other urban documentary sources. Marie-Luise Heckmann (2008, 2015), coming from the field of historical seismology and seemingly unconnected to discussions in historical climatology, developed a combined temperature/precipitation index that differentiates winters and summers by weather description and phenological phenomena; this index was applied to documentary data from late-medieval Prussia and Livonia (1200-1500 CE). *Pro-pluvia* rogation ceremonies have been analysed as a proxy for the winter North Atlantic Oscillation between 1824 and 1931 CE in the Extremedura region of Spain (Bravo-Paredes et al., 2020).

Sea ice reconstructions for the seas around Iceland have been developed by Astrid Ogilvie, the pioneer of Icelandic climate history (Ogilvie, 1984, 1992; Ogilvie and Jónsson, 2001). She developed a monthly resolution sea-ice index based on historical observations in 37 sectors of the sea around Iceland (Ogilvie, 1996), including sightings of sea-ice in ships' logbooks, whalers' and sealers' charts, diaries, letters, books and newspapers. The index values hence vary from 1 to (theoretically) 37, with data weighed by source reliability. Pre-1900 CE records report single observations of icebergs and varying concepts of sea-ice have to be taken into consideration. The record is presented as a 5-year summarised value for the period 1600-1784 CE, with monthly and annual values given from 1785 to present.

**3. Climate indices in Asia**

**3.1. Origins of documentary-based indices in Asia**

The use of the index approach in Asia is limited to research in China and India. With the exception of Japan, historical climatology research is either in its infancy or completely absent in other parts of the continent (Adamson and Nash, 2018). Very little work to reconstruct climate from documentary sources has occurred in southeast Asia, for example, and efforts to utilise records from the Byzantine Empire (Telelis, 2008; Haldon et al., 2014) and Muslim world (e.g. Vogt et al., 2011; Domínguez-Castro et al., 2012a) are only recently emerging. In Korea, only Kong and Watts (1992) have developed anything resembling climate indices, categorising individual years as warm/cold or dry/humid using information from diaries and histories.

Climate reconstruction work in China has developed largely independently from European historical climatology traditions. The Central Meteorological Bureau of China has published several fundamental works on Chinese wet/dry series. In 1981, a milestone work showed 120 cities with a five-point wet/dry series for the whole of China spanning the period 1470 to 1979 CE (Central Meteorological Bureau of China, 1981). Nowadays, most reconstructions (including coldness, drought, frost,

hail and others) are based on the *Compendium of Chinese Meteorological Records of the Last 3,000 Years* edited by Zhang De'er (2004). This compendium provides details of a wide range of historical meteorological phenomena from across China at a daily level. However, due to an imbalance in population distribution, records are more abundant for eastern than western China (Ge et al., 2013). In India, the only study to use an index approach (Adamson and Nash, 2014) was developed from Nash and Endfield's work in southern Africa (see section 4); there were, however, several differences in approach, notably the inclusion of calibration tables.

One country where the field of historical climatology is relatively well-developed is Japan. Japan has weather data recorded in documents dating back to at least 55 CE (Ingram et al., 1981), and diaries in particular have been utilised to reconstruct climate conditions (e.g. Mikami, 2008; Zaiki et al., 2012; Ichino et al., 2017; Shō et al., 2017). Access to documentary data on past weather phenomena is provided by detailed collections that evaluate historical sources (Mizukoshi, 2004-2014; Fujiki, 2007). However, Japanese historical climatology has no tradition of using indices, instead tending to use information in documentary sources to reconstruct units of meteorological measurement such as temperature and precipitation directly. For example, Mikami (2008) correlated mean monthly summer temperature with number of rain days. Mizukoshi (1993) and Hirano and Mikami (2008) used historical records to provide detailed reconstructions of weather patterns. Mizukoshi (1993) divided rainy seasons into three types: "heavy rain type", "light rain type" and "clear rainy season type", although these are not indices *per se*. In a similar way, Itō (2014) distinguished precipitation in categories such as "persisting rainfall" or "long downpour", depending on seven keywords for each category. He used a similar approach to define indicators for cold spells, using keywords such as "cold", "frost", and "put on cotton [clothes]". This keyword method for climatic conditions is also applied by Tagami (2015). There has also been much effort to reconstruct climate from climate-dependent phenomena such as cherry blossom or lake freezing dates (e.g. Aono and Kazui, 2008; Mikami, 2008; Aono and Saito, 2010).

**3.2. Types of documentary evidence used to create index series**

Historical climate index development in India has used a similar range of sources to those noted above for Europe – specifically newspapers and private diaries spanning the period 1781 to 1860 CE, supplemented by government records, missionary materials and some reports (Adamson and Nash, 2014). The sources used for the development of climate indices in China, however, are very different and require further explanation.

The earliest known written weather records in China, inscribed onto oracle bones, bronzes and wooden scripts, date to the Shang dynasty (~1600 BCE). These records were intended for weather forecasting, but later included actual weather observations (Wang and Zhang, 1988). Emperors of succeeding dynasties compiled more systematic records to allow them to better understand the weather, forecast harvests and hence maintain social stability (Tan et al., 2014). Some scholars use an old Chinese concept of *Tien* (or *Tian*, meaning Heaven) to explain the tradition. *Tien* was viewed as a medium used by gods and divinities to forward messages. Natural hazards (e.g. droughts and floods) were regarded as displaying *Tien*'s displeasure with the emperor and his court and were often followed by uprisings and rebellions (Perry, 2001; Pei and Forêt, 2018). To help them understand the long-term pattern of such hazards, imperial governments appointed specialists such as *Taishi* (imperial historians) or *Qintian Jian* (imperial astronomers) to record unusual and/or extreme weather events. Later, related environmental and socioeconomic events, such as early or late blossoming, agricultural conditions, famine, plagues and locust outbreaks, were also recorded (see Wang et al., 2018, for further details). This long tradition of chronicling has resulted in an exceptional range of materials for understanding and reconstructing past climates. It is worth noting, however that – due to a desire in imperial China to generalise details (Hansen, 1985) – phenomena were often only recorded as narrative descriptions with magnitude categorised as large, medium, or small.

The earliest official chronicle was *Han Shu* ('The Book of Han') written by Ban Gu (32-92 CE). However, many earlier historical books incorporate climate observations, including *Shi Ji* ('Records of the Grand Historian') by Sima Qian (145-86

BCE) and *Chun Qiu* ('Spring and Autumn Annals') compiled by Confucius (551-479 BCE) for the history of the *Lu Kingdom* (722- 481 BCE) (Wang and Zhang, 1988). Classic literature called *Jing Shi Zi Ji* was compiled in *Si Ku Quan Shu* ('Complete Library in Four Branches of Literature') published in 1787 (full-text digital versions are accessible at websites including Scripta Sinica: http://hanchi.ihp.sinica.edu.tw/ihp/hanji.htm). The *Shi* (meaning 'history') branch contains, but is not limited to, the 'Twenty-Four Histories' (later expanded to 'Twenty-Five Histories' by adding *Qing Shi Gao*, the 'Draft History of Qing'), other historical books, documents of the central administration, local gazettes and private diaries (Ge et al., 2018).

While providing consistency in recording practices, the spatial coverage of official historical books was often limited to national capitals or other important locations. However, the writing of *Fang Zhi* – local chronicles or gazettes, popular in the Ming (1368-1643 CE) and especially Qing (1644-1911 CE) dynasties – substantively expanded the availability of documentary sources. Local gazettes contain unusual weather- and climate-related statements like those in the official chronicles, but incorporate additional details at provincial, prefectural, county or township levels depending on the local administrative unit. For more information, see Ge et al. (2018) and a database of local gazettes at http://lcd.ccnu.edu.cn/#/index.

In the 1980s, the Central Meteorological Bureau of China initiated a massive project for the compilation of weather- and climate-related records. The work resulted in the most influential publication in contemporary Chinese climate literature, The *Compendium of Chinese Meteorological Records of the Last 3,000 Years* edited by Zhang De'er (2004); this contains more than 150,000 records quoted from 7,930 historical documents, mostly local gazettes. To maximise the availability of the compendium, Wang et al. (2018) have digitised the records into the REACHES database (Figure 3). The quantity of records peaks in the last six hundred years, during the Ming and Qing dynasties. This is due to a large number of local gazettes spread across the country; however, only a few are available for the Tibetan Plateau and arid western regions. The Institute of Geographic Sciences and Natural Resources Research (Chinese Academy of Sciences) has also collated phenological records from historical documents (Zhu and Wang, 1973; Ge et al., 2003).

Two sources of documentary evidence are of particular importance for historical climate reconstruction in China. Daily observations of sky conditions, wind directions, precipitation types and duration are recorded in *Qing Yu Lu* ('Clear and Rain Records') (Wang and Zhang, 1988). The records, however, are descriptive and only available for selected areas; these include Beijing (1724-1903 with six missing years), Nanjing (1723-1798), Suzhou (1736-1806), and Hangzhou (1723-1773). *Yu Xue Fen Cun* ('Depth of Rain and Snow') reported the measured depth of rainfall infiltration into the soil or depth of snow accumulation above ground in the Chinese units *fen* (~3.2mm) and *cun* (~3.2cm). From 1693 to the end of the Qing dynasty in 1911, these measurements were taken in eighteen provinces; however, many records include imprecise measurements and/or dates (Ge et al., 2005; Ge et al., 2011). Despite their descriptive and semi-quantitative nature, the two documentary sources are valuable for reconstructing past climate, especially for summer precipitation (Gong et al., 1983; Zhang and Liu, 1987; Zhang and Wang, 1989; Ge et al., 2011) and meiyu (or 'plum rains', marking the beginning of the rainy season; see Wang and Zhang, 1991) in different cities depending on the record length as described above. They are also useful for cross checking and/or validating climate indices derived from other documentary sources.

**3.3. Temperature indices**

The availability of documentary temperature indices for Asia is restricted to China. Zhu (1973) was the first Chinese scholar to use historical weather records and phenological evidence to identify temperature variability over the last 5,000 years (~3000 BCE to 1955 CE). He consulted a range of data sources for his reconstruction, including the dates of lake/river freezing/thawing, the start/end dates of snow and frost seasons, arrival dates of migrating birds, the distribution of plants

such as bamboo, lychee and orange, the blossoming dates of cherry trees and harvest records. However, the study did not

clearly indicate his methodology.

Winter temperature anomalies were initially regarded as key indicators of temperature changes in China (Zhang and Gong,

1979; Zhang, 1980; Gong et al., 1983; Wang and Wang, 1990a; Shen and Chen, 1993; Ge et al., 2003), as (i) there were

more temperature-related descriptions in winter than in other seasons and (ii) winter temperatures have higher regional

uniformity than summer temperatures (Wang and Zhang, 1992). However, this uniformity mainly reflects changes in the

Siberian High system, so reconstructions of summer (and other season) temperature and precipitation anomalies to reflect

other aspects of monsoon circulation soon received increasing attention (see, for example, Zhang and Liu, 1987; Wang and

Wang, 1990b; Yi et al., 2012).

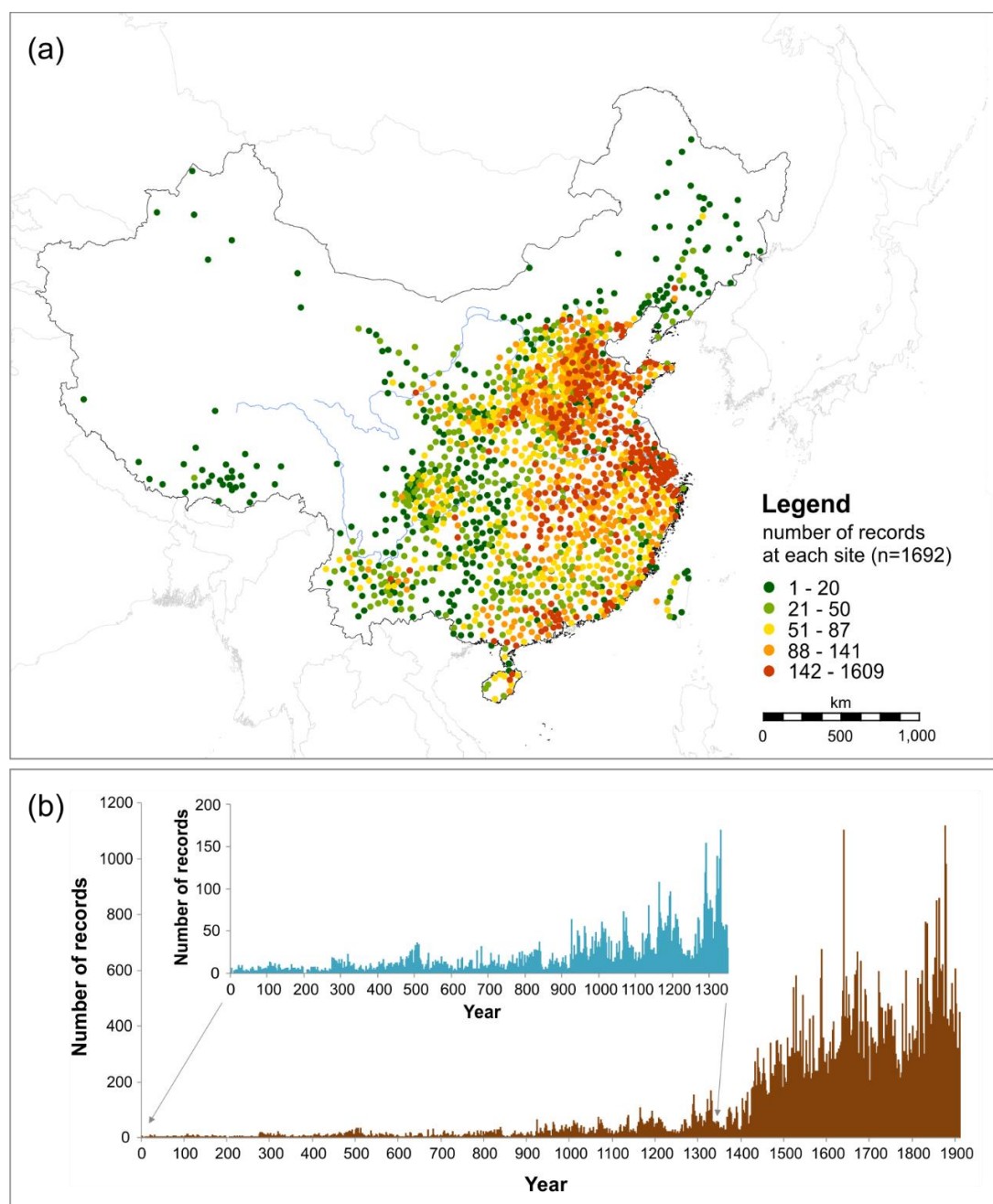

**Figure 3**: Numbers of historical documentary records in the REACHES database for China. (a) Spatial distribution of

records at 1,692 geographical sites across China. (b) Temporal evolution of the records in the database from 1 to 1911 CE

(brown series); inset (blue series) shows the same data for 1 to 1350 CE but with an expanded vertical axis.

Zhu's (1973) pioneering work has had a great influence upon the development of historical climatology in China. Successive studies used a similar approach to reconstruct winter temperature indices for every decade from the 1470s to 1970s by counting the frequency of years with cold- or warm-related records (Zhang and Gong, 1979; Zhang, 1980; Shen and Chen, 1993; Zheng and Zheng, 1993). Zhang (1980) adopted binary (cold/warm) categories and further developed an equation to derive decadal temperature indices for the period 1470-1970 CE (see Section 8.2); this approach was applied in several studies (Gong et al., 1983; Wang and Wang, 1990b; Zheng and Zheng, 1993; Man, 1995).

The formal development of an ordinal-scale temperature index was first introduced by Wang and Wang (1990b) who used a four-point scale to build decadal winter cold index series for the period 1470-1979 CE in eastern China (0: no or light snow or no frost; 1: heavy snow over several days; 2: heavy snow over months; 3: heavy snow and frozen ground until the following spring). This approach was widely applied in subsequent series in different regions, for different seasons and at differing temporal resolutions (Wang and Wang, 1990a; Wang and Wang, 1990b; Wang et al., 1998; Wang and Gong, 2000; Tan and Liao, 2012; Tan and Wu, 2013). For example, Wang and Gong (2000) developed a fifty-year resolution winter cold index for eastern China spanning the period 800-2000 CE. Tan and colleagues adapted the approach to reconstruct decadal temperature index series (-2: rather cold; -1: cold; 0: normal; 1: warm) in the Ming (1368-1643 CE; Tan and Liao, 2012) and Qing dynasties (1644-1911 CE; Tan and Wu, 2013) in the Yangtze delta region.

**3.4. Drought/flood and moisture indices**

China has a particularly rich legacy of documents describing historical floods and droughts, and using such records to define drought-flood series has a long tradition. Zhu (1926) and Yao (1943) presented the earliest drought-flood series for all of eastern China (206 BCE-1911 CE), although their temporal and spatial resolutions are vague. Due to the higher number of available records for the last several hundred years, reconstructions using frequency counts were avoided in their series; instead the ratio between flood and drought events was used to build moisture indices (see section 8.2). Examples of other early studies include Yao (1982), Zhang and Zhang (1979), Zheng et al. (1977) and Gong and Hameed (1991).

Beginning in the 1970s, the Central Meteorological Administration initiated a project to reconstruct historic annual precipitation. This adopted a five-point ordinal scale (1: very wet; 2: wet; 3: normal; 4: dry; 5: very dry) to form drought-flood indices for 120 locations in China spanning the period 1470-1979 CE (Academy of Meteorological Science of China Central Meteorological Administration, 1981). The indices were compiled based on the evaluation of historical descriptions (section 8.2), with the series later extended to 2000 CE (Zhang and Liu, 1993; Zhang et al., 2003). Most reconstructions in China now use this five-point index (Zheng et al., 2006; Tan and Wu, 2013; Tan et al., 2014; Ge et al., 2018). For example, Zhang et al. (1997) used the approach to establish six regional series of drought-flood indices for eastern China (from the North China Plain to the Lower Yangtze Plain) spanning the period 960-1992 CE. Zheng et al. (2006) developed a dataset covering 63 stations across the North China Plain and the middle and lower reaches of the Yangtze Plain and reconstructed a drought-flood index series spanning 137 BCE to 1469 CE.

Adamson and Nash (2014) also adopted a five-point index series when reconstructing monsoon precipitation in western India (Figure 4). Where data quality allowed, indices were derived for individual 'monsoon months' (May/June, July, August and September/October) and summed to produce an index value for each entire monsoon season. Where monthly-level indices could not be constructed, indices pertaining to the whole monsoon were generated directly from narrative evidence. The five-point index was chosen to correspond with the terminology currently used by the Indian Meteorological Department for their seasonal forecasts (from 'deficient' to 'excess' rainfall) and regular reports of rainfall conditions (a 4-point scale from 'scanty' to 'excess', with a fifth category 'heavy' added by the authors). As each of these correspond to percentage deviations from a rainfall norm, this allowed the generation of calibration tables within an instrumental overlap

period, to assign descriptive terms to specific index points (e.g. the term 'seasonable rain' to the category +1 'excess'). This should allow the same methodology to be repeated elsewhere in India but limits the methodology to the subcontinent.

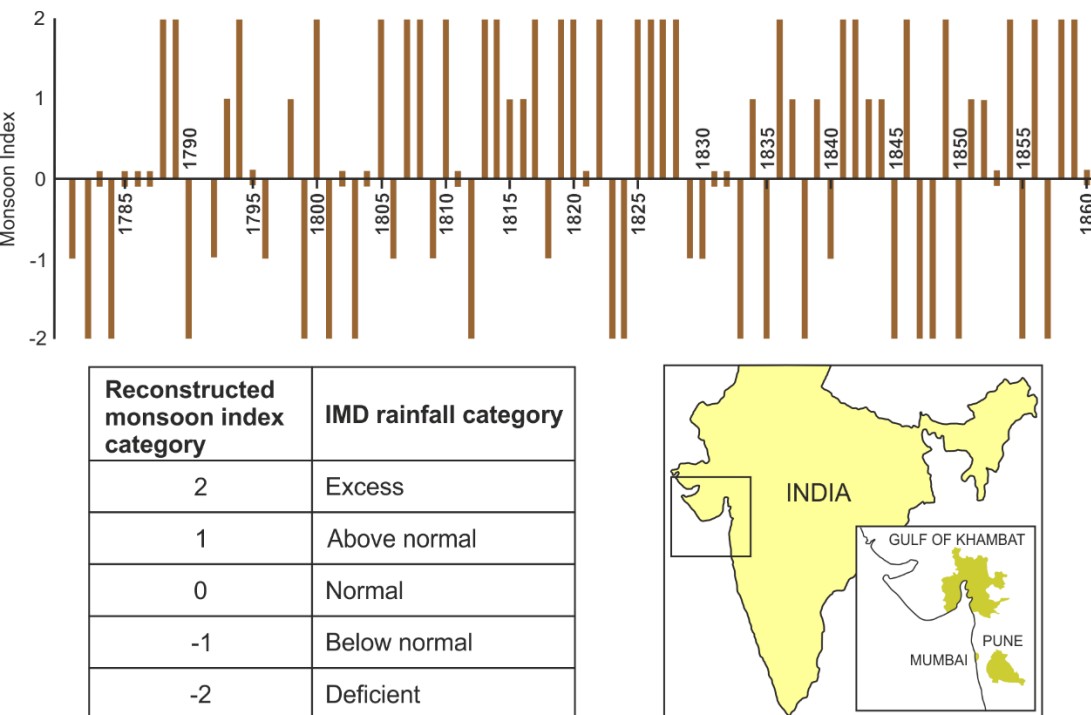

**Figure 4**: Five-point Western India Monsoon Rainfall reconstruction for 1780-1860. The reconstruction is a combination of separate series for Mumbai, Pune and the Gulf of Khambat (see inset). Monsoon index categories map broadly onto Indian Meteorological Department (IMD) descriptors of seasonal monsoon rainfall (data for reconstruction from Adamson and Nash, 2014). Zero values are shown as small bars; years with insufficient data to generate an index value are left blank.

### 3.5. Other series

Several other studies have used weather descriptions within documentary records to reconstruct past climate series in China. These include reconstructed winter thunderstorm frequency (Wang, 1980, spanning 250 BCE-1900 CE), dust fall (Zhang, 1984, for the period 1860-1898 CE; Fei et al., 2009, for the past 1700 years) and typhoon series in Guangdong (Liu et al., 2001, 1000-1909 CE) and coastal China (Chen et al., 2019, 0-1911 CE). Many scholars have also used information in *Qing Yu Lu* and *Yu Xue Fen Cun* to count and build winter snowfall days series (Zhou et al., 1994; Ge et al., 2003), while Hao et al. (2012) have further used the series to regress annual winter temperatures over the middle and lower reaches of the Yangtze River since 1736.

Phenology-related phenomena have also been widely used in China to indicate past climate variability (Liu et al., 2014). Flower blossom dates in Hunan between 1888 and 1916 (Fang et al., 2005) and in the Yangtze Plain from 1450 to 1649 (Liu, 2017) were used to indicate temperature change. The date of the first recorded 'song' of the adult cicada has also been used to reconstruct precipitation change during the rainy season in Hunan from the late 19th to early 20th century (on the principle that cicada growth to adulthood requires sufficient humidity, and this coincides with the peak rainy season; Xiao et al., 2008). In recent years, researchers have been able to reconstruct various series including typhoons (Chen et al., 2019; Lin et al., 2019) and droughts (Lin et al., 2020) from the compendium of Chinese records compiled by Zhang (2004).

Using descriptions of agricultural outputs in the *Twenty-Four Histories* and *Qing History*, Yin et al. (2015) developed a grain harvest yield index and used this to infer temperature variations from 210 BCE to 1910 CE. Details of outbreaks of Oriental migratory locusts in these same histories have been used by Tian et al. (2011) to construct a 1910-year-long locust index

through which precipitation and temperature variations can be inferred. *The History of Natural Disasters and Agriculture in Each Dynasty of China*, published by the Chinese Academy of Social Science (1988), includes details of disasters such as famines to reconstruct indices of climate variability during the imperial era.

## 4. Climate indices in Africa

### 4.1. Origins of documentary-based indices in Africa

Compared to the wealth of documentary evidence available for Europe and China, there are relatively few collections of written materials through which to explore the historical climatology of Africa (Nash and Hannaford, 2020). The bulk of written evidence stems from the late 18th century onwards, with a proliferation of materials for the 19th century following the expansion of European missionary and other colonial activity.

Most historical rainfall reconstructions for Africa use evidence from one or more source type. A small number of studies are based exclusively upon early instrumental meteorological data. Of these, some (e.g. the continent-wide analysis by Nicholson et al., 2018) combine early rain gauge data with more systematically collected precipitation data from the 19th to 21st centuries, to produce quantitative time series. Others, such as Hannaford et al. (2015) for southeast Africa, use data digitised from ships' logbooks to generate quantitative regional rainfall chronologies. Most climate reconstructions, however, make use of narrative accounts to develop relative rainfall chronologies based on ordinal indices, either for the whole continent or for specific regions.

While drawing upon European traditions and sharing many similar elements, methodologies for climate index development in Africa have evolved largely in isolation from approaches in Europe (see section 8.3). The earliest work by Sharon Nicholson, for example, was published around the same time that Hubert Lamb was developing his index approach (Nicholson, 1978a, 1978b, 1979, 1980). Her early methodological papers on precipitation reconstruction (Nicholson, 1979, 1981, 1996) use a qualitative approach to identify broadly wetter and drier periods in African history. A seven-point index (+3 to -3) integrating narrative evidence with instrumental precipitation data was introduced in Nicholson (2001) and expanded in Nicholson et al. (2012a) and Nicholson (2018).

The many regional studies in southern Africa owe their approach to the work of Coleen Vogel (Vogel, 1988, 1989), who drew on Nicholson's research but advocated the use of a five-point index to classify rainfall levels in the Cape region of South Africa (+2: very wet, severe floods; +1: wet, good rains; 0: seasonal rains; -1: dry, months of no rain reported; -2: very dry, severe drought). Subsequent regional studies, starting with Endfield and Nash (2002) and Nash and Endfield (2002), have adopted the same five-point approach.

### 4.2. Precipitation indices

The main continent-wide index-based series for Africa originate from research undertaken by Sharon Nicholson (e.g. Nicholson et al., 2012a). This series uses a seven-point scale and has been used to explore both temporal (Figure 5) and spatial (Figure 6) variations in historical rainfall across Africa during the 19th century. One regional rainfall reconstruction is available for West Africa (Norrgård, 2015, spanning 1750-1800 CE and using a seven-point scale ) and one for Kenya (Mutua and Runguma, 2020, spanning 1845-1976 CE with a five-point scale). The greatest numbers of regional reconstructions – all using a five-point scale – are available for southern Africa. These include chronologies covering all or part of the 19th century for the Kalahari (Endfield and Nash, 2002; Nash and Endfield, 2002, 2008) and Lesotho (Nash and Grab, 2010), and – most recently – Malawi (Nash et al., 2018) and Namibia (Grab and Zumthurm, 2018). Several reconstructions are available for South Africa, including separate 19th century series for the Western and Eastern Cape, Namaqualand and present-day KwaZulu-Natal (Vogel, 1988, 1989; Kelso and Vogel, 2007; Nash et al., 2016). Most studies,

including the continent-wide series, reconstruct rainfall at an annual level, but, where information density permits, it has been possible to construct rainfall at seasonal scales (e.g. Nash et al., 2016). Regional studies from southern Africa have recently been combined with instrumental data and other annually-resolved proxies (including sea surface temperature data derived from analyses of fossil coral) to produce two multi-proxy reconstructions of rainfall variability (Neukom et al., 2014a; Nash et al., 2016).

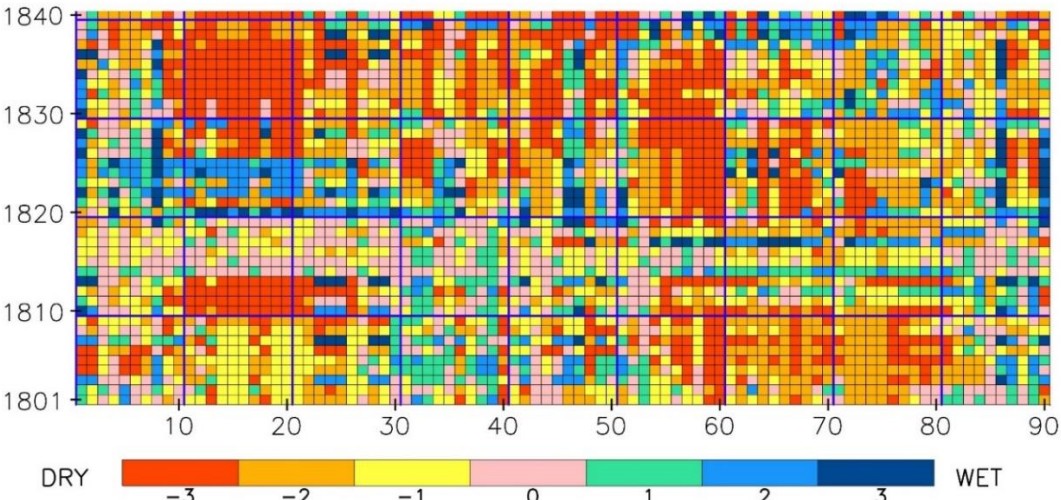

**Figure 5**: Seven-point "wetness" index series for 1801 to 1840 for the 90 homogenous rainfall regions of Africa indicated across the x-axis. This series is reconstructed using documentary and instrumental data, with data gaps infilled using substitution and statistical inference (see section 8.3 and Nicholson et al., 2012a). From left to right, the regions approximately extend by latitude from the northern (region 1 – Northern Algeria/Tunisia) to southern (region 84 – western Cape, South Africa) extremes of the continent. Anomalies in the numbering sequence are regions 85, 86, 90 (all equatorial Africa), 87 (eastern Africa) and 88, 89 (Horn of Africa).

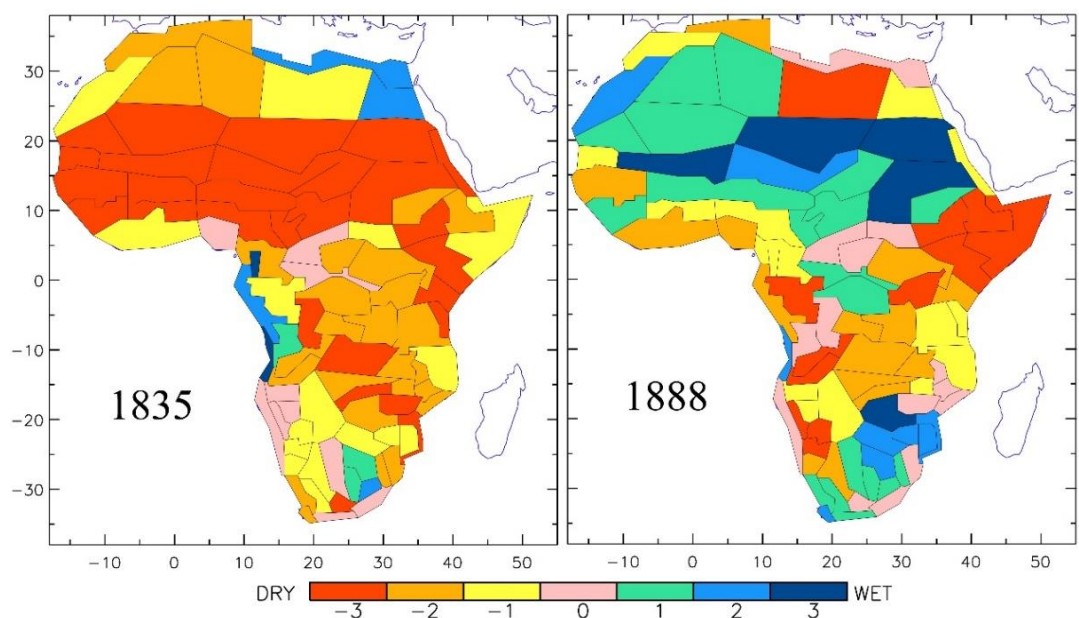

**Figure 6**: Rainfall anomaly patterns for 1835 and 1888 for the 90 homogenous rainfall regions of Africa delineated on the maps (modified after Nicholson et al., 2012b).

### 4.3. Temperature indices

To date, the only study exploring temperature variations in Africa using an index approach is an annually-resolved chronology of cold season variability spanning 1833-1900 CE for the high altitude kingdom of Lesotho in southern Africa (Grab and Nash, 2010). This uses a three-point index for winter severity (normal/mild; severe; very severe) and identifies more severe and snow-rich cold seasons during the early- to mid-19th century (1833-1854) compared with the latter half of the 19th century (Figure 7). A reduction in the duration of the frost season by over 20 days during the 19th century is also identified.

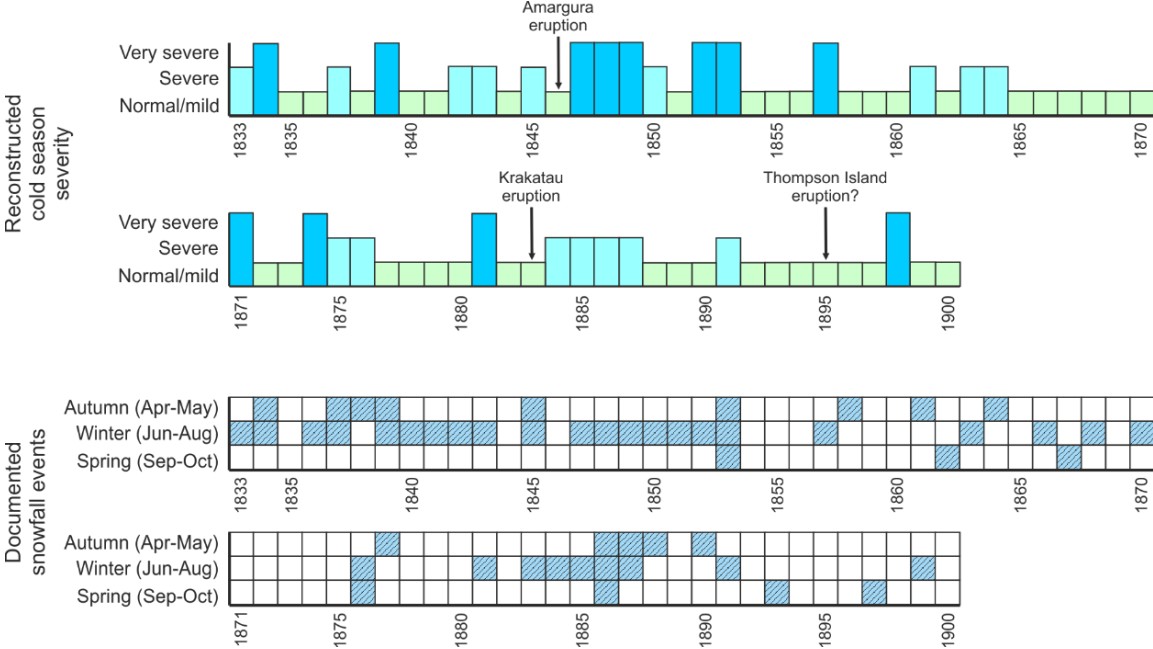

**Figure 7**: Three-point "cold season severity" index for Lesotho and surrounding areas during the 19th century (top), with major volcanic eruptions indicated. The occurrence of snowfall events (bottom) during the same period is also shown (modified after Grab and Nash, 2010).

## 5. Climate indices in the Americas

### 5.1. Origins of documentary-based indices in the Americas

The use of the index approach in climate reconstruction is variable across the Americas. Although sufficient historical records exist in some regions, particularly the north-eastern United States since the 18th century, few researchers have generated climate indices for the USA or Canada (White, 2018). Mexico, in contrast, has produced pioneering studies in climate history, especially on extreme droughts (see Prieto and Rojas, 2018; Prieto et al., 2019). In South America, documentary evidence is overall lower in quality and quantity compared to Europe, so more complex indices have been replaced by simpler ones, which extend to the 1500s CE.

### 5.2. Temperature, precipitation and river-flow indices

The only index-based temperature and precipitation reconstructions for the USA and Canada are those produced by William Baron and collaborators. Although influenced by the work of Pfister, Baron (1980, 1982) used a distinct content analysis of weather diaries (see section 8.4) to produce open-ended seasonal indices of New England temperature and precipitation for 1620-1800 CE, a period overlapping with the first local instrumental temperature series (which began in the 1740s). He later

combined seasonal indices, early instrumental records and phenological observations to create annual temperature and precipitation series and reconstruct frost-free periods (Baron et al., 1984; Baron, 1989, 1995).

There are a number of valuable compilations of extreme droughts in Mexico (e.g. Florescano, 1969; Jáuregui, 1979; Castorena et al., 1980; Endfield, 2007) and research that has identified climate trends across the country for 1450-1977 CE (Metcalfe, 1987; Garza Merodio, 2002). Garza Merodio systemised the frequency and duration of climatic anomalies in the Basin of Mexico for 1530-1869 CE. García-Acosta et al. (2003) developed an unprecedented catalogue of historic droughts in central Mexico for 1450-1900 CE. Later work compared this information with a tree-ring series and found a significant correlation between major droughts and ENSO years over the same period (Mendoza et al., 2005). Mendoza et al. (2007) constructed a similar series of droughts on the Yucatan Peninsula for the 16th to 19th centuries. Garza Merodio (2017) improved this index and extended it back in time (see Hernández and Garza Merodio, 2010), based on the frequency and complexity of rogation ceremonies (16th to 20th centuries). This approach identified droughts in bishoprics and towns of Mexico. Most recently, Dominguez-Castro et al. (2019) developed series for rainfall, temperature and other meteorological phenomena for Mexico City using information recorded in the books of Felipe de Zúñiga and Ontiveros; these volumes provide meteorological data with daily resolution for the twelve years spanning 1775 to 1786 CE.

In South America, the most detailed available historical information is on the scarcity or abundance of water. For investigations into historical rainfall and river flow rates, most studies construct 5-7 classes of data with annual or seasonal resolution. For example, a number of flood series have been compiled for rivers in Argentina (Prieto et al., 1999; Herrera et al., 2011; Prieto and Rojas, 2012, 2015; Gil-Guirado et al., 2016) – see Figure 8. In Bolivia, Gioda and Prieto (1999) and Gioda et al. (2000) developed a precipitation series for Potosí beginning in 1574 CE. In northern Chile, Ortlieb (1995) also compiled a detailed precipitation series for the 1800s CE. In Colombia, Mora Pacheco has developed a drought series for the Altiplano Cundiboyacense spanning the period 1778-1828 CE (Mora Pacheco, 2018). Finally, Dominguez-Castro et al. (2018) present a precipitation instrumental series from Quito (1891-2015 CE) and a series of wet and dry extremes from rogation ceremonies from 1600 to 1822 CE. In contrast, temperature records are less reliable and generally begin with the earliest instrumental data in the late 1800s CE (Prieto and García-Herrera, 2009; Prieto and Rojas, 2018), but there are exceptions (e.g. Prieto, 1983, which covers the 17th and 18th centuries). Most temperature-related indices use three classes.

Some of the world's most important index-based chronologies of the El Niño Southern Oscillation (ENSO) derive from the analysis of ENSO-related impacts recorded in South American documentary evidence. This area of research was pioneered by William Quinn and colleagues (Quinn et al., 1987; Quinn and Neal, 1992), with Quinn's chronologies revised and improved by various authors using additional primary documentary sources (e.g. Ortlieb, 1994; Ortlieb, 1995, 2000; García-Herrera et al., 2008).

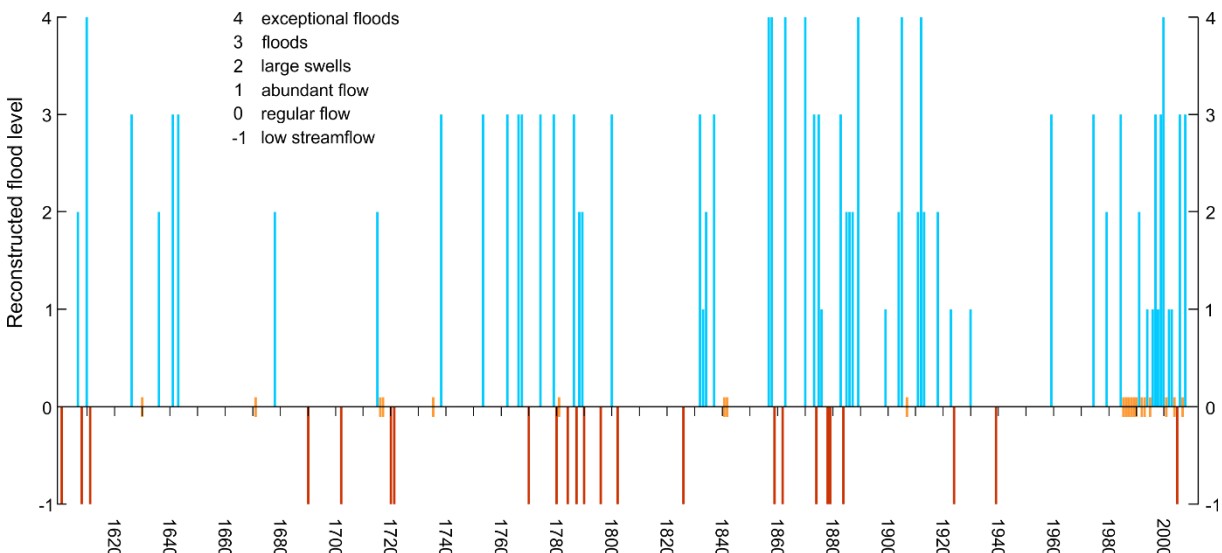

**Figure 8**: Six-point index series of historical flow in the Bermejo River (northern Argentina) between 1600 and 2008 CE
based on documentary evidence. These annual-level data were used to create the decadal-scale flood series in Prieto and
Rojas (2015). Zero values are indicated by short orange bars.

## 5.3. Sea-ice and snowfall indices

Relatively few studies have developed indices of winter conditions for the Americas. Building on their content analysis
approach and that of Astrid Ogilvie in Iceland (see section 2.6), Catchpole and Faurer (1983) and Catchpole (1995) produced
open-ended annual sea-ice indices for the western and eastern Hudson Bay, spanning the period 1751-1869 CE. A different
type of three-class index was developed for snowfall in the Andes at 33°S spanning 1600-1900 CE, based on the number of
months per year that the main mountain pass between Argentina and Chile was closed (Prieto, 1984).

## 6. Climate indices in Australia

### 6.1. Origins of documentary-based indices in Australia

Like Africa, Australia has a limited history of using documentary records for developing regional climate indices. Aside
from early compilations of 19th century colonial documents and newspaper records (Jevons, 1859; Russell, 1877), or climate
almanacs published by the Australian Bureau of Meteorology (Hunt, 1911, 1914, 1918; Watt, 1936; Warren, 1948), few
attempts were made in the 20th century to use historical sources to develop climate indices. Those that were developed
focussed predominantly on drought conditions (see, for example, Foley, 1957; McAfee, 1981; Nicholls, 1988). However,
considerable effort has been given in recent years to reconstruct climate variability in south-eastern Australia since British
colonisation in 1788 CE using both historical documents and instrumental observations (e.g. Gergis et al., 2009; Fenby,
2012; Fenby and Gergis, 2013; Gergis and Ashcroft, 2013; Ashcroft et al., 2014a; Ashcroft et al., 2014b; Gergis et al., 2018;
Ashcroft et al., 2019; Gergis et al., 2020). There have also been attempts to reconstruct storms and tropical cyclones along
the east coast of Australia (e.g. Callaghan and Helman, 2008; Callaghan and Power, 2011, 2014; Power and Callaghan,
2016), although these are not index-based.

Documentary-based indices for Australia have focussed on regional rainfall histories using largely material from previously
published drought and/or rainfall compilations (Fenby and Gergis, 2013). These compilations contained a vast collection of
primary source material including newspaper reports, unpublished diaries and letters, almanacs, observatory reports, 19th
century Australian publications and official government reports. For example, the seminal 19th century sources of Jevons

(1859) and Russell (1877), that formed the foundation of the Fenby and Gergis (2013) analysis, contain 79 primary sources, including 40 accounts from personal diaries, letters and correspondence between a range of people in the colony with the authors. Most recently, Gergis et al. (2020) compiled colonial newspaper and government reports to identify daily temperature extremes of snowfall and heatwaves from South Australia back to 1838. Although a temperature index has not yet been developed from this material, there is great potential to do so alongside recently homogenised 19th century instrumental temperature observations from the Adelaide region.

## 6.2. Precipitation and drought indices

The most extensive analysis of documentary records was compiled by Fenby (2012) and Fenby and Gergis (2013) as part of a large-scale project to reconstruct climate in south-eastern Australia using palaeoclimate, early instrumental and documentary data (Gergis et al., 2018). Fenby and Gergis (2013) used twelve secondary source compilations to collate monthly summaries of drought conditions experienced in five modern states in south-eastern Australia between 1788 and 1860 CE into a three-point index (wet, normal, drought). As explained in section 8.5, agreement between sources and several months of dry conditions was required before a period was considered a drought, rather than just 'normal' low summer rainfall. In coastal New South Wales, months of above average rainfall were only compiled where sufficiently detailed rainfall information was available (Fenby and Gergis, 2013). Given that Australian rainfall has high spatial variability, and many of the secondary sources only contained descriptions of localised floods or severe storm events, there were insufficient local reports from other regions to reconstruct larger-scale rainfall conditions using the sources considered.

To combine instrumental and documentary data into a single series spanning European settlement of Australia (1788 CE-present), Gergis and Ashcroft (2013) developed a three-point drought and wet year index based on instrumental rainfall observations from a five-station network in the Sydney region (spanning 1832-1859) and a 45-station rainfall network from across south-eastern Australia (1860-2008). As with the "wetness" index for Africa (Figure 5), the instrumental data were converted to an index so they could be combined with the documentary-based index of Fenby and Gergis (2013) to create a single, complete rainfall reconstruction. Good agreement was found during the overlapping period between instrumental and documentary-derived indices (1832-1860), and between the eastern New South Wales index and the wider south-eastern Australian indices. This provides some confidence that the two indices could be combined, and that data from the very early period, when only eastern New South Wales records are available, are indicative of conditions experienced in the broader region.

Given the exploratory nature of this work in south-eastern Australia, the aim of these studies was to use documentary and instrumental data to simply identify the occurrence of wet and dry years in the first instance, rather than develop a more finely resolved scale of the magnitude of the rainfall anomalies. The recent digitisation and analysis of daily instrumental rainfall data from Sydney, Melbourne and Adelaide (Ashcroft et al., 2019) provides an excellent opportunity to develop indices combining documentary and instrumental data from these regions in the future.

## 7. Climate indices and the world's oceans

### 7.1. Challenges in generating documentary-based indices for the world's oceans

The oceans constitute a challenging environment for historical climatologists. Written evidence of past weather at sea is generally local in scope, especially before the 17th century, and direct weather observations scarcely extend beyond the coast before the 15th century. Historical climatologists can use two categories of information to create reconstructions of past oceanic climate: (i) direct observations of weather, water, and sea ice conditions; and (ii) records of activities that were influenced by weather and water conditions. Such information can be found in documents written at sea (on ships, boats or,

from the twentieth century, submarines; Figure 9), documents written on the coast within sight of the sea, and documents written inland that record weather or activities at sea.

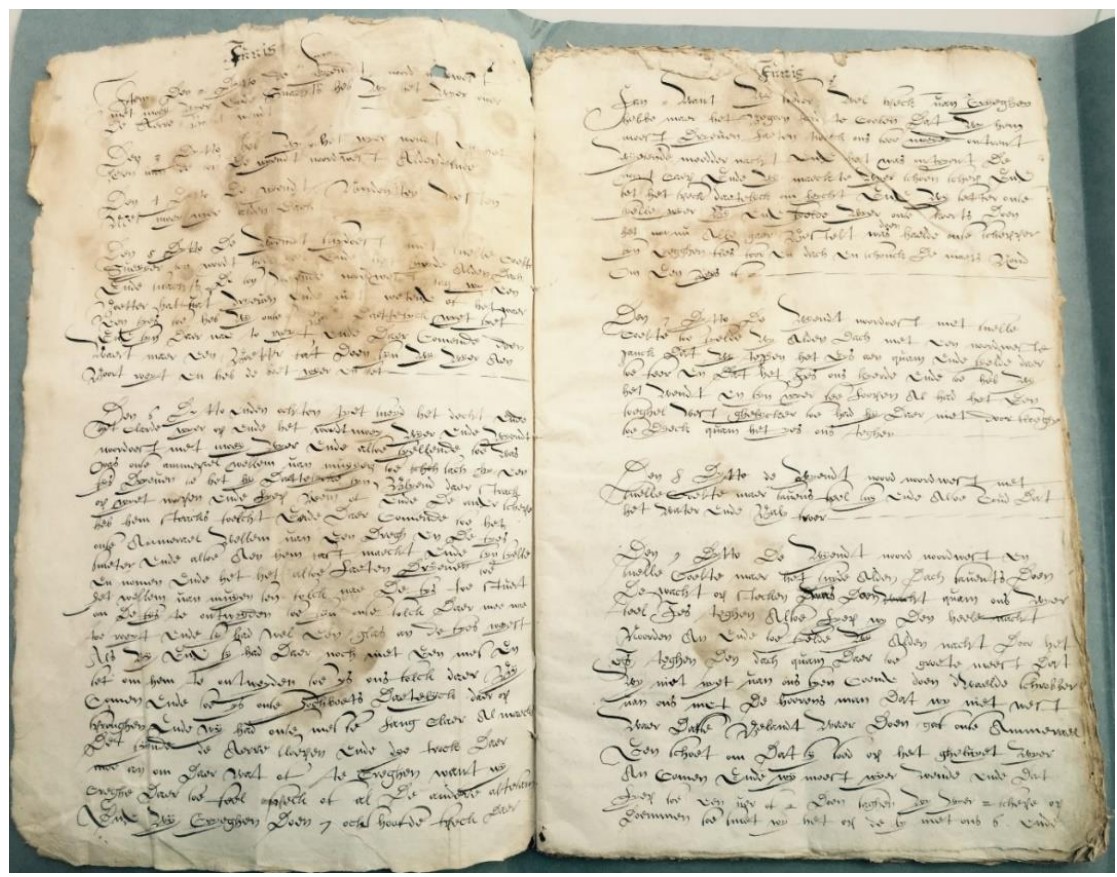

**Figure 9**: Journal written by a Dutch whaler during a voyage to the "Greenland Fishery," between Jan Mayen and Svalbard, 1615. Source: 0120 Oud archief stad Enkhuizen 1353-1815 (1872), Westfries Archief, Hoorn.

Between the 16th and 20th centuries, ships' logbooks are perhaps the most useful source type (see Wheeler, 2005a, 2005b; Wheeler and Garcia-Herrera, 2008; Ward and Wheeler, 2012; García-Herrera and Gallego, 2017; Degroot, 2018). Sailors originally recorded the speed and direction of the wind in order to calculate their location, and their compass-aided measurements of wind direction are often assumed to be true instrumental observations (Gallego et al., 2015). Yet naval officers on different ships in the same fleet could record slightly different measurements, and they did not always accurately estimate their longitude, or consistently describe whether recorded wind directions related to real or magnetic north (Wilkinson, 2009; García-Herrera et al., 2018). Logs kept by flag officers – which survive in larger quantities in early periods than logs kept by subordinate officers – may not include systematic weather observations. Ships did not sail in sufficient numbers prior to the 18th and 19th centuries for scholars to use surviving logbooks for comprehensive regional weather reconstructions, and many logbooks have been lost. Finally, logbooks written aboard some ships copied wind measurements earlier recorded in simple tables and should therefore be considered secondary sources for the purpose of climate reconstruction (Norrgård, 2017).

Logbooks of the 16th and 17th centuries, in particular, are most valuable when used alongside other documentary evidence. Journals kept during exceptional voyages may provide similar environmental data but in a narrative format. Accounts of the passage of ships through ports and tollhouses; the annual catch brought in by fishermen or whalers; or the duration of voyages may provide evidence of changes in the distribution of sea ice or patterns of prevailing wind. Correspondence, diary entries, intelligence reports, newspaper articles and chronicles may describe weather at sea, or weather blown in from the

sea, often at high resolution and occasionally for decades. Paintings, illustrations, and even literature may provide insights into the changing frequency or severity of weather events at sea. These sources can supplement other human records of the oceanic climate, including oral histories, or shipwrecks distributed in areas of heavy trade (Chenoweth, 2006; Trouet et al., 2016).

## 7.2. Indices of wind direction and velocity

If carefully contextualised, information in written records of oceanic weather – especially ships' logbooks and accounts of naval voyages – can be quantified and entered into databases. The Climatological Database of the World's Oceans (CLIWOC; Figure 10), for example, quantified nearly 300,000 logbooks from 1750 to 1850 CE, and their data are now among 456 million marine reports within the International Comprehensive Ocean-Atmosphere Data Set (ICOADS) (García-Herrera et al., 2005b; Koek and Konnen, 2005; García-Herrera et al., 2006). By using such datasets, or by creating databases of their own, scholars have reconstructed aspects of past climate at sea, in many cases verifying or extending reconstructions compiled by scientists using other means. High resolution reconstructions of regional trends in the frequency of winds from different directions, for example, reveal broadscale atmospheric circulation changes associated with stratovolcanic eruptions, ENSO, the North Atlantic Oscillation (NAO) or the monsoons of the Northern and Southern Hemispheres (e.g. Garcia et al., 2001; Küttel et al., 2010; Barriopedro et al., 2014; Barrett, 2017; Barrett et al., 2018; García-Herrera et al., 2018).

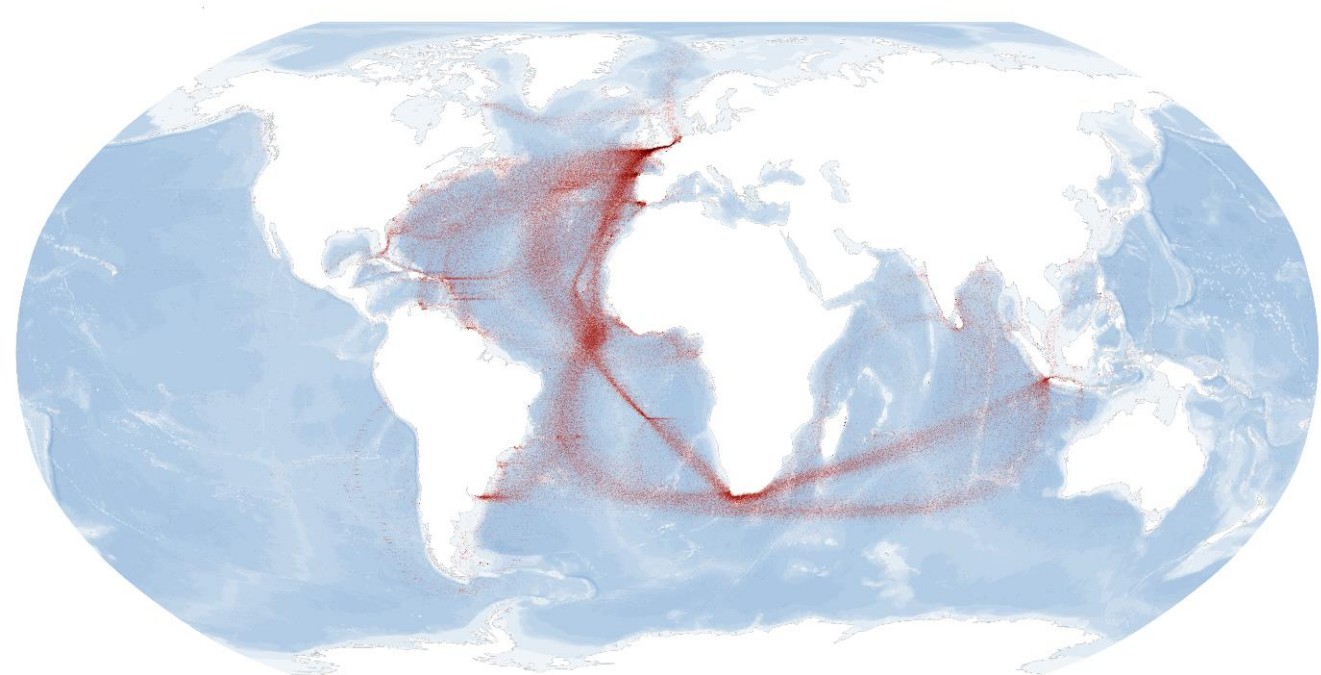

**Figure 10**: Plot of the position of all ships' logbook entries in the CLIWOC database (Degroot and Ottens, 2020). The map is derived from the open source variant of the CLIWOC database (García-Herrera et al., 2005b) held at https://www.historicalclimatology.com.

## 7.3. Indices of sea-ice extent

Records of sea ice in harbours and heavily trafficked waterways – or records of dues paid at ports and tollhouses – yield easily quantified data. However, reports of sea ice at high latitudes in correspondence, logbooks or journals written before the 19th century often give unclear descriptions of sea ice density, which makes it harder to determine how much sea ice there might have been in different regions from year to year (Prieto et al., 2004). The resolution and precision of Arctic index-based sea ice or iceberg reconstructions that rely on early modern documents is accordingly quite low (Catchpole and Faurer, 1983; Catchpole and Halpin, 1987; Catchpole and Hanuta, 1989). An emerging way to circumvent this issue is to

focus on particular regions where warm and cold ocean currents mixed, and that were sensitive to (a) changes in sea and air surface temperatures and (b) current strength, for example, around western Svalbard or the Yugorsky Strait (Degroot, 2015). Logbook reports of the presence of sea ice in these target areas can be quantified, indexed, and used to develop reconstructions that suggest broadscale shifts in the strength of marine currents (Degroot, 2020).

**7.4. Indices of precipitation and storms**

Some ships' logbooks note the occurrence of precipitation at sea, and most record winds that must have influenced precipitation on land. Historical climatologists have therefore used logbooks to classify and graph precipitation at or near the sea (e.g. Wheeler, 2005b; Hannaford et al., 2015). Moreover, most documents that directly describe weather at sea or blown in from the sea faithfully report storms and at least approximately note their severity (Lamb, 1992; García-Herrera et al., 2004; García-Herrera et al., 2005a; Chenoweth and Divine, 2008; Wheeler et al., 2010). Reconstructions based on written evidence of damage inflicted along the coast, however, can be more problematic, as damage reflected both complex social conditions and environmental circumstances beyond the severity of storms (de Kraker, 2011; Degroot, 2018).

## 8. Methods for the derivation of climate indices

The preceding sections have highlighted the variable number of classes used in index-based climate reconstructions and hinted at the variety of different approaches to index development. This section summarises the main methodological approaches used to derive indices on the different continents, with an emphasis on temperature and rainfall series.

**8.1. Climate index development in Europe – "Pfister indices"**

In Europe, the most widely adopted approach to the reconstruction of temperature and rainfall variability for climatically-homogenous regions is through the development of seven-point ordinal indices (Pfister, 1984; Pfister et al., 2018), which the climate historian Franz Mauelshagen has termed "Pfister indices" (Mauelshagen, 2010). These indices are normally generated at a monthly level through the analysis of (bio)physically-based proxies and contemporary reports of climate and related conditions. This is not without its challenges, and requires a source-critical understanding of the evidence-base in addition to a knowledge of regional climates (Brázdil et al., 2010). To aid interpretation, any contemporary report should be accompanied by a range of information, including details of the date, time, location affected, author and source quality (see Brázdil et al., 2010; Pfister et al., 2018). The criteria used to allocate a specific month to a specific index category will vary from place to place according to regional climatic variability. Table 1, for example, illustrates the indicators used to classify individual months as either "warm" (+2/+3) or "cold" (-2/-3) in a temperature reconstruction for Switzerland (Pfister, 1992); these include regionally relevant phenomena such as the timing and duration of snowfall and various plant-phenological indicators. Pfister et al. (2018) recommend that monthly rankings of above +1 and below -1 should only be attributed based on proxy data such as phenological evidence, with values of -3 and +3 reserved only for exceptional months. An index value of 0 should only be used where reports of climate suggest normal conditions – an absence of data should be reported as a gap in the time series rather than a 0 value.

**Table 1**: Criteria used in the generation of seven-point temperature indices for "warm" (+2/+3) or "cold" (-2/-3) months in Switzerland (after Pfister, 1992; Pfister et al., 2018). *Italics* indicate criteria grounded in statistical analyses.

| Month | "Cold" (index values of -2/-3) | "Warm" (index values of +2/+3) |
|---|---|---|
| Dec, Jan, Feb | Uninterrupted snow cover<br>*Freezing of lakes* | Scarce snow cover<br>Early vegetation activity |
| Mar | Long duration of snow cover<br>Frequent snowfalls | Early sweet cherry flowering<br>No snowfall |
| Apr | Several days of snow cover<br>Frequent snowfalls | *Beech tree leaf emergence*<br>*Early vine flower* |
| May | *Late grain and grape harvest*<br>*Late vine flower* | *Early grain and grape harvest*<br>*Start of barley harvest* |
| Jun | *Late vine flower*<br>Several low altitude snowfalls | *Early grain and grape harvest*<br>*High vine yields* |
| Jul | *Low vine yields*<br>Snowfalls at higher altitudes | *High vine yields* |
| Aug | *Low tree ring density*<br>Low sugar content of vine<br>Snowfalls at higher altitudes | *High tree ring density*<br>High sugar content of vine |
| Sep | Low sugar content of vine<br>Snowfalls at higher altitudes | High sugar content of vine |
| Oct | Snowfalls, snow cover | Second flowering of spring plants |
| Nov | Long duration of snow cover | Second flowering of spring plants<br>No snowfall |

Once monthly index values have been generated, these are then summed to produce seasonal or annual classifications where required. Three-month seasonal values can, as a result, fluctuate from -9 to +9 and annual values from -36 to +36 (see Pfister, 1984). It should be remembered, however, that indexation generates ordinal data, with no guarantee that the intervals between each index level are equal, so that the sum for a specific season or year can only approximate the magnitude of a meteorological phenomenon. The process of summation may result in positive index values for relatively warmer/wetter months during the year being cancelled out by negative index values for relatively colder/drier months. For example, a year containing a run of extremely dry months followed by a run of extremely wet months may produce a summed index value close to zero – even though the year includes two periods of 'extreme' climate. Careful assessment is therefore required when reporting summed indices to avoid any loss of information, particularly concerning extreme events. The approach used by Nicholson et al. (2012a) for African precipitation series may be helpful here, where individual years were flagged if documentary sources suggested wetter and drier extremes across the year that differed by more than two index classes.

Implicit in this methodological approach is that runs of monthly indices are available with almost no gaps (e.g. Litzenburger, 2015) or that, where gaps occur, there is a high probability that conditions during a given month reflect the longer-term average for that month (e.g. Dobrovolný et al., 2009). Variations in source density, however, mean that it may not always be possible to define indices at a monthly level. Such variations could simply be due to a scarcity of available sources, or could be the product of seasonal variability that results in observations of a climate phenomenon being concentrated in specific parts of the year (e.g. observations of rainfall in areas of Europe with a Mediterranean climate are likely to be concentrated between September and April). In these situations, researchers should (i) choose an appropriate temporal resolution (i.e. seasonal or annual) based on the number and quality of available records, and (ii) develop specific seasonal- or annual-level criteria – see, for example, the temperature and precipitation reconstructions for Belgium, Luxembourg and The Netherlands generated by Camenisch (2015a) or the Mediterranean temperature series by Camuffo et al. (2010). The methods used for calibration and verification are outlined in the following section.

In the development of his seven-point scale, Pfister assumed that monthly temperature and precipitation followed a Gaussian distribution. Initially, Pfister (1984) developed duodecile classes based on the frequency distribution of monthly temperature/precipitation means for the sixty-year reference period 1901-1960 as the standard of comparison (Table 2). The

723 most extreme months (i.e. those given an index value of -3/+3) were those that fell into duodecile classes 1 and 12,

representing the 8.3% driest (or coldest) or 8.3% wettest (or warmest) months, respectively. Other index categories were

defined using 16.6% intervals. In the later version of his indices, Pfister (1999 and onwards) discontinued the use of

duodecile classes, using instead the standard deviation from the mean temperature/precipitation for the 1901-1960 reference

period to define index categories: -/+180% (of the standard deviation from the mean of the reference period) for index values

728 -3/+3, -/+130% for values -2/+2, and +/-65% for values +1/-1.

**Table 2**: The definition of the weighted temperature and precipitation index values used in the creation of initial (pre-1999)
seven-point "Pfister" indices (after Pfister, 1992).

|  | Lowest |  |  |  |  |  | Highest |
|---|---|---|---|---|---|---|---|
|  | 8.3% | 16.6% | 16.6% | 16.6% | 16.6% | 16.6% | 8.3% |
| Duodecile | 1 | 2-3 | 4-5 | 6-7 | 8-9 | 10-11 | 12 |
| Index | -3 | -2 | -1 | 0 | 1 | 2 | 3 |

## 8.2. Climate index development in Asia

In China, the quantification of historical records to reconstruct climate change originated with a Semantic Differential

Method based on an analysis of each record's content (see Central Meteorological Bureau of China, 1981; Su et al., 2014;

Yin et al., 2015). Temperature series were traditionally established at a decadal scale only. In creating a series, each year was

first defined as 'cold', 'warm' or 'normal' according to direct weather descriptions or environmental and phenological

evidence. In contrast to the Pfister method (see section 8.1), 'normal' was also used when there was insufficient information

available to determine temperature abnormalities. This approach reflects the nature of most Chinese documents, where the

primary mission of the recorders was to detail abnormal or extreme events; fewer descriptions of abnormal events are

therefore interpreted as indicating conditions closer to normal. After each year had been defined as cold, warm or normal, an

equation was then used to derive the decadal indices. The earliest example was published by Zhang (1980): $T_i = -[n_1 +$

$0.3(10 - \overline{(n_1 + n_2)})]$, where $T_i$ is the decadal winter temperature index, $n_1$ the number of cold years, $n_2$ the number of warm

743 years, and 0.3 the empirical coefficient (see also Zhang and Crowley, 1989). The resulting value is always negative; the

744 lower the value, the more severe the coldness.

A second approach to the construction of ordinal scale indices was developed by the Wangs in the 1990s (e.g. Wang and

Wang, 1990a; Wang and Wang, 1990b; Wang et al., 1998). This used a four-point scale (0, 1, 2, 3) (Table 3). As in Europe,

indices were generated through the analysis of phenological descriptions and contemporary reports of climate and related

phenomena. Like Europe, criteria for individual index categories could also be adjusted for specific places at specific

seasons according to geographical and climatic attributes. However, unlike the Pfister method, an index value of 0 could be

used where there were missing data. The Wangs further introduced a statistical method to compare phenological evidence

with modern (1951-1985 CE) and early instrumental data (1873-1972 CE in Shanghai) and allocate temperature ranges to

ordinal scales (Wang and Wang, 1990b). An index value of -0.5 corresponded to a -0.5~-0.9°C temperature anomaly, a value

of -1.0 to a -1.0~-1.9°C anomaly and a value of -2.0 to an anomaly of <=-2.0°C; values of 1.5 were added to indicate warm

temperatures and -3.0 to capture extreme cold periods. These cold indices were then regressed with the decadal mean

temperature (1873-1972 CE) to derive a coefficient through which the index value could be transferred into a 'real'

temperature.

**Table 3**: Criteria used in the development of temperature indices in China.

| Cold index values | | | | Temperature index values | |
|---|---|---|---|---|---|
| Wang, R. and Wang, S. (1990) | | Wang, S. and Wang, R. (1990) | | Tan and Wu (2013), adapted from Chen and Shi (2002) | |
| Index value | Criteria (winter) | Index value | Criteria (distinguishing four seasons; example of winter) | Index value | Criteria (winter and summer; example of winter) |
| 0 | No record of ice/frost; no snow; light snow | 1.5 | Warm records | 1 | Warm records such as 'winter warm as spring' |
| 1 | River/lake freezing; heavy snow over several days or several cm depth | -0.5 | Heavy snow; freezing rain; ice glaze on trees | 0 | No specific records |
| 2 | River/lake frozen for weeks to allow human passage; heavy snow for months; snow frozen for months | -1.0 | Frozen river or lake | -1 | Heavy snow; freezing rain; ice glaze on trees |
| 3 | River/lake frozen for months to allow horse-drawn wagons or carriages to cross; heavy snow for months; ice melt in following spring | -2.0 | Extreme cold; ocean water and large lakes or rivers frozen | -2 | River/lake frozen for months to allow horse-drawn wagons or carriages to cross |
| | | -3.0 | River/lake frozen for months to allow horse-drawn wagons or carriages to cross | | |

Chen and Shi (2002) built upon Zhang (1980) and the Wangs' approaches in developing an equation to calculate decadal temperature indices: $T_i = 10 - 2n_1 - n_2 + n_3$, where $n_1$= number of extremely cold years, $n_2$= number of cold years, $n_3$= number of warm years. A resulting decadal temperature index value of 10 denotes average conditions; <10 anomalous cold; and >10 anomalous warm. Successive work (Tan and Liao, 2012; Tan and Wu, 2013) adopted the Chen and Shi (2002) approach with a slight modification of the index criteria while retaining the four-point ordinal scale. The temperature series generated using this approach have been incorporated into multi-proxy temperature reconstructions (e.g. Yi et al., 2012; Ge et al., 2013). Zheng et al. (2007) and Ge et al. (2013) provide useful reviews of the approach used to generate temperature indices in China.

As noted in section 3.2, drought-flood index reconstruction in China has a long tradition. Two main approaches are used. Earlier studies adopted a proportionality index approach (Zhu, 1926; Yao, 1943). As explained by Gong and Hameed (1991), Zhu used the equation $I = D/F$ to calculate the index, where $D$ represents the number of droughts and $F$ the number of floods in a given time period. This equation is poorly defined if $F$ or $D$ is zero. Brooks (1949) modified the equation and used the flood percentage, $I = 100 \times F/(F + D)$, to derive moisture conditions in Britain and some European regions from 100 BCE onwards at a 50-year resolution. Gong and Hameed (1991) further modified the equation as $I = 2F(F + D)$ to derive indices at a 5-year resolution. Their index takes the values $0 \leq I \leq 2$, with larger values reflecting wetter conditions. Zhang and Zhang (1979) adopted a slightly different approach by counting the number of places with reported drought events: $I_D = 2D/N$, where $D$ represents the number of places having extreme drought (grade 5) and drought (grade 4) events in a given year (see Table 4), and $N$ is the total number of places.

The Academy of Meteorological Science of China Central Meteorological Administration (1981) adopted a five-point ordinal scale approach to reconstruct annually resolved drought-flood indices in China. The key descriptors for each classification (see Table 4) are mainly based on accounts of the onset, duration, areal extent and severity of each drought or flood event in each location. They then assume a probability distribution of the five grades following a normal distribution: 1 (10%), 2 (25%), 3 (30%), 4 (25%), and 5 (10%). For the period of overlap between written and instrumental records (after

1950 CE), the graded series were compared against the observed May-September (major rainy season) precipitation and

regressed to transform the indices into numerical series (Table 4). Based on the five-point ordinal scale, Wang et al. (1993)

and Zheng et al. (2006) developed further formulae to calculate decadal drought-flood indices that can be applied to earlier

periods (i.e. before 1470) when less information is available.

**Table 4**: Criteria used in the generation of five-point drought-flood indices in China (Academy of Meteorological Science of

China Central Meteorological Administration, 1981). For more details, see Zhang and Crowley (1989), Zhang et al. (1997),

and Yi et al. (2012).

| Index value | Norm | Transfer function for precipitation amount |
|---|---|---|
| 1 (Very wet) | Prolonged heavy rain, continuous flood over two seasons, extensive flood, unusually heavy typhoon rain | $R_i > (\bar{R} + 1.17\sigma)$, where, $\bar{R}$ is mean May-Sep precipitation, $\sigma$ is standard deviation, $R_i$ is precipitation in the $i^{\text{th}}$ year |
| 2 (Wet) | Spring or autumn prolonged rain with moderate damage, local flood | $(\bar{R} + 0.33\sigma) < R_i \leq (\bar{R} + 1.17\sigma)$ |
| 3 (Normal) | Favourable weather, usual case, or nothing special to be noted in records | $(\bar{R} - 0.33\sigma) < R_i \leq (\bar{R} + 0.33\sigma)$ |
| 4 (Dry) | Minor impacts of drought in a single season, local minor drought disaster | $(\bar{R} - 1.17\sigma) < R_i \leq (\bar{R} - 1.33\sigma)$ |
| 5 (Very dry) | Severe drought over a season, drought continued for several months, severe drought over an extensive area, or records describing extensive areas of barren land | $R_i \leq (\bar{R} - 1.17\sigma)$ |

## 8.3. Climate index development in Africa

Historical climate reconstructions for Africa use two different approaches to index development. The continent-wide rainfall

reconstruction by Nicholson et al. (2012a) is based upon 90 regions that are homogeneous with respect to interannual rainfall

variability. An underpinning assumption is that historical information for any location within a region – be it narrative or

instrumental – can be used to produce a precipitation time series representing that region. Instrumental rainfall data are

converted into seven "wetness" classes (-3 to +3) based on standard deviations from the long-term mean. A wetness index

value of zero corresponds to annual rainfall totals within +/-0.25 standard deviations of the mean. Index values of −1/+1 are

assigned to annual values between −0.25/+0.25 and −0.75/+0.75 standard deviations. Values of −2/+2 are given to annual

totals between −0.75/+0.75 and −1.25/+1.25 standard deviations, with more extreme departures classed as −/+3.

Documentary data are integrated by first assigning individual pieces of narrative evidence to a specific region; each piece of

evidence is then classified into one of the seven "wetness" categories. Like the approach used by Pfister, the presence of key

descriptors of climate conditions is used to distinguish these categories. The scores for each item of evidence for a specific

region/year are summed and averaged. Where there are several sources, a '0 index' value represents an average of

conditions. Where only single sources are available, some contain so much climate-related information that, as in China,

absence of evidence for a specific season is taken to infer "normal" conditions; such cases are indicated in the original data

file accompanying the Nicholson et al. (2012a) reconstruction.  Algorithms are then used to weight and combine

documentary and instrumental data for each region and year. These are defined subjectively according to the accuracy of the

quantitative versus qualitative indicators. For example, when one of each type is available, the qualitative indicator is

weighted twice as much as the gauge because of the inherent spatial variability within African rainfall. A second assumption

is that when the correlation between rainfall in two regions is >0.5 the regions are appropriate substitutes for each other

(Nicholson, 2001). In this way, classifications for regions without evidence for a given year can be derived by substitution.

Statistical inference (termed 'spatial reconstruction' by Nicholson) is then used to generate classifications for any remaining

regions. The cutoff of 0.5 was selected based on examination of time series that correlate with each other at various levels.
Those with a correlation of 0.5 showed marked similarity, though it should be noted that, in most cases, the correlation was
much higher, with the statistical significance being >0.001.

Regional rainfall reconstructions in southern Africa use an approach much closer to the Pfister method to classify
documentary evidence into one of five rainfall classes (-2 to +2); these classes are ordinal rather than based on statistical
distributions. Like the Pfister method, a '0 index' value is only awarded where narrative evidence suggests normal
conditions – years with inconclusive or no data are left unclassified. Owing to the relatively paucity of documentary data for
Africa compared to Europe, conditions for specific rainy seasons are categorised at a quarterly (e.g. Nash et al., 2016) or
more commonly annual level. Again, key descriptors are used to distinguish the various index classes. The main point of
divergence with the approach used by Nicholson is that – rather than assigning individual pieces of evidence to wetness
classes and averaging – qualitative analysis is undertaken of all quotations describing weather and related conditions for an
entire quarter/year (see Nash, 2017). These different methodological approaches, as well as the type of documentary
evidence used, can introduce discrepancies between rainfall series for overlapping regions. Hannaford and Nash (2016) and
Nash et al. (2018) note, for example, that the reconstructions in Nicholson et al. (2012a) for KwaZulu-Natal during the first
decade of the 19th century and Malawi for the 1880s-1890s show generally drier conditions than overlapping series
generated using different methods.

**8.4. Climate index development in the Americas**

Temperature, precipitation and phenological indices for North America have been based on a distinctive content analysis
approach. This method was first applied to historical climatology in the 1970s to reconstruct freeze and break-up dates
around Hudson Bay for the period 1714-1871 CE by quantifying the frequency and co-occurrence of key weather descriptors
in Hudson's Bay Company records (Catchpole et al., 1970; Moodie and Catchpole, 1975). The resulting indices are open-
ended, since more and stronger descriptors in the sources could generate indefinitely larger (positive or negative) values.
Baron (1980) adapted content analysis to analyse historical New England diaries, by ranking and then numerically weighting
descriptors of several types of weather found in those sources. In subsequent publications, he and collaborators adopted
different scales for annual and seasonal temperature and precipitation depending on the level of detail in the underlying
sources (e.g. Baron, 1995).

In Mexico, Mendoza et al. (2007) constructed a series of historical droughts for the Yucatan Peninsula using the method of
Holmes and Lipo (2003). In this investigation, historical drought data were transformed into a series of pulse width
modulation types (1 drought, 0 no drought) and linked to the Atlantic Multidecadal Oscillation and Southern Oscillation
Index. Other studies have used key descriptors as the basis for index development. Garza Merodio (2017), for example,
classified rogation ceremonies into five ordinal levels based on Garza and Barriendos (1998), creating drought series for
México, Puebla, Morelia, Guadalajara, Oaxaca, Durango, Sonora, Chiapas and Yucatán. Dominguez-Castro et al. (2019)
generated binary series (presence or absence) for precipitation, frost, hail, fog, thunderstorm and wind in Mexico City.
Temperature indices for Mexico have been developed using the applied content analysis approach of Baron (1982) and
Prieto et al. (2005).

In South America, the methodology used to analyse historical sources for climate reconstruction initially followed Moodie
and Catchpole's (1975) content analysis approach, but was later adapted in a number of papers by María del Rosario Prieto
(e.g. Prieto et al., 2005). As noted in section 5, most historical rainfall and river flow index series use 5–7 annually- or
seasonally-resolved classes based on key descriptors, while most temperature-related series use 3 classes. To date, all South
American rainfall and temperature series are ordinal in nature and do not make background assumptions about the statistical
distribution of climate-related phenomena. However, the method used to derive '0 index' values is not always clearly stated,

and many series do not discriminate between 'no data' and 'normal' years (both of which are expressed as zero values). For example, in many studies that use rogation ceremonies as the basis for rainfall index development, months when there are no ceremonies are categorised as zero. There are exceptions, e.g. Dominguez-Castro et al. (2018), who explicitly identify an absence of ceremonies as 'no data', and Prieto and Rojas (2015), who clearly differentiate between normal years and no data. A systematic reanalysis of many series would be useful to determine exactly how each was constructed.

The approach used by Quinn et al. (1987) and Quinn and Neal (1992) to construct El Niño series over the past four and a half centuries is slightly different. The relative strengths of individual El Niño events were based on a range of subjective and objective measures in documentary sources from coastal Peru. These include descriptions of relative rainfall, the extent of flooding and the degree of physical damage and destruction associated with each event, alongside accounts of impacts on shipping (e.g. wind and current effects on travel times between ports), fisheries (e.g. changes to fish catches, changes in fish meal production), and marine life (e.g. mass mortality of endemic marine organisms and guano birds, extent of invasion by tropical nekton) (Quinn et al., 1987). This broad approach was continued in subsequent studies by Ortlieb (1994, 1995, 2000), García-Herrera et al. (2008) and others.

**8.5. Climate index development in Australia**

Australian efforts have largely been based on the Pfister approach (section 8.1) and regional-scale historical climatology investigations in southern Africa (section 8.3), although instrumental and documentary sources have been analysed separately. Fenby and Gergis (2013) and Gergis and Ashcroft (2013) converted documentary and instrumental data into a three-point scale of wet, normal and drought conditions. Historical data availability along with high spatial variability and known non-linearities in Australian rainfall meant that wet and dry conditions were assessed differently. Years were classified as 'normal' if they failed to reach either wet or dry criteria. To avoid introducing errors or biases in the record, years with missing data were marked as missing, rather than given a value of zero.

For droughts, agreement between a minimum of three of the twelve documentary sources used was required for drought conditions to be identified in a given month. Droughts were identified regionally in one of five modern southeastern Australian states. To avoid issues associated with exaggerated accounts of dry conditions and/or localised drought, a year was classified as a 'drought year' only when at least 40% of historical sources indicated dry conditions for at least six consecutive months during the May-April 'ENSO' year (the period with strongest association between south-eastern Australian rainfall variations and ENSO; Fenby and Gergis, 2013). Dry conditions were defined as times where a lack of rainfall was perceived as severe by society, or negatively impacted upon agriculture or water availability.

Months of above average rainfall in coastal New South Wales were identified using the annual rainfall summaries of Russell (1877), as this was the only source with consistent yearly information about rainfall events and impacts. Along with specific reports of good rainfall, monthly classifications of wet conditions were also based on accounts of flooding, abundant crops, excellent pasture and the occurrence of insect plagues (Fenby and Gergis, 2013). Six months of high rainfall were required for a year (May-April) to be defined as wet.

Combining the documentary-based indices with an instrumentally-derived index enabled the development of a single index of wet and dry conditions for eastern New South Wales from 1788 to 2008 CE. Each year of the instrumental rainfall datasets – the nine-station network for the Sydney region (1832-1860 CE) and a larger 45-station network representing the wider south-eastern Australian region – was assigned an index value of wet (1), normal (0) or dry (–1) based on normalised precipitation anomalies. Years with a normalised precipitation anomaly greater than the 70th percentile were counted as wet for that station, while those with an anomaly below the 30th percentile were counted as dry. Overall, a year was classified as wet or dry for the region if at least 40% of the stations with data available were in agreement, in line with the documentary classification of Fenby and Gergis (2013). Similar methods were employed by Ashcroft et al. (2014a) who used half a

894 standard deviation above or below the 1835–1859 CE mean to build three-point indices of temperature, rainfall and pressure

variability in southeastern Australia before 1860 CE using early instrumental data.

## 8.6. Climate index development in the oceans

The most common indices for marine climate reconstruction quantify shifts in prevailing wind direction. Most convert

directional measurements from the 32-point system used by mariners in logbooks to one, four- or (very recently) eight-point

indices – in part, because sailors were biased towards four, eight, and 16-point compass readings (Wheeler, 2004, 2005a).

These "directional indices" resemble the ordinal scales used to quantify qualitative temperature and rainfall observations on

land. Few calculate error or confidence in their reconstructions, in part because those considerations are difficult to quantify

(see García-Herrera et al., 2018). Recent studies have quantified the uncertainty involved in connecting logbook observations

to broadscale circulation changes by using a calibration process that correlates wind directions in a target area traversed by

ships to, for example, the strength of a monsoon (Gallego et al., 2015; Gallego et al., 2017).

Wind velocity and storm intensity or frequency indices have also made use of observations recorded in logbooks. Beginning

in the 19th century, mariners made these measurements using the 12-point Beaufort wind force scale. Before that,

measurements refer to the sails that mariners needed to furl or unfurl in winds of different velocity. The measurements are

therefore more subjective than those of wind direction, yet they can still be roughly translated into Beaufort indices (see

García-Herrera et al., 2003; Koek and Konnen, 2005). It is therefore possible to use these indices to develop high-resolution

reconstructions of trends in average wind velocities and storm frequency and intensity (Degroot, 2014). Yet because shifts in

wind direction were more objectively recorded by sailors than changes in wind velocity, and are equally indicative of

broadscale circulation changes, directional reconstructions are generally favoured by historical climatologists (Ordóñez et

al., 2016).

## 9. Calibration, verification and dealing with uncertainty

### 9.1. Calibration and verification in index development

There are several approaches for calibrating and verifying index series used globally. Where overlapping meteorological data

are available, long series of temperature and precipitation indices can be converted into quantitative meteorological units by

using statistical climate reconstruction procedures; some of these have been inherited from fields such as dendroclimatology

(see Brázdil et al., 2010, for a full discussion of statistical methods). For regions of the world lacking long instrumental

records, simple cross-checking of climate indices against shorter periods of overlapping data is often used.

In Europe, Pfister (1984) was the first to use a calibration and verification process in the development of his indices. His

approach – an example of best practice for regions where there is a lengthy period of instrumental overlap with the

documentary record – is summarised by Brázdil et al. (2010) and Dobrovolný (2018) and illustrated in Figure 11. However,

even where a period of overlap is lacking, indices from documentary sources can still be used to cross-check reconstructions

from proxy data (e.g. Bauch et al., 2020) or modelling results and observations (e.g. Bothe et al., 2019). The aim of

calibration is to develop a transfer function between an index series and the measured climate variable, with verification

against an independent period or subset of the overlapping meteorological data used to check the validity of this transfer

function. In studies where there is a multi-decadal period of overlap, the instrumental data are normally divided into two

subperiods; the index series is first calibrated to the earlier subperiod and then verified against the later subperiod

(Dobrovolný, 2018). If only a short period of overlap is available, then cross-validation procedures are required.

The transfer function derived from a calibration period is normally evaluated by statistical measures (e.g. squared correlation

$r^2$, standard error of the estimate) before being applied in the verification period. During verification, index values calibrated

to physical units (e.g. temperature degrees or precipitation amount) are compared with the instrumental data and, again,

evaluated statistically using $r^2$, reduction of error and the coefficient of efficiency (see Cook et al., 1994; Wilson et al.,

2006). If the calibrated data series, derived by applying the transfer function obtained for the calibration period, expresses

the variability of the climate factor under consideration with satisfactory accuracy in the verification period, then the index

series can be considered as useful for climate reconstruction back beyond the instrumental period (Brázdil et al., 2010).

Caution is needed, however, as transfer functions, which are usually derived from relatively modern periods, may not be

stable through time (e.g. where phenological series have been influenced by the introduction of new varieties or different

harvesting technologies; Pfister, 1984; Meier et al., 2007).

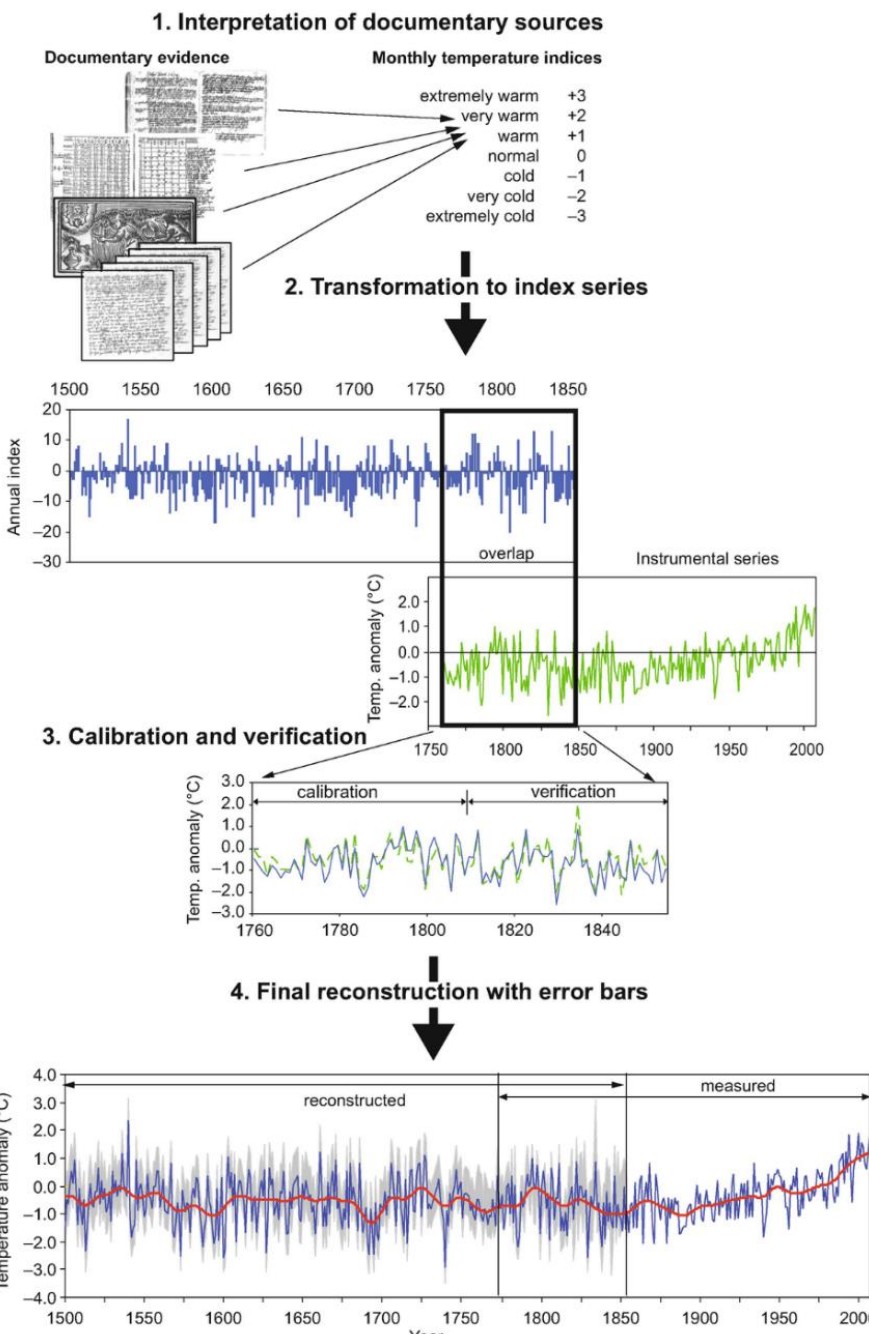

**Figure 11**: The main steps in quantitative climate reconstruction based on temperature or precipitation indices derived from

documentary evidence. Historical documentary sources are analysed to generate seven-point monthly indices (step 1), which

are then summed to produce annual index series (step 2). Calibration and verification are carried out on periods of

overlapping instrumental data (step 3), with statistical results from verification used to define error bars for the final

reconstruction (step 4). Reprinted by permission from: Brázdil, R., Dobrovolný, P., Luterbacher, J., Moberg, A., Pfister, C.,

Wheeler, D., and Zorita, E.: European climate of the past 500 years: new challenges for historical climatology, Climatic

Change, 101, 7-40 (© Springer 2010).

Like the European approach, calibration and verification methods in China are applied to reconstructed temperature and

drought-flood indices by comparing the series overlap between instrumental and documentary periods. Shanghai has the

longest instrumental data coverage (1873 CE onwards), with Beijing, Suzhou, Nanjing, and Hangzhou also having century-

long data series (Chen and Shi, 2002; Zhang and Liu, 2002). As a result, most calibration is performed with reference to

these cities. Wang and Wang (1990a) compared their temperature series with these instrumental data to estimate correlation

coefficients and allocate corresponding values to their indices. A transfer function was also estimated between the number of

snow days (or number of lake freezing days) and observed temperatures by using multiple regression methods (Zhang, 1980;

Gong et al., 1983; Zhang and Liu, 1987; Wang and Gong, 2000; Ge et al., 2003). However, the statistical correlation reports

in these earlier studies appear incomplete.

The Academy of Meteorological Science of China Central Meteorological Administration (1981) have used precipitation

data (1951-2000 CE) to validate drought-flood indices. However, the approach used focused on determining the probability

distribution function of their five index classes to make the series comparable with instrumental data, rather than calibration

*per se* (Yi et al., 2012; Shi et al., 2017). A special feature of calibration and verification in China is the utilisation of records

in the *Qing Yu Lu* and *Yu Xue Fen Cun* (Hao et al., 2018; see section 3.2), where comparisons can be made between

reconstructed drought-flood indices and observed precipitation patterns (Zhang and Wang, 1990). Such correlations can

further be compared and calibrated using instrumental data, for example for Beijing (Zhang and Liu, 2002), Suzhou, Nanjing

and Hangzhou (Zhang and Wang, 1990).

Validation within the Nicholson et al. (2012a) rainfall reconstruction for continental Africa was carried out by comparing

time series based on those entries with instrumental rainfall data available for the same time and region. Quality control in

the final seven-class combined instrumental-historical reconstruction was provided by comparing the spread of estimates

from the various sources. If more than a two-class spread existed among the entries for an individual region and year, each of

those entries was re-evaluated. In most, it was found that an error was made in determining the location or year of a piece of

documentary evidence. Only eight "conflicts" in the Nicholson series could not be resolved in this way. The various regional

studies in southern Africa employ a simpler approach, using short periods of overlap with available instrumental data for

qualitative cross-checking/validation purposes (e.g. Nash and Endfield, 2002; Kelso and Vogel, 2007; Nash and Grab, 2010;

Nash et al., 2016).

The content analysis method developed for North American historical climatology uses replication by other researchers to

test the reliability of the quantification process and compared results from multiple independent sources to test validity

(Baron, 1980, pp.150-170). Subsequent studies have elaborated on this method, but many also draw on the Pfister index

approach as summarised in section 8.1. For South America, Neukom et al. (2009) created "pseudo-documentary" series to

quantify the relationship between document-derived  precipitation indices and instrumental data (see also Mann and

Rutherford, 2002; Pauling et al., 2003; Xoplaki et al., 2005; Küttel et al., 2007). Following European conventions, index

series were transformed to instrumental units by linear regression with overlapping instrumental data. The skill measures

were quantified based on two calibration/verification intervals, using the first and second half of the overlap periods as

calibration and verification period, respectively and vice versa (Neukom et al., 2009). A similar approach has been used in

southern Africa to integrate documentary-derived index series with other annually-resolved proxy data for the 19th century

as part of multiproxy rainfall reconstructions (Neukom et al., 2014a; Nash et al., 2016).

Calibration and verification of indices in Australia (Figure 12) has been conducted using overlapping and largely

independent instrumental data products, similar to approaches used in African reconstructions. In an example of good

practice for future studies, independent high-resolution palaeoclimate reconstructions and records of water availability, such

as lake levels, were also used for verification (Gergis and Ashcroft, 2013). Disagreements between these different sources

were examined closely and often attributed to spatial variability in individual sources. For example, the 1820s in south-

eastern Australia were identified as wetter than average in a regional palaeoclimate reconstruction (Gergis et al., 2012), but

drier than average in a documentary-derived index and in historical information about water levels in Lake George, New

South Wales (Gergis and Ashcroft, 2013). This was put down to geographical differences between the datasets – the

palaeoclimate reconstruction was biased towards rainfall variability in southern parts of south-eastern Australia while the

lake records and documentary index represented the east.

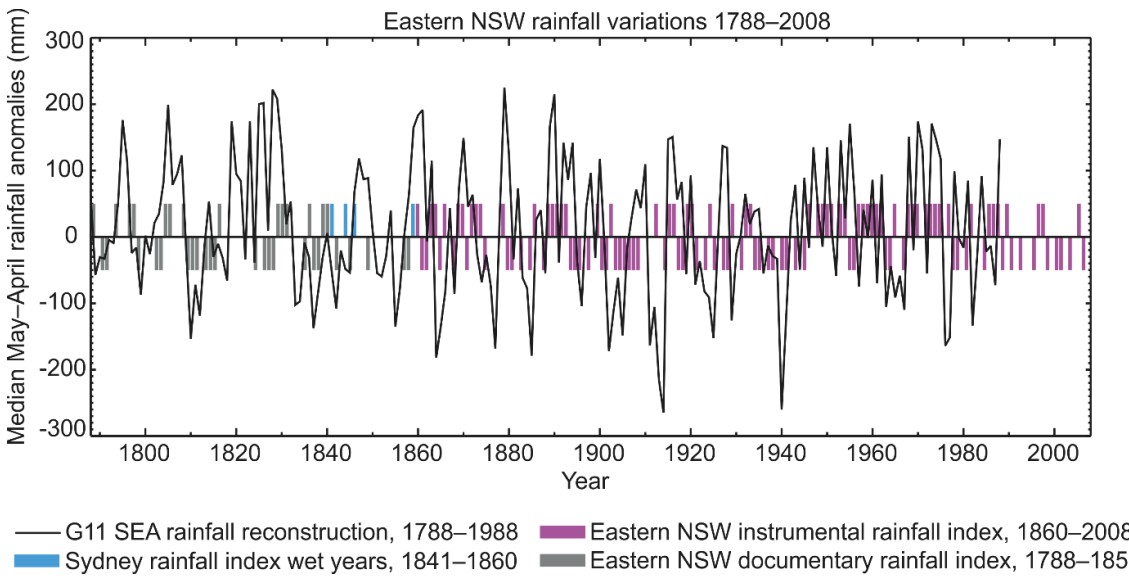

**Figure 12**: Wet and dry years for eastern New South Wales (Australia) identified using the nine-station network (1860–

2008, purple) and a documentary index (1788–1860, grey). The median rainfall reconstruction (1788–1988) from Gergis et

al. (2012) is also plotted as anomalies (mm) relative to a 1900–1988 base period. Note that 1841, 1844, 1846 and 1859 have

been classified as wet, in accordance with a rainfall index derived from observations in the Sydney region (blue). Adapted

from Gergis and Ashcroft (2013).

It is a long-standing best practice in marine historical climatology to verify weather observations by comparing different

kinds of documentary evidence, or alternative different examples of the same evidence (e.g. multiple logbooks in the same

fleet). Despite the very real challenges of interpreting measurements even in logbooks, there are indications that

reconstructions that use these sources are reliable. There appears to be a high consistency and homogeneity both within wind

measurements derived entirely from ships' logbooks, and between such measurements and data obtained from diverse

sources that register the marine climate. Researchers have therefore linked documentary weather observations in, for

example, the CLIWOC database, with datasets that homogenise and synthesise evidence from both textual and natural

proxies, such as the National Oceanic and Atmospheric Administration's International Comprehensive Ocean-Atmosphere

Data Set (ICOADS) (Jones and Salmon, 2005; Barriopedro et al., 2014).

**9.2. Reporting confidence and uncertainty in index-based climate series**

Two forms of uncertainty are encountered when developing index-based climate series: (i) uncertainties related to the

compilation of the index series themselves from documentary evidence; and (ii) uncertainties within any resulting index-

based climate reconstruction. The first form of uncertainty relates mainly to the nature of information contained within specific source types. A detailed discussion is beyond the scope of this review. However, where indices are compiled from a unique documentary source – such as a private diary or diaries (e.g. Brázdil et al., 2008; Adamson, 2015; Domínguez-Castro et al., 2015), a series of correspondence (e.g. Rodrigo et al., 1998; Nash and Endfield, 2002; Fernández-Fernández et al., 2014) or a series of acts of municipal and ecclesiastical institutions for a location (e.g. Barriendos, 1997; Dominguez-Castro et al., 2018) – it is easier to identify and correct unexpected bias or homogeneity problems. Other index series draw together information from many different documentary sources (e.g. Camuffo et al., 2010; Nash and Grab, 2010; Fenby and Gergis, 2013; Brázdil et al., 2016), allowing the analysis of longer periods or larger regions but at the risk of incorporating non-homogeneities. Methodological differences – for example in the way in which '0 index' values are derived (see section 8) – may also mask uncertainties introduced by data gaps.

While compiling this review, it became apparent that relatively few index-based climate series provide an assessment of the degree of uncertainty in the compilation of their indices – in effect, something akin to the error bars used in quantitative climate reconstructions (e.g. Dobrovolný et al., 2010). Further, very few studies report directly on potential biases in their series due to the well-known tendency for documentary evidence to better record extreme events. The incorporation of statistical error is achievable where index-based series have been subject to full calibration and verification (section 9.1). However, it is less straightforward for climate reconstructions in regions (or for time periods) where a lack of overlapping instrumental data renders full calibration impossible.

To overcome this issue, Australian studies include some assessment of confidence by showing details of the number of sources in agreement, and the proportion of the study regions affected (see Fenby and Gergis, 2013). Independent high-resolution palaeoclimate and historical records were also used to verify each year of the reconstruction to assess confidence in the results (Fenby and Gergis, 2013; Gergis and Ashcroft, 2013).

One innovation from African historical climatology is the introduction by Clare Kelso and Coleen Vogel (2007) of a qualitative three-point 'confidence rating' (CR) for the classification of each rainy season in their climate history of Namaqualand (South Africa). The rating for each season (Figure 13) was derived from the number of sources consulted combined with the number of references to that particular climatological condition. CR=1 was awarded where there was only one source referring to the climatic condition. In contrast, years awarded CR=3 were those that had more than three date- and place-specific references describing climatic conditions. This approach has been adopted in subsequent studies in southern Africa by Nash et al. (2016), Nash et al. (2018) and Grab and Zumthurm (2018), with slight variations in the criteria used to award specific ratings according to source density.

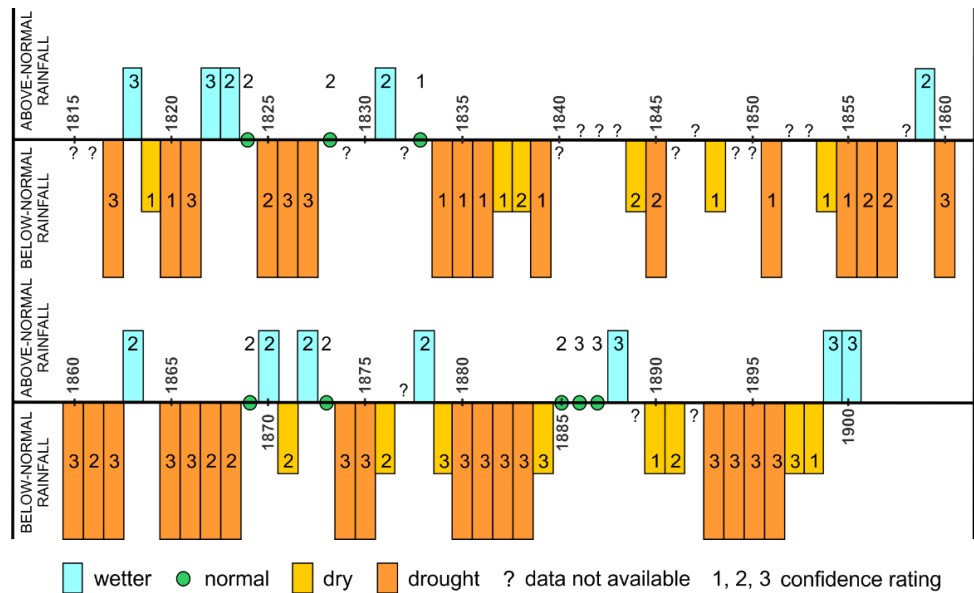

**Figure 13**: Five-point index of rainfall variability in Namaqualand (South Africa) during the 1800s, including the first use of confidence ratings in relation to annual classifications in a documentary-derived index series (1 = low confidence, 3 = high confidence). Data from Kelso and Vogel (2007).

A similar approach was adopted by Quinn et al. (1987) and Quinn and Neal (1992) in their development of El Niño indices for Peru. El Niño events with a confidence rating of 1 were those that lacked a source reference or informational basis; these were not incorporated into the final list of reconstructed events. CR=2 was awarded when an event was based on limited or circumstantial evidence; CR=3, when additional information was needed to confirm the time of occurrence or intensity of an event; CR=4, when the occurrence time and intensity information was generally satisfactory, but additional evidence was needed to confirm the spatial extent of the event; and CR=5, when the available information concerning the occurrence and intensity of the event was considered to be satisfactory.

The second form of uncertainty relates specifically to index-based climate reconstruction. Where uncertainties can be quantified (either formally with statistics or less formally by comparison with other reconstructions), index-based reconstructions can be made fully comparable to natural proxy-based quantitative reconstructions. One example of this approach is the central Europe temperature series by Dobrovolný et al. (2010), the only documentary series used as part of the PAGES 2k Consortium (2013) continent-by-continent temperature reconstruction. Calibrated temperature series from China, including Zhang (1980) and Wang and Wang (1990a), are also incorporated into the PAGES 2k Consortium (2017) community-sourced database of temperature-sensitive proxy records.

## 10. Towards best practice in the use of climate indices for historical climate reconstruction

### 10.1. Regional variations in the development and application of climate indices

This review has shown that there are multiple approaches globally to the development and application of indices for historical climate reconstruction. Returning to the themes identified in the introduction, three categories of variability can be recognised. First, there is variability in the types of climate phenomena reconstructed in different regions (Table 5). Studies of the historical climatology of Europe and Asia span the greatest range of climate phenomena. This is partly a product of the range of climate zones present in these continents, and therefore the diversity of weather phenomena to which observers might be exposed and document. However, it also reflects the relative abundance of documentary materials available for analysis and the richness of climate-related information they contain. Where smaller volumes of documentary evidence are

available, reconstructions naturally tend to be skewed towards the climate parameters that were of sufficient importance to people that they captured them in writing or as artefacts – hence the emphasis on precipitation reconstructions for Africa and Australia and on winds and storm events over the oceans.

**Table 5**: Types of historical environmental phenomena reconstructed using an index approach in different parts of the world, with a qualitative indication of the relative emphasis of studies in each region (3 indicates a large number of studies, 1 a small number of studies, - indicates no studies).

| Region | Temperature | Precipitation | Floods | Drought | Snow/ice | Wind/ storms |
|--------|-------------|---------------|--------|---------|----------|--------------|
| **Europe** | 3 | 3 | 3 | 2 | 1 | 1 |
| **Africa** | 1 | 2 | - | 1 | 1 | 1 |
| **Americas** | 1 | 1 | 1 | 1 | 1 | 1 |
| **Asia** | 2 | 2 | 2 | 1 | 1 | 1 |
| **Australia** | - | 1 | 1 | 1 | - | - |
| **Oceans** | - | 1 | - | - | 1 | 2 |

Second, there is variability in the way that historical evidence is treated to develop individual index series. Such variability arises, in part, from the extent to which analytical methods have developed independently. Thus, approaches to index-based climate reconstruction in parts of Asia are very different to those used in Europe. Chains of influence in practice can also be identified with, for example, elements of the 'Pfister method' from Europe being adopted by regional studies in southern Africa from the 1980s and then feeding into more recent precipitation reconstructions in Australia. There are common features of most historical treatments, regardless of tradition. These include the use of key descriptors or indicator criteria to match either individual observations (e.g. the continent-wide precipitation series for Africa developed by Nicholson) or sets of monthly, seasonal or annual observations (as per the Pfister method) to specific index classes. Most reconstructions are ordinal but, particularly where long runs of overlapping instrumental data are available, many are grounded in statistical distributions and present semi- or fully-quantified climate series.

The final source of variability across index-based investigations is in the number of index points used in individual reconstructions. A snapshot of this variability can be seen from investigations in Europe (Table 6). While most index-based reconstructions of European temperature and precipitation published since the 1990s employ the seven-point Pfister approach, some use up to nine classes. The number of classes used in European flood and drought reconstruction is usually smaller but, even here, may extend to seven-point classifications. There are also some commonalities. For example, most temperature and precipitation reconstructions use an odd number of classes – to allow the mid-point of the reconstruction to reflect 'normal' conditions – while open-ended unidirectional climate-related phenomena such as droughts and floods may be classified using either an even or odd number of classes. Similar patterns can be seen in other parts of the world (Table 7). In the rare instances where authors justify the number of index categories they use, most point to limitations in the quantity and/or richness of the historical evidence available for reconstruction as the reason for a smaller number of index categories.

**Table 6**: Variability in the number of index classes used in index-based historical climate reconstructions across Europe.

| Climate phenomenon | Number of index classes used in climate reconstructions | Examples |
|--------------------|---------------------------------------------------------|----------|
| **Temperature** | 7-point most common (but also 2-, 3-, 5- and 9- point) | e.g. Pfister (1984), Alexandre (1987), Brázdil and Kotyza (1995, 2000), Van Engelen et al. (2001), Glaser (2013), Litzenburger (2015) |

| Precipitation | 7-point most common (but also 3- and 5-point) | e.g. Alexandre (1987), Pfister (1992), Glaser et al. (1999), Van Engelen et al. (2001), Rodrigo and Barriendos (2008) |
| Floods | 3-, 4- 5-point all common | e.g. Pfister (1999), Rohr (2006, 2013), Wetter et al. (2011), Brázdil et al. (2012), Garnier (2015), Kiss (2019) |
| Drought | 3-point most common (but also 5- and 7-point) | e.g. Pfister et al. (2006), Brázdil et al. (2013b), Garnier (2018), Erfurt and Glaser (2019) |

**Table 7**: Variability in the number of index classes used in index-based historical climate reconstructions in Africa, the Americas, Asia, Australia and over the oceans.

| Region | Number of index classes used in climate reconstructions | Examples |
|---|---|---|
| **Africa** | 3-point for temperature; 5- or 7-point for precipitation | e.g. Nicholson (2001), Nash and Endfield (2002), Kelso and Vogel (2007), Grab and Nash (2010), Nicholson et al. (2012a), Nash et al. (2016), Grab and Zumthurm (2018) |
| **Americas** | 3-point for temperature, 5- or 7-point for floods / precipitation; 3-point for snowfall | e.g. Baron et al. (1984), Prieto (1984), Baron (1989, 1995), Prieto et al. (1999), Prieto and Rojas (2015), Gil-Guirado et al. (2016) |
| **Asia** | 4- or 5-point most common for temperature / precipitation and floods/drought | e.g. Zhu (1926), Zhang and Zhang (1979), Wang and Wang (1990a), Academy of Meteorological Science of China Central Meteorological Administration (1981), Wang and Wang (1990b), Wang et al. (1998), Tan and Wu (2013), Tan et al. (2014), Ge et al. (2018) |
| **Australia** | 3-point for precipitation | e.g. Fenby and Gergis (2013), Gergis and Ashcroft (2013), Gergis et al. (2018) |
| **Oceans** | 1-, 4- or 8-point for wind direction, 12-point for wind speed | e.g. Garcia et al. (2001), Prieto et al. (2005), Küttel et al. (2010), Barriopedro et al. (2014), Barrett et al. (2018), García-Herrera et al. (2018) |

**10.2. Guidelines for generating future documentary-based indices**

The diversity of practice revealed in this review raises two issues. First, different approaches to index development make it harder for climate historians and historical climatologists working in different parts of the world to compare their climate indices directly, since each will include indices with differing climatological boundaries. Second, they make it harder for (palaeo)climatologists to use the resulting time series in synthesis and modelling studies without recourse to the methodology used in each original study. As noted in section 9.2, fully calibrated series have been included within global climate compilations such as the PAGES 2k Consortium (2013, 2017) temperature syntheses. Non-calibrated index series have also been incorporated into multi-proxy reconstructions using the "Pseudo proxy" approach of Mann and Rutherford (2002) – see, for example, Neukom et al. (2014a) and Neukom et al. (2014b) – but these types of reconstruction are relatively rare.

Having a standard approach to index-based climate reconstruction would clearly have its benefits. However, we recognise that a 'one size fits all' approach is neither appropriate for all climate phenomena nor for all source types. The reconstruction of historical wind patterns over the oceans from ships' logbooks and the identification of precipitation variability through the analysis of descriptions of rogation ceremonies, for example, already have well-developed methodologies and protocols. We further recognise that the most widely used approaches such as the Pfister method would require modification to be useful for temperature and/or rainfall reconstruction in all regions, since climates with strong seasonality may not have documentary evidence available year-round. Their use would, in some areas, also override the legacy of decades of methodological effort and require the reanalysis of enormous volumes of documentary evidence.

Rather than suggest a prescriptive method, we instead offer a series of guidelines as best practice for generating indices from collections of historical evidence. The guidelines are of greatest relevance to index-based reconstructions of temperature and precipitation from multiple source types but also have resonance for other climate phenomena (e.g. winter severity) and for many single source types (e.g. annals, chronicles, letters, diaries/journals, newspapers). The guidelines are based, in part, on the excellent reviews by Brázdil et al. (2010) and Pfister et al. (2018), but also incorporate insights from this study:

1. Researchers should be familiar with the climatology of their study region, as this may influence the temporal distribution of documentary evidence. Indices should, ideally, be based on collections of historical records that overlap with a climatically homogenous region with respect to the phenomena to be reconstructed.

2. Researchers should be familiar with the strengths and weaknesses of each of their historical sources prior to their use in climate reconstruction.

3. Researchers should select an appropriate temporal resolution for their index series according to the quantity, quality and richness (in terms of climate information) of available historical sources. This may be monthly, seasonal, annual or longer. For information-rich areas, a monthly resolution is optimal as it offers the greatest potential for comparison with early instrumental series (which may be published as monthly averages prior to the wider availability of daily data) and the greatest flexibility for comparison with more coarsely-resolved sources, such as palaeoclimate reconstructions. For regions with marked variations in the quantity and quality of climate information across the year, the choice of resolution may be dictated by the length of period during the year when information is most sparse.

4. Whether to develop a three-, five- or seven- (or more) point index series may also be influenced by the legacy of previous studies in a region if direct comparisons are required; however, following guideline 3, researchers should only generate series with higher numbers of index classes if source density and richness permit.

5. Transforming the information in historical documents to numbers on a scale requires a high degree of expertise to minimise subjectivity and should, ideally, be undertaken by experienced researchers with a good knowledge of the climate of a region and an understanding of the language of the time period in which sources were written.

6. Historical records should ideally be sorted chronologically prior to analysis, with indices developed in a stepwise manner. Pfister et al. (2018, p.120) recommend that indexing begin with the most recent period (a process referred to by Brázdil et al., 2010, as 'hind-casting'), which for most studies will also be the period with the greatest volume of documentary evidence. This allows researchers to become familiar with the vagaries of their evidence during well-documented periods before working backwards to periods where information may be less complete.

7. For regions and periods where large volumes of historical information are available, indices should always be generated using evidence from more than one independent contemporary observer or record. If weather in a region is documented within a single contemporary record, appropriate levels of uncertainty should be noted in the final reconstruction (see Pfister et al., 2018).

8. It is advisable to sum-up index series – either in time (i.e. from monthly to seasonal or annual) or in space (i.e. by combining several index series from a climatologically homogeneous region). Careful assessment is needed, however, to avoid any loss of information during the process of summation, particularly for extreme events (see section 8.1). Potential seasonal biases within documentary sources should also be considered as these will influence annual totals.

9. Where possible, index series should be developed independently from the same set of historical sources by more than one researcher to minimise subjectivity. The final index series for southeast Africa produced by Nash et al. (2016), for example, was first developed independently by two members of the research team who then met to agree the final series.

10. To maximise their wider usefulness, index series should, ideally, overlap with runs of local or regional instrumental data to permit calibration and verification. Where instrumental data are not available, overlaps with independent high-resolution palaeoclimate records may be useful for comparison and testing, noting that palaeoclimate records may have their own biases.

11. If fully calibrated, statistical measures of error should be incorporated into the presentation of any reconstruction.

12. Where insufficient overlapping instrumental data are available to permit full calibration and verification, some form of "Confidence Rating" (see section 9.2 and Kelso and Vogel, 2007) should be incorporated into the presentation of any reconstruction.

13. Finally, as Pfister et al. (2018, p.121) identify, the purpose and process of index development should be "fully transparent and open to critical evaluation", with the method of index development described in detail and a source-critical evaluation of the underlying evidence included.

There remain vast collections of documentary evidence from all parts of the globe that that have yet to be explored for information about past climate. We hope that, if such collections are scrutinised following these guidelines, they will lead to index-based reconstructions of climate variability that can be used to both extend climate records and contextualise studies of climate-society relationships to the wider benefit of humankind.

**Dedication.** This paper is dedicated to the memory of María del Rosario Prieto, a pioneer in historical climatology and active promoter of climate history studies in South America, who sadly passed away in 2020 during the preparation of the first draft of this manuscript. Rest in peace, María.

**Author contributions.** DJN, MB, CC and TL conceived the original study. Overall manuscript development was led by DJN. All authors contributed to the writing of the first draft of the paper and to the preparation of the final manuscript.

**Competing interest.** The authors declare that they have no conflict of interest.

**Acknowledgements.** The authors would like to thank: PAGES (Past Global Changes) for supporting CRIAS Working Group meetings in Bern (2018) and Leipzig (2019) that led to the conception and subsequent development of this publication; the Leibniz Institute for the History and Culture of Eastern Europe (Leipzig, Germany), Oeschger Center for Climate Change Research (Bern, Switzerland) and Education University of Hong Kong (Peoples Republic of China) for supporting open access publication charges; and Lina Lerch (Leipzig) for help on Japanese climate-historical sources.

**Financial support.** The meetings that underpinned this article were supported by PAGES (Past Global Changes). The article processing charges for this open-access publication were covered by the Leibniz Institute for the History and Culture of Eastern Europe, Oeschger Center for Climate Change Research and Education University of Hong Kong.

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
