# Peer review of "Climate indices in historical climate reconstructions: A global state-of-the-art"

_Climate of the Past, 2020_

## Referee Comment (RC1) · Anonymous Referee #1 · 4 Nov 2020

General comments This paper focuses on a very important topic concerning methodological issues related to interpretation of historical documents and compilation of so-called index series of various meteorological elements. In this sense, the paper may help to understand better the methodology to scientists outside the historical climatology community. This contribution provides an overview how index series are compiled and how they are used in climate reconstruction in different parts of the world. The paper is well written with a clear structure. In spite of that, there are several issues requiring clarification or better explanation.

Specific comments In the introductory part, three main categories of information are mentioned that appear in historical documents and inscriptions (lines 32–35) and in the following paragraph authors state that the generation of ordinal-scale indices is a

common approach for the analysis of the third category – descriptive (or narrative) evidence. However, in the following sections, they mention numerous examples of indexing approach also for the two remaining categories – e.g. sea-ice index (Ogilvie, 1996), phenology-related phenomena from China (section 3.5) or even indices derived from early instrumental measurements (Figure 5 or section 6.2, lines 519–520). I would very recommend to provide somewhere in the introductory part at least some explanation why such type of information (quite often already existing at least on the ordinal scale) is transformed to indices. It would be quite useful to add some simple categorization of indices.

Sections 2–7 provide a detail overview of various index types that different authors compiled at individual continents and ocean according to the meteorological element reconstructed. Too much space is devoted to the scale of index series. At the same time, it is mentioned several times in the text that number of points (or granularity) is dictated above all by the quality and abundance of documentary evidence (e.g. lines 136, 614). In my opinion, more information should be provided on different characteristics of the index series in this part of the text. Those are e.g. the completeness of the index series, their temporal coverage, the way the missing information is handled, meaning of the "zero" category, overlap with the target data for quantitative reconstruction and so on. Authors mention such characteristics only sporadically.

The 3.2 section provides very detailed description of diverse Chinese documentary sources, often not used for index series construction. Moreover, this part is quite long, not directly related to the topic of indices in some cases and it has no corresponding counterpart e.g. for Europe.

Section 8 on methodological approaches used to derive indices appears the most important for those searching for "good practice for future studies" and for advice how to derive indices from their own data. In this sense, however, at least some approaches mentioned here would deserve a short comment or some sort of critics (Section 8.3, end of the first paragraph: Correlation coefficient is a relative measure and the value

of 0.5 means that compared data sources share only 25% of common variability. Statistical significance of the correlation would be much more relevant).

The same holds for some statements in Section 9. Please check lines 821–829. The whole paragraph is hard to understand and it does not make sense – at least from statistical point of view. It is not clear how "... chi-square tests, comparisons with the eigenvectors ... and the standard error of the estimate" can be used "to derive transfer functions". For instance, the standard error of the estimate is the result of the transfer function calculation. Thus, it cannot be used to derive it. Similarly: "Such correlations can further be compared and calibrated using instrumental data". Please re-formulate as correlations (of what?) can be hardly "calibrated".

In section 9.2 on confidence and uncertainty there is a discrepancy between the title of this section and the text that follows. "Uncertainties in index-based climate reconstructions" are different from uncertainties related to the index series compilation. Both types of uncertainty are very important, however, they have several different reasons and different origin. Unfortunately, the text provides only some examples of the second type of uncertainty (related to the index series compilation). It would be very useful to mention at least some examples of the first one (Dobrovolný et al., 2010). Ability to quantify uncertainties in the index-based reconstructions (either formally – with some statistics or less formally – by comparison with other reconstructions) makes them fully comparable to natural proxy-based quantitative reconstructions.

It is obvious that this overview cannot refer to all relevant studies. However, I would recommend to mention in the text several other studies especially from Europe. They can be an important example of the indexing approach (Koslowski and Glaser 1999; Dobrovolný et al., 2015), example of multiproxy reconstructions using temperature (Luterbacher et al. 2004) or precipitation indices (Pauling et al. 2006) or papers important from the methodological point of view (Dobrovolný et al. 2009, Brázdil et al. 2016).

A suggestion for the Section 10.2, concluding recommendations: Even if the index series are constructed at several-degree scales (7 or more points), indexing always means suppressed variability of index series compared either to target data (instrumental measurements) or to natural proxies (e.g. tree rings). It is advisable to sum-up index series – either in time (from monthly to seasonal or annual) or in space (put together several index series form climatologically homogeneous region). This approach may well approximate index series to natural climate variability.

Minor comments Line 43 – the term "unweighted" index may be misleading here. Line 396 – ". . . that Henry Lamb was developing . . ." Here should be "Hubert Lamb", I guess. Line 626 – ". . . to define index categories: -/+180% for index values -3/+3, -/+130% for values -2/+2, and +/-65% for values +1/-1." Percent of what? This text is confusing. Please add more explanation. Line 637 – add "decadal" otherwise not clear: "where ðÌŚĞðÌŚŰ is the DECADAL winter temperature index. . ." Lines 694 – 695 ". . . the presence of key descriptors is used to distinguish these categories." Not clear, please re-formulate. Line 696 – "Algorithms are then used to weight and combine documentary and instrumental data" Not clear, please re-formulate. Table 3, 5 – There are some empty fields, please add something like "not available" or "not relevant" to avoid misinterpretation. In case of Table 5 please explain "qualitative indication" XXX means the best quality?

References Brázdil R, Dobrovolný P, Trnka M, Buntgen U, ÅŸezníčková L, Kotyza O,Valášek H, Štěpánek P. 2016. Documentary and instrumental-based drought indices for the Czech Lands back to AD 1501. Clim. Res. 70: 103–117. https://doi.org/10.3354/cr01380.

Dobrovolný P, Brázdil R, Valášek H, Kotyza O, Macková J, Halíčková M. 2009. A standard paleoclimatological approach to temperature reconstruction in historical climatology: an example from the Czech Republic, AD 1718–2007. Int J Climatol 29(10):1478–1492. doi:10.1002/joc.1789

[Figure]

Dobrovolná P, Brázdil R, Trnka M, Kotyza O, Valašek H. 2015. Precipitation reconstruction for the Czech Lands, AD 1501–2010. Int. J. Climatol. 35: 1–14. https://doi.org/10.1002/joc.3957.

Koslowski G, Glaser R. 1999. Variations in reconstructed ice winter severity in the Western Baltic from 1501 to 1995, and their implications for the North Atlantic oscillation. Climatic Change 41 (2), 175–191.

Luterbacher J, Dietrich D, Xoplaki E, Grosjean M, Wanner H. 2004. European seasonal and annual temperature variability, trends and extremes since 1500. Science 303:1499–1503. doi:10.1126/science.1093877

Pauling A, Luterbacher J, Casty C,Wanner H. 2006. Five hundred years of gridded high-resolution precipitation reconstructions over Europe and the connection to large-scale circulation. Clim. Dyn. 26: 387–405. https://doi.org/10.1007/s00382- 005-0090- 8.
* * *

---

## Referee Comment (RC2) · Anonymous Referee #2 · 8 Nov 2020

This paper provides a review of the work done when trying to convert documentary evidences into ordinal or quantitative indices. There are two aims: 'provide a global state-of-the-art review of the development and application of the index approach in historical climate reconstruction' and 'identify best practice for future investigations'. After the introduction sections 2-9 review the previous work done in the different world areas, while section 10 provides some recommendations for future work.

I think that the paper is rather successful in the first objective but fails in achieving the second one because the abundant description of previous work included in sections 2-9 is not followed by a critical analysis in section 10. I provide details below.

Regarding objective one, the review is exhaustive reflecting most of the previous work based on ordinal indices that consider the departure from normality as the main crite-

[Figure]

rion to produce an anomaly scale with several levels of intensity. However, references to other approaches to build ordinal indices are missing. For instance, several papers have built ENSO chronologies from documentary evidences from different areas of S America reporting different impacts associated to ENSO (Quinn and Neal, 1992; Ortlieb 2000; Garcia-Herrera et al 2008). In my view this type of approach should also be acknowledged in the paper.

Lines 777-778 In the recent years directional wind indices over the oceans have gone beyond decadal reconstructions of wind force trends, as stated in the paper. This methodology has allowed the generation of the longest series of the wind circulation in the North Atlantic and generating new indices for circulation patterns as the NAO or the East Atlantic pattern (Mellado-Cano et al 2020). Besides, they have been useful in studying different features of the global monsoon system: the impact of volcanic eruptions on the West African Summer monsoon during the 19th century (Gallego et al 2015), the onset of the Indian Summer Monsoon (Ordoñez et al 2016) or secular trends in the Australian Summer Monsoon (Gallego et al 2017) among others.

Lines 865-873. Over the Oceans the uncertainties associated to the limited sampling in a given area and period have been also quantified, see for instance Gallego et al (2015).

In my view the second objective is not achieved because there is not a critical analysis of the work described in sections 2-9. Consequently, the link from the recorded evidences to the identification of the best practices is missing. This should have been done in section 10, but this is again very descriptive. Tables 5-7 do not identify best practice, instead they just summarize the variables studied in every region or the number of classes used. Having missed this analytical part, many of the statements lack of support. The authors claim that they are based on two previous reviews and 'also incorporate insights from this study'. This is not evident at all from the text, because of this lack of critical analysis in the manuscript. For instance (lines 950-952) why do the authors "recognise that the most widely used approaches such as the Pfister method

would require modification to be useful for temperature and/or rainfall reconstruction in all regions"? Which of the previous papers are the support of this statement? What are the main reasons for this recognition? The authors do not provide any evidence of the limitations of these approaches and they should do it based on their extensive previous review. Is it because indices derived from a certain type of documents and for a given climate cannot be applied mimetically to different documentary sources and climates? If so, the authors should provide supporting evidence. Otherwise this is just their opinion.

I find several problems with the guidelines. Firstly, they should be clearly supported from the previous review, which is not the case. The review should allow identifying best practices and the analysis of these cases should lead to the guidelines, but this sequence is not followed in the paper. Bets practices are not identified, and, consequently, guidelines are not supported by them.

Additionally, I think that a climate component is missing in some of the guidelines. According to my previous experience, the final indexation should be a compromise among the historical records characteristics, their availability and the climate of the region to be studied. The mere translation of indices built for a certain climate to other areas may lead to biases or inadequacies if applied to other regions. For example, the translation of precipitation indices developed for central Europe should be applied with care to Mediterranean climates, where most of the precipitation is concentrated between September and April and occurs mostly in the form of a few intense events.

Thus, I think that guideline 2 could be rephrased from: Researchers should be familiar with the strengths and weaknesses of each of their historical sources prior to their use in climate reconstruction. To something as: Researchers should be familiar with the local climate and the strengths and weaknesses of each of their historical sources prior to their use in climate reconstruction.

Guideline 3 reads: "Researchers should select an appropriate temporal resolution for

their index series according to the quantity and richness (in terms of climate information) of available historical sources. This may be monthly, seasonal, annual or longer, although for information-rich areas, a monthly resolution is the most desirable." Again the climate factor is missing, for instance if you build monthly series for precipitation in the Mediterranean, you should be aware that during the dry months, the signal-to-noise ratio should be very low and this can bias the results. This guideline should be rephrased, as, for example: "Researchers should select an appropriate temporal resolution for their index series according to the quantity and richness (in terms of climate information) of available historical sources and the local climate. This may be monthly, seasonal, annual or longer, although for information-rich areas, a monthly resolution is desirable depending on the climate type and variable studied."

Guideline 4 reads: "Whether to develop a three-, five- or seven- (or more) point index series will also depend upon data quantity and quality but may be influenced by the legacy of previous studies in a region if direct comparisons are required". Two comments here. I do not understand the mention to the previous legacy, I find this confusing. Do you mean that things should be done as they were done in the past, just to compare? Even if you have identified problems in the legacy? This needs to be clarified. Applying previous indexation without a careful examination of its adequacy to a given climate and data set is not a good practice. Apart from this, a mention to the climate is also required, since the number of points in the scale may also depend on the type of climate and climate variable studied. So, for instance, this could read something as: "Whether to develop a three-, five- or seven- (or more) point index series will also depend upon data quantity and quality, the local climate and climate variable to be indexed".

Guideline 9 reads "To maximise their wider usefulness, index series should, ideally, overlap with runs of local or regional instrumental data to permit calibration and verification. Where instrumental data are not available, overlaps with independent high-resolution palaeoclimate records may be used for calibration" I think that using palaeoclimate proxies to calibrate an index is not the best recommendation. These proxies have their own weaknesses and uncertainties and using them as the 'truth' to calibrate an index may introduce unexpected biases. Calibrating an index with a proxy, implies two transfer functions from the variable to the proxy and from the proxy to the index, posing additional uncertainties. I think that a comparison with proxies is fine, but using them to calibrate is far too dangerous.

Summing up, I think that the paper requires and extensive revision before being acceptable for publication. The good practices need to be well identified in the text and the support of the guidelines must be clearly linked to the previous evidence. The authors have made a highly valuable effort in compiling the previous work. Improving the analysis by better illustrating the good practices and providing a clear background and support to the guidelines, would lead to a highly interesting paper, but these issues need to be solved.

Minor issue Some of the authors references are made in a strange way. For instance, line 455 'Garza Merodio who was a student of . . ..' Is this so relevant? Why are not the other academic linkages mentioned? To me this is relevant if you want to tell the history of the researchers involved in this topic, which is not the case. Line 401 'the work of Coleen Vogel'. Line 407,' Sharon Nicholson' and several others. Why some authors are cited by their full names (not the usual practice) and other just by the surname? Not clear to me.

Gallego, D., Ordóñez, P., Ribera, P., Peña-Ortiz, C., and García-Herrera, R.: An instrumental index of the West African Monsoon back to the 19th century, Q. J. Roy. Meteor. Soc., 141, 3166-3176, doi: 10.1002/qj.2601, 2015. Gallego D. García-Herrera R., Peña-Ortiz C. and Ribera P. 'The steady increase of the Australian Summer Monsoon in the last 200 years'. Scientific Reports, 7, Article number: 16166. 2017. doi:10.1038/s41598-017-16414-1. García-Herrera R., Díaz H.F., García R.R., Prieto M.R., Barriopedro D., Moyano R., Hernández, E. (2008): A chronology of El Niño events from primary documentary sources in Northern Peru. Journal of Climate,

21, 9, 1948-1962, doi: 10.1175/2007JCLI1830.1. Mellado-Cano J., Barriopedro D., García-Herrera R., Trigo R., Hernández A., (2019): Examining the North Atlantic Oscillation, East Atlantic Pattern, and Jet Variability since 1685, Journal of Climate, 32, 6285–6298, doi: https://doi.org/10.1175/JCLI-D-19-0135.1 Mellado-Cano J., Barriopedro D., García-Herrera R., Trigo R., (2020): New observational insights into the atmospheric circulation over the Euro-Atlantic sector since 1685, Climate Dynamics, doi: https://doi.org/10.1007/s00382-019-05029-z Ordóñez P, David Gallego, Pedro Ribera, Cristina Peña-Ortiz and Ricardo García-Herrera. 'Tracking the Indian Summer Monsoon onset back to the pre-instrumental period'. J. Climate, 29 (22), 8115-8127, 2016. doi:10.1175/JCLI-D-15-0788.1. Ortlieb, L., 2000: The documentary historical record of El Niño events in Peru: An update of the Quinn record. El Niño and the Southern Oscillation: Multiscale Variability and Global and Regional Impacts, H. F. Diaz and V. Markgraf, Eds., Cambridge University Press, 207–297 Quinn W, and V.T. Neal, 1992: The historical record of El Niño events. Climate Since A.D. 1500, R. S. Bradley and P. D. Jones, Eds., Routledge, 623–648

---

## Short Comment (SC1) · 1 Dec 2020

**Comment on "Climate indices in historical climate reconstructions: A global state-of-the-art" by David J. Nash et al.**

F. Domínguez-Castro and J.M. Vaquero

The documentary sources provide a huge quantity of information about climate variability of the past. Nevertheless, there is high variability in the quality, quantity, formats, contents, objectives, etc. of the climatic information provided by these sources. This makes very difficult to unify indexing methodologies for all the documentary sources. For this reason, a review article of the indexing methodologies and the identification of indexing best practice is necessary, pertinent, and really welcomed. We want to provide some comments that can be useful to broaden the discussion about indexing.

We consider that an article that wants to provide a "Guidelines for generating future documentary-based indices" must analyse the advantages, disadvantages as well as the limitations of the different indexing methods used until now. We think that this discussion is missing in the manuscript and we provide here some topics that could be interesting to address:

**Use of different kinds of documentary sources**: there are indices constructed from a unique documentary source, e.g a private diary (Brázdil et al., 2008; Domínguez-Castro et al., 2015), series of acts of municipal and ecclesiastical institutions for a location (Barriendos, 1997; Domínguez-Castro et al., 2018), series of correspondence (Fernández-Fernández et al., 2014; Rodrigo et al., 1998)… However, other indices are constructed putting together information from many different documentary sources (e.g Brázdil et al., 2016; Camuffo et al., 2010). Probably in the first case, it is easier to detect and correct unexpected bias or homogeneity problems. In the second case you can usually analyse longer periods or larger regions, but at the risk of including inhomogeneities.

**Subjectivity in the indexation**: probably, the unique objective indices are those based on the presence or absence of a meteorological phenomenon, e.g rainfall, snowfall, wind, fog… If one has this information at daily scale, the index is almost directly comparable with the instrumental series e.g. Domínguez-Castro et al., 2019. It is easy to think that the subjectivity increases with the number of categories of the indices, but more categories provide more variability. On the other hand, there are also many "uncategorized indexes", a few examples are: appearance of ice and freeze-up dates (Takács et al., 2018), flowering /grape harvest dates (Aono and Kazui, 2008), number of days under drought conditions (Domínguez-Castro et al., 2008)… This difference could be discussed deeply.

**Resolution**: the time resolution of an index can be daily, weekly, monthly, seasonal, annual or larger time resolution. Are there problems when, for instance the annual/seasonal indices are built just adding seasonal/monthly indices? Frequently, the documentary sources have bias to some season due to the climate and/or the nature of the documents. It is common that the documentary sources provide more information about extreme seasons (winter or summer), or in specific periods of the year in which the climate was determinant for some agricultural labour. For this reason, one must be very carefully when computing annual index just from the addition of seasonal indices and this could be reflected in the text.

**The meaning of "0 index"**: frequently "0 index" is considered as "normal" condition, but this can be confounded with cases when no information is available. We need to be sure to assume it, that the consulted documentary sources cover the entire period with the same quality. This is easier to evaluated when only one documentary source is used for the whole studied period.

**The distribution of the index values**: For reconstruction purposes, it is useful that the index shows similar distribution to the variable to be reconstructed. Nevertheless, this is not always possible, as in some examples provided in this paper (e.g. figure 2 shows a possible bias to negative values, and Mertz´s reconstruction shows less "-1 values" (14) than "-2 values" (20), when probably the opposite is expected). Something similar happens in Figure 8 that shows less "1 values" (abundant flows, mainly concentrated in the last decades) than large swells ("2 values"). Probably, this is because the used documentary sources record better the extreme than the more common events. Anyway, we think it is an interesting topic to discuss in the paper.

In general, the classic methodology that uses three, five or seven indices is deeply discussed in the paper compared to other methodologies for each type of documentary source that have appeared in recent years. We consider that these particular methodologies have enormous advantages for performing reconstructions and they should be discussed.

We understand that it is impossible to cite all the published articles in the field or to analyse all the methodologies developed in a single paper for all the meteorological variables or events. However, we think that it is possible to include more recent publications. Here we provide some examples.

Regarding the European temperature indices, the more recent work cited dates from 2015. We recommend to include some recent papers as (Brázdil et al., 2019; Fernández-Fernández et al., 2017; Fila et al., 2016; Filipiak et al., 2019; Mrgic, 2018; Rodrigo, 2019). Some of the methodological approaches of these papers are interesting and have not been discussed in the manuscript.

With respect to European precipitation indices, the most recent work cited dates from 12 years ago. Some proposals to update these references are (Brázdil et al., 2019; Bullón, 2011; Fernández-Fernández et al., 2015; Filipiak et al., 2019; Metzger and Tabeaud, 2017; Rodrigo, 2019). Again, these papers have interesting methodologies that have not been discussed.

Moreover, the authors affirm in the section "Climate indices in Europe" that the information about temperature is more frequent and with a better quality than the precipitation information (e.g., line 124 "Temperature is the most common meteorological phenomenon analysed in Europe" or line 154 "Often the same scale is applied for both temperature and precipitation indices; however, precipitation indices may show more gaps than their temperature counterparts as data may be seasonal or more sporadic"). We think this assumption is true for central and northern Europe, but it is not for the Mediterranean region, for example, where the references to the lack or excess of precipitation are clearly more frequent than those for temperature. This is because the Mediterranean climate is more temperate but has important precipitation extremes. This

is a very important point in the article, because the effect of the different climates in the indexing is almost missing in the review, and this is a key issue in a review wanting to cover different regions of the globe.

A mention to pro pluvia rogation ceremonies as a documentary proxy of European droughts is missing. This proxy has been used in the last decades to understand drought variability in preinstrumental period in various European countries as Spain (Barriendos, 2010; Bravo-Paredes et al., 2020; Domínguez-Castro et al., 2008, 2010, 2012; Tejedor et al., 2018), France (Garnier, 2010), Italy (Piervitali and Colacino, 2001) or Portugal (Fragoso et al., 2018). Moreover, different methodologies have been used to extract the climate information of this proxy as the liturgical act (Martín-Vide and Vallvé, 1995), the expenses (Álvarez Vázquez, 1986), the period of time with continuous rogations (Domínguez-Castro et al., 2008) or the area occupied by the text dedicated to the celebration in the chapter acts (Gil-Guirado et al., 2019). All these methodologies are different to the three, five or seven indices discussed in this paper, and have their advantages and disadvantages that would require a detailed analysis.

Moreover, it would be interesting to cite articles of other variables, as sea level (Camuffo et al., 2017; Camuffo and Sturaro, 2003) in which interesting methods have been developed or about snowfalls reconstruction (Enzi et al., 2014).

Additionally, we can point out one minor suggestion:

Line 474 "Finally, Dominguez-Castro et al. (2018) built a long precipitation series for 1891–2015 CE based on descriptions of rain ceremonies in Quito, Ecuador". This is not correct. Domínguez-Castro et al. (2018) presents a precipitation instrumental series from Quito (1891-2015) and a series of wet and dry extremes from rogation ceremonies from 1600.

Finally, we would like to thank the authors to undertake a work of this magnitude, and we hope that our comments will be useful.

References

Álvarez Vázquez, J. A.: Drought and Rainy Periods in the Province of Zamora in the 17th, 18th, and 19th Centuries, in Quaternary Climate in Western Mediterranean, pp. 221–233, Universidad Autonoma de Madrid., 1986.

Aono, Y. and Kazui, K.: Phenological data series of cherry tree flowering in Kyoto, Japan, and its application to reconstruction of springtime temperatures since the 9th century, Int. J. Climatol., 28(7), 905–914, doi:10.1002/joc.1594, 2008.

Barriendos, M.: Climatic variations in the Iberian Peninsula during the late Maunder minimum (AD 1675-1715): An analysis of data from rogation ceremonies, Holocene, 7(1), 105–111, doi:10.1177/095968369700700110, 1997.

Barriendos, M.: Climate change in the Iberian Peninsula: Indicator of rogation ceremonies (16th-19th centuries) , Rev. d Hist. Mod. Contemp., 57(3), 131–159, doi:10.3917/rhmc.573.0131, 2010.

Bravo-Paredes, N., Gallego, M. C., Domínguez-Castro, F., García, J. A. and Vaquero, J.

M.: Pro-pluvia rogation ceremonies in extremadura (Spain): Are they a good proxy of winter NAO?, Atmosphere (Basel)., 11(3), doi:10.3390/atmos11030282, 2020.

Brázdil, R., Černušák, T. and Řezníčková, L.: Weather information in the diaries of the Premonstratensian Abbey at Hradisko, in the Czech Republic, 1693-1783, Weather, 63(7), 201–207, doi:10.1002/wea.264, 2008.

Brázdil, R., Dobrovolný, P., Trnka, M., Büntgen, U., Řezníčková, L., Kotyza, O., Valášek, H. and Štěpánek, P.: Documentary and instrumental-based drought indices for the Czech Lands back to AD 1501, Clim. Res., 70(2–3), 103–117, doi:10.3354/cr01380, 2016.

Brázdil, R., Valášek, H., Chromá, K., Dolák, L., Řezníčková, L., Bělínová, M., Valík, A. and Zahradníček, P.: The climate in south-east Moravia, Czech Republic, 1803-1830, based on daily weather records kept by the Reverend Šimon Hausner, Clim. Past, 15(4), 1205–1222, doi:10.5194/cp-15-1205-2019, 2019.

Bullón, T.: Relationships between precipitation and floods in the fluvial basins of Central Spain based on documentary sources from the end of the 16th century, Nat. Hazards Earth Syst. Sci., 11(8), 2215–2225, doi:10.5194/nhess-11-2215-2011, 2011.

Camuffo, D. and Sturaro, G.: Sixty-CM submersion of Venice discovered thanks to Canaletto's paintings, Clim. Change, 58(3), 333–343, doi:10.1023/A:1023902120717, 2003.

Camuffo, D., Bertolin, C., Barriendos, M., Dominguez-Castro, F., Cocheo, C., Enzi, S., Sghedoni, M., della Valle, A., Garnier, E., Alcoforado, M.-J., Xoplaki, E., Luterbacher, J., Diodato, N., Maugeri, M., Nunes, M. F. and Rodriguez, R.: 500-Year temperature reconstruction in the Mediterranean Basin by means of documentary data and instrumental observations, Clim. Change, 101(1), 169–199, doi:10.1007/s10584-010-9815-8, 2010.

Camuffo, D., Bertolin, C. and Schenal, P.: A novel proxy and the sea level rise in Venice, Italy, from 1350 to 2014, Clim. Change, 143(1–2), 73–86, doi:10.1007/s10584-017-1991-3, 2017.

Domínguez-Castro, F., Santisteban, J. I., Barriendos, M. and Mediavilla, R.: Reconstruction of drought episodes for central Spain from rogation ceremonies recorded at the Toledo Cathedral from 1506 to 1900: A methodological approach, Glob. Planet. Change, 63(2–3), 230–242, doi:10.1016/j.gloplacha.2008.06.002, 2008.

Domínguez-Castro, F., García-Herrera, R., Ribera, P. and Barriendos, M.: A shift in the spatial pattern of Iberian droughts during the 17th century, Clim. Past, 6(5), 553–563, doi:10.5194/cp-6-553-2010, 2010.

Domínguez-Castro, F., Ribera, P., García-Herrera, R., Vaquero, J. M., Barriendos, M., Cuadrat, J. M. and Moreno, J. M.: Assessing extreme droughts in Spain during 1750-1850 from rogation ceremonies, Clim. Past, 8(2), 705–722, doi:10.5194/cp-8-705-2012, 2012.

Domínguez-Castro, F., García-Herrera, R. and Vaquero, J. M.: An early weather diary from Iberia (Lisbon, 1631-1632), Weather, 70(1), 20–24, doi:10.1002/wea.2319, 2015.

Domínguez-Castro, F., García-Herrera, R. and Vicente-Serrano, S. M.: Wet and dry extremes in Quito (Ecuador) since the 17th century, Int. J. Climatol., 38(4), 2006–2014, doi:10.1002/joc.5312, 2018.

Domínguez-Castro, F., Gallego, M. C., Vaquero, J. M., García Herrera, R., Peña-Gallardo, M., El Kenawy, A. and Vicente-Serrano, S. M.: Twelve years of daily weather descriptions in North America in the eighteenth century (Mexico City, 1775–86), Bull. Am. Meteorol. Soc., 100(8), 1531–1547, doi:10.1175/BAMS-D-18-0236.1, 2019.

Enzi, S., Bertolin, C. and Diodato, N.: Snowfall time-series reconstruction in Italy over the last 300 years, Holocene, 24(3), 346–356, doi:10.1177/0959683613518590, 2014.

Fernández-Fernández, M. I., Gallego, M. C., Domínguez-Castro, F., Trigo, R. M., García, J. A., Vaquero, J. M., González, J. M. M. and Durán, J. C.: The climate in Zafra from 1750 to 1840: history and description of weather observations, Clim. Change, 126(1–2), 107–118, doi:10.1007/s10584-014-1201-5, 2014.

Fernández-Fernández, M. I., Gallego, M. C., Domínguez-Castro, F., Trigo, R. M. and Vaquero, J. M.: The climate in Zafra from 1750 to 1840: precipitation, Clim. Change, 129(1–2), 267–280, 2015.

Fernández-Fernández, M. I., Gallego, M. C., Domínguez-Castro, F., Trigo, R. M. and Vaquero, J. M.: The climate in Zafra from 1750 to 1840: temperature indexes from documentary sources, Clim. Change, 141(4), 671–684, doi:10.1007/s10584-017-1910-7, 2017.

Fila, G., Tomasi, D., Gaiotti, F. and Jones, G. V: The Book of Vinesprouts of Kőszeg (Hungary): a documentary source for reconstructing spring temperatures back to the eighteenth century, Int. J. Biometeorol., 60(2), 207–219, doi:10.1007/s00484-015-1018-6, 2016.

Filipiak, J., Przybylak, R. and Oliński, P.: The longest one-man weather chronicle (1721–1786) by Gottfried Reyger for Gdańsk, Poland as a source for improved understanding of past climate variability, Int. J. Climatol., 39(2), 828–842, doi:10.1002/joc.5845, 2019.

Fragoso, M., Carraça, M. D. G. and Alcoforado, M. J.: Droughts in Portugal in the 18th century: A study based on newly found documentary data, Int. J. Climatol., 38(15), 5522–5541, doi:10.1002/joc.5745, 2018.

Garnier, E.: Exceptional meanness water and hot weather 500 years of drought and heat wave in France and neighboring countries | Bassesses extraordinaires et grandes chaleurs. 500 ans de sécheresses et de chaleurs en France et dans les pays limitrophes, Houille Blanche, (4), 26–42, doi:10.1051/lhb/2010039, 2010.

Gil-Guirado, S., José Gómez-Navarro, J. and Pedro Montávez, J.: The weather behind words-new methodologies for integrated hydrometeorological reconstruction through documentary sources, Clim. Past, 15(4), 1303–1325, doi:10.5194/cp-15-1303-2019, 2019.

Martín-Vide, J. and Vallvé, M. B.: The use of rogation ceremony records in climatic reconstruction: a case study from Catalonia (Spain), Clim. Change, 30(2), 201–221, doi:10.1007/BF01091842, 1995.

Metzger, A. and Tabeaud, M.: Reconstruction of the winter weather in east Friesland at the turn of the sixteenth and seventeenth centuries (1594–1612), Clim. Change, 141(2), 331–345, doi:10.1007/s10584-017-1903-6, 2017.

Mrgic, J.: Intemperate weather in violent times – narratives from the western Balkans during the little ice age (17-18th centuries) , Geogr. Res. Lett., 44(1), 137–169,

doi:10.18172/cig.3380, 2018.

Piervitali, E. and Colacino, M.: Evidence of drought in western Sicily during the period 1565-1915 from liturgical offices, Clim. Change, 49(1–2), 225–238, doi:10.1023/A:1010746612289, 2001.

Rodrigo, F. S.: The climate of Granada (southern Spain) during the first third of the 18th century (1706-1730) according to documentary sources, Clim. Past, 15(2), 647–659, doi:10.5194/cp-15-647-2019, 2019.

Rodrigo, F. S., Esteban-Parra, M. J. and Castro-Diez, Y.: On the use of the Jesuit order private correspondence records in climate reconstructions: A case study from Castille (Spain) for 1634-1648 A.D., Clim. Change, 40(3–4), 625–645, doi:10.1023/a:1005316118817, 1998.

Takács, K., Kern, Z. and Pásztor, L.: Long-term ice phenology records from eastern-central Europe, Earth Syst. Sci. Data, 10(1), 391–404, doi:10.5194/essd-10-391-2018, 2018.

Tejedor, E., de Luis, M., Barriendos, M., Cuadrat, J. M., Luterbacher, J. and Saz, M. Á.: Rogation ceremonies: key to understand past drought variability in northeastern Spain since 1650, Clim. Past Discuss., (June), 1–22, doi:10.5194/cp-2018-67, 2018.

---

## Author Response (AR1)

**"Climate indices in historical climate reconstructions: A global state-of-the-art"**

**Changes to manuscript in response to review comments**

**1. Response to Anonymous Reviewer #1 [RC1]**

We thank the anonymous reviewer for their time and thought, which will help to improve significantly the overall quality of the manuscript. We respond to each question raised in turn:

**[RC1]** In the introductory part, three main categories of information are mentioned that appear in historical documents and inscriptions (lines 32–35) and in the following paragraph authors state that the generation of ordinal-scale indices is a common approach for the analysis of the third category – descriptive (or narrative) evidence. However, in the following sections, they mention numerous examples of indexing approach also for the two remaining categories – e.g. sea-ice index (Ogilvie, 1996), phenology-related phenomena from China (section 3.5) or even indices derived from early instrumental measurements (Figure 5 or section 6.2, lines 519–520). I would very recommend to provide somewhere in the introductory part at least some explanation why such type of information (quite often already existing at least on the ordinal scale) is transformed to indices. It would be quite useful to add some simple categorization of indices.

*[Response to reviewer] The reviewer makes a good point here. We do indeed include examples where ordinal scale data are converted to indices as part of the reconstruction process, and this is especially true for regions outside Europe or at its margins, where narrative information is less available. Almost invariably this occurs when quantitative data are integrated with information from narrative sources to generate indices. Even where instrumental measurements or quantifiable phenological data exist, it may be desirable to develop ordinal indices so that these quantitative data can be combined with descriptive, qualitative information. In this way, it is possible to develop longer, more continuous and homogenous series with a consistent resolution (monthly or seasonal) and hopefully reconstruct both low-frequency and high-frequency variability. To address this point, we will add additional text to the introduction to explain why this is the case and reiterate this point where appropriate in relevant sections of the manuscript (e.g. in the sections on African and Asian index series).*

*[Changes made] Additional text has been added to the final paragraph of section 1 to address this point, with examples flagged in sections 2 to 7.*

**[RC1]** Sections 2–7 provide a detail overview of various index types that different authors compiled at individual continents and ocean according to the meteorological element reconstructed. Too much space is devoted to the scale of index series. At the same time, it is mentioned several times in the text that number of points (or granularity) is dictated above all by the quality and abundance of documentary evidence (e.g. lines 136, 614). In my opinion, more information should be provided on different characteristics of the index series in this part of the text. Those are e.g. the completeness of the index series, their temporal coverage, the way the missing information is handled, meaning of the "zero" category, overlap with the target data for quantitative reconstruction and so on. Authors mention such characteristics only sporadically.

*[Response to reviewer] We take the reviewers point here. We will edit the text to reduce descriptions of the scale of index series and remove any repetitive statements about how the quality and abundance of documentary evidence influences the granularity of index series. We will also add information throughout the manuscript on the completeness of index series and their temporal coverage. The comment about how missing information is handled is a*

*particularly important one. There are two main approaches used to define "0 index" values. One – implicit in the Pfister method – is that no description means no number: a gap in the time series rather than a 0. Other studies make an implicit assumption that, in some circumstances, no weather description can be taken as an indication of normal conditions. We will insert additional text about this in Section 8 and include a paragraph in section 9.2 where we discuss confidence and uncertainty in index-based climate reconstructions.*

*[Changes made] We now note in the introduction that the quantity, resolution and/or richness of the original historical evidence influences the granularity of any reconstruction and have edited down repetitive mentions elsewhere. We have flagged up how "0 index" values are generated throughout section 8 and added text on this issue in section 9.2.*

[RC1] The 3.2 section provides very detailed description of diverse Chinese documentary sources, often not used for index series construction. Moreover, this part is quite long, not directly related to the topic of indices in some cases and it has no corresponding counterpart e.g. for Europe.

*[Response to reviewer] We thought carefully about exactly this point when we were compiling the original manuscript. The nature of documentary sources is well discussed in climate history literature for most parts of the world. However, to our knowledge, there has been no corresponding detail made available for the diverse range of Chinese documentary sources. Hence, even though this text adds to the length of the manuscript, we consider it important for a climate history and historical climatology audience. The same is true for Japanese and early Russian materials, hence the reason we also say more about sources for these regions. We will add an objective to the paper regarding 'the promotion of studies from regions beyond Europe' to encourage specialists in these areas to engage in further work on climate index production.*

*[Changes made] We have added text to the final paragraph of section 1 to justify the inclusion of greater detail about Chinese, Japanese and Russian sources. In the case of Chinese documents there are only a handful of overviews of source types, and for Japan and Russia, to our knowledge, none.*

[RC1] Section 8 on methodological approaches used to derive indices appears the most important for those searching for "good practice for future studies" and for advice how to derive indices from their own data. In this sense, however, at least some approaches mentioned here would deserve a short comment or some sort of critics (Section 8.3, end of the first paragraph: Correlation coefficient is a relative measure and the value of 0.5 means that compared data sources share only 25% of common variability. Statistical significance of the correlation would be much more relevant).

*[Response to reviewer] Thank you for this observation. We will review the text in section 8.3 to ensure that the discussion of index development is sufficiently critical.*

*[Changes made] Two sentences of additional explanation have been added to section 8.3.*

[RC1] The same holds for some statements in Section 9. Please check lines 821–829. The whole paragraph is hard to understand and it does not make sense – at least from statistical point of view. It is not clear how "… chi-square tests, comparisons with the eigenvectors … and the standard error of the estimate" can be used "to derive transfer functions". For instance, the standard error of the estimate is the result of the transfer function calculation. Thus, it cannot be used to derive it. Similarly: "Such correlations can further be compared and calibrated using instrumental data". Please re-formulate as correlations (of what?) can be hardly "calibrated".

*[Response to reviewer] Thank you for this comment. We will review the text in lines 821-829 to improve readability and ensure that it is accurate in its use of statistical terminology.*

*[Changes made] This paragraph has been edited as identified above.*

**[RC1]** In section 9.2 on confidence and uncertainty there is a discrepancy between the title of this section and the text that follows. Both types of uncertainty are very important, however, they have several different reasons and different origin. Unfortunately, the text provides only some examples of the second type of uncertainty (related to the index series compilation). It would be very useful to mention at least some examples of the first one (Dobrovolny et al., 2010). Ability to quantify uncertainties in the index-based reconstructions (either formally – with some statistics or less formally – by comparison with other reconstructions) makes them fully comparable to natural proxy-based quantitative reconstructions.

*[Response to reviewer] Thank you for this very helpful comment. In this section, we are focusing mainly on uncertainties related to index series compilation. We will clarify the text to make sure that this is obvious to the reader, but also mention the suggested example of wider uncertainties in index-based climate reconstruction.*

*[Changes made] We have made edits throughout section 9.2 to (i) note the two forms of uncertainty and (ii) add in some examples of uncertainty in reconstruction.*

**[RC1]** It is obvious that this overview cannot refer to all relevant studies. However, I would recommend to mention in the text several other studies especially from Europe. They can be an important example of the indexing approach (Koslowski and Glaser 1999; Dobrovolny et al., 2015), example of multiproxy reconstructions using temperature (Luterbacher et al. 2004) or precipitation indices (Pauling et al. 2006) or papers important from the methodological point of view (Dobrovolny et al. 2009, Brázdil et al. 2016).

*[Response to reviewer] Thank you for these helpful suggestions. We will review each of the recommended papers and add them to the manuscript where appropriate.*

*[Changes made] Where appropriate, we have woven the recommended papers into section 2 – but note the caveat added to the end of the introduction that we do not include studies unless they include primary documentary reconstructions.*

**[RC1]** A suggestion for the Section 10.2, concluding recommendations: Even if the index series are constructed at several-degree scales (7 or more points), indexing always means suppressed variability of index series compared either to target data (instrumental measurements) or to natural proxies (e.g. tree rings). It is advisable to sum-up index series – either in time (from monthly to seasonal or annual) or in space (put together several index series form climatologically homogeneous region). This approach may well approximate index series to natural climate variability.

*[Response to reviewer] Thank you for these helpful suggestions. We will add a bullet point to this effect to the series of recommendations in section 10.2.*

*[Changes made] We have added a new recommendation (8) to reflect this point.*

Minor comments
**[RC1]** Line 43 – the term "unweighted" index may be misleading here.

*[Response to reviewer] Thank you. We will clarify the text.*

*[Changes made] We have removed the word 'unweighted' from the sentence for clarity.*

**[RC1]** Line 396 – "…that Henry Lamb was developing…" Here should be "Hubert Lamb", I guess.

*[Response to reviewer] Well spotted!*

*[Changes made] Text corrected.*

**[RC1]** Line 626 – "…to define index categories: -/+180% for index values -3/+3, -/+130% for values -2/+2, and +/-65% for values +1/-1." Percent of what? This text is confusing. Please add more explanation.

*[Response to reviewer] Thank you. We will clarify the text.*

*[Changes made] More explanation has been added.*

**[RC1]** Line 637 – add "decadal" otherwise not clear: "where… is the DECADAL winter temperature index…"

*[Response to reviewer] Thank you. We will clarify the text.*

*[Changes made] Corrected.*

**[RC1]** Lines 694 – 695 "…the presence of key descriptors is used to distinguish these categories." Not clear, please re-formulate.

*[Response to reviewer] Thank you. We will expand the text to clarify this.*

*[Changes made] Text clarified.*

**[RC1]** Line 696 – "Algorithms are then used to weight and combine documentary and instrumental data" Not clear, please re-formulate.

*[Response to reviewer] Thank you. We will expand the text to explain this more fully.*

*[Changes made] We have added two short sentences to section 8.3 to clarify this.*

**[RC1]** Table 3, 5 – There are some empty fields, please add something like "not available" or "not relevant" to avoid misinterpretation.

*[Response to reviewer] Thank you. In the case of Table 3, the problem arises from having five index classes in the middle column and only four classes in columns one and three. We will review to see if we can present the table more clearly. For Table 5, we will add text to the table caption to explain the empty fields.*

*[Changes made] We have reviewed Table 3 and can think of no better way to present the data. We have updated Table 5 to provide clarity.*

**[RC1]** In case of Table 5 please explain "qualitative indication" XXX means the best quality?

*[Response to reviewer] Thank you. We will clarify this in the table caption.*

*[Changes made] We have revised the table and expanded the caption.*

**2. Response to Anonymous Reviewer #2 [RC2]**

We thank the anonymous reviewer for their time and thought, which will help to improve significantly the overall quality of the manuscript. We respond to each question raised in turn:

**[RC2]** Regarding objective one [of the paper: 'provide a global state-of-the-art review of the development and application of the index approach in historical climate reconstruction'], the review is exhaustive reflecting most of the previous work based on ordinal indices that consider the departure from normality as the main criterion to produce an anomaly scale with several levels of intensity. However, references to other approaches to build ordinal indices are missing. For instance, several papers have built ENSO chronologies from documentary evidences from different areas of S America reporting different impacts associated to ENSO (Quinn and Neal, 1992; Ortlieb 2000; Garcia-Herrera et al 2008). In my view this type of approach should also be acknowledged in the paper.

*[Response to reviewer] Thank you. This is a very valid point. We will add information about index-based approaches to the development of ENSO chronologies to the most relevant part of the manuscript (Section 5, dealing with 'Climate indices in the Americas').*

*[Changes made] We have added a short paragraph on ENSO chronologies to section 5 and an account of how uncertainties are dealt with as part of index compilation in section 9.2.*

**[RC2]** Lines 777-778 In the recent years directional wind indices over the oceans have gone beyond decadal reconstructions of wind force trends, as stated in the paper. This methodology has allowed the generation of the longest series of the wind circulation in the North Atlantic and generating new indices for circulation patterns as the NAO or the East Atlantic pattern (Mellado-Cano et al 2020). Besides, they have been useful in studying different features of the global monsoon system: the impact of volcanic eruptions on the West African Summer monsoon during the 19th century (Gallego et al 2015), the onset of the Indian Summer Monsoon (Ordoñez et al 2016) or secular trends in the Australian Summer Monsoon (Gallego et al 2017) among others. Lines 865-873. Over the Oceans the uncertainties associated to the limited sampling in a given area and period have been also quantified, see for instance Gallego et al (2015).

*[Response to reviewer] Thank you for these very helpful comments. We will review each of these studies and update sections 7, 8.6 and 9.1 where appropriate.*

*[Changes made] We have edited section 8.6 in light of these suggestions.*

**[RC2]** In my view the second objective is not achieved because there is not a critical analysis of the work described in sections 2-9. Consequently, the link from the recorded evidences to the identification of the best practices is missing. This should have been done in section 10, but this is again very descriptive. Tables 5-7 do not identify best practice, instead they just summarize the variables studied in every region or the number of classes used. Having missed this analytical part, many of the statements lack of support. The authors claim that they are based on two previous reviews and 'also incorporate insights from this study'. This is not evident at all from the text, because of this lack of critical analysis in the manuscript. For instance (lines 950-952) why do the authors "recognise that the most widely used approaches such as the Pfister method would require modification to be useful for temperature and/or rainfall reconstruction in all regions"? Which of the previous papers are the support of this statement? What are the main reasons for this recognition? The authors do not provide any evidence of the limitations of these approaches and they should do it based on their extensive previous review. Is it because indices derived from a certain type of documents and for a given climate cannot be applied mimetically to different

documentary sources and climates? If so, the authors should provide supporting evidence. Otherwise this is just their opinion.

*[Response to reviewer] We take the reviewer's point about lines 950-952 and will expand the text as suggested. We will also review section 9.1 in general to ensure that statements are backed up with examples from the preceding sections. We do not, however, agree with the other views - particularly the suggestion there is insufficient critical analysis of the examples discussed in sections 2 to 9. We embed critical analysis throughout the manuscript. The global coverage of historical climate studies is such that, for many parts of the world, there are not overlapping series that would allow for a direct comparison of the outcomes arising from the use of different methodologies. Where historical studies do overlap, for example in Europe and Africa, we have commented on similarities and differences. Indeed, figure 2 explicitly presents the results of two different studies of overlapping areas in a European context. The purpose of section 9, and section 9.2 in particular, is to not only synthesise the different approaches used to reconstruct climate indices in different parts of the world but also to identify weaknesses. Many of these weaknesses feed directly into the recommendations in section 10.*

*[Changes made] We have edited the second paragraph of section 10.2 in light of the reviewer's comments about lines 950-952 in the original manuscript.*

**[RC2]** I find several problems with the guidelines. Firstly, they should be clearly supported from the previous review, which is not the case. The review should allow identifying best practices and the analysis of these cases should lead to the guidelines, but this sequence is not followed in the paper. Bets practices are not identified, and, consequently, guidelines are not supported by them.

*[Response to reviewer] As discussed in our previous response we do not agree with this viewpoint. Further, section 10 is intended as a conclusion and synthesis. Adding supporting evidence to underpin each of the 12 recommendations would add unnecessary length to an already very long manuscript.*

*[Changes made] No changes are necessitated in response to this comment but see below.*

**[RC2]** Additionally, I think that a climate component is missing in some of the guidelines. According to my previous experience, the final indexation should be a compromise among the historical records characteristics, their availability and the climate of the region to be studied. The mere translation of indices built for a certain climate to other areas may lead to biases or inadequacies if applied to other regions. For example, the translation of precipitation indices developed for central Europe should be applied with care to Mediterranean climates, where most of the precipitation is concentrated between September and April and occurs mostly in the form of a few intense events. Thus, I think that guideline 2 could be rephrased from: Researchers should be familiar with the strengths and weaknesses of each of their historical sources prior to their use in climate reconstruction. To something as: Researchers should be familiar with the local climate and the strengths and weaknesses of each of their historical sources prior to their use in climate reconstruction.

*[Response to reviewer] We agree fully with the reviewer here. However, the recommendations need to be considered as a whole. We already discuss the idea that indices should be developed for climatically homogeneous regions in guideline 1. We do not anywhere suggest that a one size fits all approach to index development would be appropriate. There are numerous examples in the manuscript of where approaches have been tailored to suit climatic variability in an area of interest.*

*[Changes made] We have edited guideline 1 to reflect the need for awareness of local climatic conditions. We specifically mention the case of Mediterranean climates in para 2 of section 8.1 as part of cautionary sentences about using monthly indices.*

**[RC2]** Guideline 3 reads: "Researchers should select an appropriate temporal resolution for their index series according to the quantity and richness (in terms of climate information) of available historical sources. This may be monthly, seasonal, annual or longer, although for information-rich areas, a monthly resolution is the most desirable." Again the climate factor is missing, for instance if you build monthly series for precipitation in the Mediterranean, you should be aware that during the dry months, the signal-to-noise ratio should be very low and this can bias the results. This guideline should be rephrased, as, for example: "Researchers should select an appropriate temporal resolution for their index series according to the quantity and richness (in terms of climate information) of available historical sources and the local climate. This may be monthly, seasonal, annual or longer, although for information-rich areas, a monthly resolution is desirable depending on the climate type and variable studied."

*[Response to reviewer] The key aspect to this particular guideline is that researchers should select an appropriate temporal resolution for their index series based on their data. If, due to the climatic characteristics of an area, observations are relatively sparse for particular seasons then it may not be appropriate to adopt a monthly time scale, regardless of how rich the observations are for other periods of the year. We think that this is explicit in the guideline as it stands.*

*[Changes made] We have edited guideline 3 to make the impact of variations in source density across the year clearer. Note also the changes made in relation to the previous point, which address this issue.*

**[RC2]** Guideline 4 reads: "Whether to develop a three-, five- or seven- (or more) point index series will also depend upon data quantity and quality but may be influenced by the legacy of previous studies in a region if direct comparisons are required". Two comments here. I do not understand the mention to the previous legacy, I find this confusing. Do you mean that things should be done as they were done in the past, just to compare? Even if you have identified problems in the legacy? This needs to be clarified. Applying previous indexation without a careful examination of its adequacy to a given climate and data set is not a good practice. Apart from this, a mention to the climate is also required, since the number of points in the scale may also depend on the type of climate and climate variable studied. So, for instance, this could read something as: "Whether to develop a three-, five- or seven- (or more) point index series will also depend upon data quantity and quality, the local climate and climate variable to be indexed".

*[Response to reviewer] We take the reviewer's points about reference to previous studies and will clarify the text. We disagree, however, about the need to include reference to climate and climate variability in the guideline. The temporal resolution of any index series hinges on the richness of available data across the year. If this varies seasonally then the resolution of the series for the whole year should reflect this.*

*[Changes made] We have edited guideline 4 to provide clarification.*

**[RC2]** Guideline 9 reads "To maximise their wider usefulness, index series should, ideally, overlap with runs of local or regional instrumental data to permit calibration and verification. Where instrumental data are not available, overlaps with independent high resolution palaeoclimate records may be used for calibration" I think that using palaeo-climate proxies to calibrate an index is not the best recommendation. These proxies have their own weaknesses and uncertainties and using them as the 'truth' to calibrate an index may introduce unexpected biases. Calibrating an index with a proxy, implies two transfer

functions from the variable to the proxy and from the proxy to the index, posing additional uncertainties. I think that a comparison with proxies is fine, but using them to calibrate is far too dangerous.

*[Response to reviewer] The reviewer correctly identifies that there is a controversy here. We will address this controversy through the insertion of additional text describing, for example, Andrea Kiss' or Martin Bauch and colleagues' work comparing written records and indices with the Old-World Drought Atlas, as this illustrates the issue well. We do not, however, wish to modify the recommendation. The key words here are "high resolution". We would not recommend calibration using low resolution palaeoclimate series.*

*[Changes made] We have added text to section 9.1 to address this and highlighted the need to use high-resolution palaeoclimate data only.*

**[RC2]** Summing up, I think that the paper requires and extensive revision before being acceptable for publication. The good practices need to be well identified in the text and the support of the guidelines must be clearly linked to the previous evidence. The authors have made a highly valuable effort in compiling the previous work. Improving the analysis by better illustrating the good practices and providing a clear background and support to the guidelines, would lead to a highly interesting paper, but these issues need to be solved.

*[Response to reviewer] Thank you for this. We will adjust the text as outlined above.*

*[Changes made] This comment requires no changes other than those described above.*

**[RC2]** Minor issue Some of the authors references are made in a strange way. For instance, line 455 'Garza Merodio who was a student of : : :.' Is this so relevant? Why are not the other academic linkages mentioned? To me this is relevant if you want to tell the history of the researchers involved in this topic, which is not the case. Line 401 'the work of Coleen Vogel'. Line 407,' Sharon Nicholson' and several others. Why some authors are cited by their full names (not the usual practice) and other just by the surname? Not clear to me.

*[Response to reviewer] First names are used sparingly throughout the manuscript to flag up key researchers who made important contributions in specific regions and/or to identify distinct schools of historical climatology that have transmitted certain methodologies.*

*[Changes made] This comment requires no changes other than those described above.*

**3. Response to Short Comment by Domínguez-Castro and Vaquero [SC1]**

We thank Drs Domínguez-Castro and Vaquero for their time and thought, which will help to improve significantly the overall quality of the manuscript. We respond to each question raised in turn:

**[SC1]** The documentary sources provide a huge quantity of information about climate variability of the past. Nevertheless, there is high variability in the quality, quantity, formats, contents, objectives, etc. of the climatic information provided by these sources. This makes very difficult to unify indexing methodologies for all the documentary sources. For this reason, a review article of the indexing methodologies and the identification of indexing best practice is necessary, pertinent, and really welcomed. We want to provide some comments that can be useful to broaden the discussion about indexing.

We consider that an article that wants to provide a "Guidelines for generating future documentary-based indices" must analyse the advantages, disadvantages as well as the limitations of the different indexing methods used until now. We think that this discussion is missing in the manuscript and we provide here some topics that could be interesting to address:

*[Response to reviewer] Thank you for these observations. As noted in our response to RC2, we do not agree that there is insufficient critical analysis in the manuscript, but we respond now to each of your suggestions in turn.*

*[Changes made] This comment requires no changes to the text.*

**[SC1]** Use of different kinds of documentary sources: there are indices constructed from a unique documentary source, e.g. a private diary (Brázdil et al., 2008; Domínguez-Castro et al., 2015), series of acts of municipal and ecclesiastical institutions for a location (Barriendos, 1997; Domínguez-Castro et al., 2018), series of correspondence (Fernández-Fernández et al., 2014; Rodrigo et al., 1998)... However, other indices are constructed putting together information from many different documentary sources (e.g. Brázdil et al., 2016; Camuffo et al., 2010). Probably in the first case, it is easier to detect and correct unexpected bias or homogeneity problems. In the second case you can usually analyse longer periods or larger regions, but at the risk of including inhomogeneities.

*[Response to reviewer] This is a useful point. We will review the manuscript and distinguish examples where we have indices based on single documentary sources or phenological proxies. This will need to draw on examples from a wider geographical range that the European examples suggested. We will also emphasise the comparative ease of detecting unexpected bias or homogeneity problems in single source series versus multiple source series.*

*[Changes made] The opening paragraph of section 9.2 now directly addresses this issue. We have also checked through the text and flagged up any examples where indices are based on single documentary sources.*

**[SC1]** Subjectivity in the indexation: probably, the unique objective indices are those based on the presence or absence of a meteorological phenomenon, e.g rainfall, snowfall, wind, fog... If one has this information at daily scale, the index is almost directly comparable with the instrumental series e.g. Domínguez-Castro et al., 2019. It is easy to think that the subjectivity increases with the number of categories of the indices, but more categories provide more variability. On the other hand, there are also many "uncategorized indexes", a few examples are: appearance of ice and freeze-up dates (Takács et al., 2018), flowering

/grape harvest dates (Aono and Kazui, 2008), number of days under drought conditions (Domínguez-Castro et al., 2008)... This difference could be discussed deeply.

*[Response to reviewer] We take your point about subjectivity in indexation (which has already been discussed at length in the literature). However, we are not sure of the relevance of what you refer to as 'uncategorized indices' for our paper. Taking the example of ice phenology from Takács et al. (2018), this study focusses on the timing of freeze up rather than developing indices of winter severity from such information. The mere existence of a dataset that might be used to produce indices does not, in our eyes, qualify a study for inclusion in our already lengthy manuscript. We will, however, check the suggested literature to make sure we haven't missed any examples of actual index approaches (note: Aono and Kazui (2008) is mentioned in section 3.1).*

*[Changes made] We have reviewed the text as indicated and added a caveat to the end of the introduction justifying the inclusion (and exclusion) of studies from the manuscript.*

**[SC1]** Resolution: the time resolution of an index can be daily, weekly, monthly, seasonal, annual or larger time resolution. Are there problems when, for instance the annual/seasonal indices are built just adding seasonal/monthly indices? Frequently, the documentary sources have bias to some season due to the climate and/or the nature of the documents. It is common that the documentary sources provide more information about extreme seasons (winter or summer), or in specific periods of the year in which the climate was determinant for some agricultural labour. For this reason, one must be very carefully when computing annual index just from the addition of seasonal indices and this could be reflected in the text.

*[Response to reviewer] Thank you for this comment. This is a good point and we will integrate additional text into section 8 where relevant.*

*[Changes made] Additional has been added to paragraph 2 of section 8.1 to address this comment.*

**[SC1]** The meaning of "0 index": frequently "0 index" is considered as "normal" condition, but this can be confounded with cases when no information is available. We need to be sure to assume it, that the consulted documentary sources cover the entire period with the same quality. This is easier to evaluated when only one documentary source is used for the whole studied period.

*[Response to reviewer] This is an issue already referred to in the second point made by RC1 – we repeat our response here: "There are two main approaches used to define "0 index" values. One – implicit in the Pfister method – is that no description means no number: a gap in the time series rather than a 0. Other studies make an implicit assumption that, in some circumstances, no weather description can be taken as an indication of normal conditions. We will insert additional text about this in Section 8 and include a paragraph in section 9.2 where we discuss confidence and uncertainty in index-based climate reconstructions."*

*[Changes made] Please see changes made in response to a similar comment by RC1.*

**[SC1]** The distribution of the index values: For reconstruction purposes, it is useful that the index shows similar distribution to the variable to be reconstructed. Nevertheless, this is not always possible, as in some examples provided in this paper (e.g. figure 2 shows a possible bias to negative values, and Mertz´s reconstruction shows less "-1 values" (14) than "-2 values" (20), when probably the opposite is expected). Something similar happens in Figure 8 that shows less "1 values" (abundant flows, mainly concentrated in the last decades) than large swells ("2 values"). Probably, this is because the used documentary sources record

better the extreme than the more common events. Anyway, we think it is an interesting topic to discuss in the paper.

*[Response to reviewer] This is a valuable point that we will mention in section 9.2 on uncertainty (also with reference to a recent publication by White, Pei 2020).*

*[Changes made] We have added an additional sentence on this issue to paragraph 2 of section 9.2.*

**[SC1]** In general, the classic methodology that uses three, five or seven indices is deeply discussed in the paper compared to other methodologies for each type of documentary source that have appeared in recent years. We consider that these particular methodologies have enormous advantages for performing reconstructions and they should be discussed.

We understand that it is impossible to cite all the published articles in the field or to analyse all the methodologies developed in a single paper for all the meteorological variables or events. However, we think that it is possible to include more recent publications. Here we provide some examples.

Regarding the European temperature indices, the more recent work cited dates from 2015. We recommend to include some recent papers as (Brázdil et al., 2019; Fernández-Fernández et al., 2017; Fila et al., 2016; Filipiak et al., 2019; Mrgic, 2018; Rodrigo, 2019). Some of the methodological approaches of these papers are interesting and have not been discussed in the manuscript.

With respect to European precipitation indices, the most recent work cited dates from 12 years ago. Some proposals to update these references are (Brázdil et al., 2019; Bullón, 2011; Fernández-Fernández et al., 2015; Filipiak et al., 2019; Metzger and Tabeaud, 2017; Rodrigo, 2019). Again, these papers have interesting methodologies that have not been discussed.

*[Response to reviewer] We accept the general point made here. We will evaluate the methodologies of the more recent literature mentioned to see if they warrant inclusion in the revised version of the manuscript. Again, the key criterion for the inclusion of any study is that it applies an indexing approach in the wider tradition described for Europe-focused research. Mere overviews on available documentary information for a specific period and region (e.g. Mrgic 2018 for the Balkans) will not qualify a study for integration. We realise that we handle this differently at times for non-European regions, as we see value in pointing the anglophone academic community to these regions. However, there is already plenty of information on available sources for Europe-focused climate historical research.*

*[Changes made] We have checked through the list of more recent publications provided by the reviewers and, where the studies fit the brief for inclusion stipulated in the revised opening paragraph of section 2.1 (i.e. they include original published index series based on primary sources and reconstruct meteorological entities), have incorporated them into the manuscript.*

**[SC1]** Moreover, the authors affirm in the section "Climate indices in Europe" that the information about temperature is more frequent and with a better quality than the precipitation information (e.g., line 124 "Temperature is the most common meteorological phenomenon analysed in Europe" or line 154 "Often the same scale is applied for both temperature and precipitation indices; however, precipitation indices may show more gaps than their temperature counterparts as data may be seasonal or more sporadic"). We think this assumption is true for central and northern Europe, but it is not for the Mediterranean region, for example, where the references to the lack or excess of precipitation are clearly

more frequent than those for temperature. This is because the Mediterranean climate is more temperate but has important precipitation extremes. This is a very important point in the article, because the effect of the different climates in the indexing is almost missing in the review, and this is a key issue in a review wanting to cover different regions of the globe.

A mention to pro pluvia rogation ceremonies as a documentary proxy of European droughts is missing. This proxy has been used in the last decades to understand drought variability in preinstrumental period in various European countries as Spain (Barriendos, 2010; Bravo-Paredes et al., 2020; Domínguez-Castro et al., 2008, 2010, 2012; Tejedor et al., 2018), France (Garnier, 2010), Italy (Piervitali and Colacino, 2001) or Portugal(Fragoso et al., 2018). Moreover, different methodologies have been used to extract the climate information of this proxy as the liturgical act (Martín-Vide and Vallvé, 1995), the expenses (Álvarez Vázquez, 1986), the period of time with continuous rogations (Domínguez-Castro et al., 2008) or the area occupied by the text dedicated to the celebration in the chapter acts (Gil-Guirado et al., 2019). All these methodologies are different to the three, five or seven indices discussed in this paper, and have their advantages and disadvantages that would require a detailed analysis.

Moreover, it would be interesting to cite articles of other variables, as sea level (Camuffo et al., 2017; Camuffo and Sturaro, 2003) in which interesting methods have been developed or about snowfalls reconstruction (Enzi et al., 2014).

*[Response to reviewer] We agree that our statement about temperature as the most common phenomenon should be differentiated with regard to the Mediterranean situation, and we will do this accordingly. The commentators identify correctly an incoherence in the manuscript. While we initially discussed and decided not to integrate proxy indices such as pro pluvia rogation ceremonies in sections concerning Europe, we softened this approach for non-European regions. We will clarify this in the revised introduction and will provide (in section 2.6) the most important references to this long-established index type. Space precludes a full coverage of these and comparable indices relating to Europe, so we will emphasise that our key focus is on narrative sources. Regarding the other variables mentioned (e.g. snowfall or sea level/submersions), most studies focus on reconstructions of occurrence or the creation of new proxy data, rather than differentiated index-like values. Hence, we do not intend to include these contributions.*

*[Changes made] We have differentiated between north/central and Mediterranean Europe more clearly in our statements regarding the relative importance of temperature and precipitation reconstructions. The rogation papers mentioned by the reviewer are very interesting. However, the indices applied in many of these studies indicate the types of rogation but not precipitation or temperature. This was the reason why the majority were not included in the original manuscript. We have reviewed the suggested studies, added mentions of those that include direct climate indices, and added a caveat to the start of section 2.1 stating that we only include studies that reconstruct meteorological entities in our review of the European literature.*

**[SC1]** Additionally, we can point out one minor suggestion:

Line 474 "Finally, Dominguez-Castro et al. (2018) built a long precipitation series for 1891–2015 CE based on descriptions of rain ceremonies in Quito, Ecuador". This is not correct. Domínguez-Castro et al. (2018) presents a precipitation instrumental series from Quito (1891-2015) and a series of wet and dry extremes from rogation ceremonies from 1600.

*[Response to reviewer] Thank you for this correction, which we will apply accordingly.*

*[Changes made] Corrected.*

---

## Referee Report (RR1)

Authors significantly improved the quality of the manuscript in this revised version. All major objections were explained/incorporated. I recommend to do only several minor changes in the final version of the manuscript (see below). I believe that the whole article and especially the final guideline will represent a significant contribution to the development of historical climatology.

**Suggested minor changes:**

Line 712: Rather there should be „…calibration and verification… ". However, as the whole paragraph is dealing with the process of index development, the last sentence may be omitted.

Line 720: "… from the mean of the reference period)"

Lines 933–934: lease consider formulation that is more correct: "During verification, index values calibrated to physical units (e.g. temperature degrees or precipitation amount) are compared with the instrumental data …"

Line 939: Please consider using e.g. "…not stable through time… " instead of "non-stationary". Several types of "stationarity" (and non-stationarity) may be defined strictly in statistical sense.

Line 1024–1025: There should be "… at the risk of incorporating non-homogeneities."

---

## Author Response (AR2)

**"Climate indices in historical climate reconstructions: A global state-of-the-art"**

**Changes to manuscript in response to review comments – second revision**

**Response to Referee #1**

I appreciate the work that the authors have taken in revising the manuscript. Some of my comments have been taken into consideration but it is clear that we do not share the same view of what a critical review is. In their reply they say that 'they embed critical analysis throughout the manuscript', but sections 1-8 are still purely descriptive of the current state of the art, which, as told in my previous review is a highly valuable effort. The new version of section 9.2 has some more analytical character but limited to the treatment of the uncertainty. I think that a stronger comparative analysis among the different approaches would lead to a more convincing paper, but I do not want to go further in this controversy.

**Response** – Many thanks for these comments. The main focus of the manuscript is – as described in the introduction – to document previous work that has used an index approach to reconstruct historical meteorological entities. We politely disagree that the paper is 'purely descriptive' and can point to numerous examples of critical review and comparative analysis. The most obvious ones are our discussion of the varying number of index points used in European reconstructions (section 2.2), our comments on the use of the '0 index' value across continents (see text throughout section 8), the comparisons of the results of overlapping reconstructions for Africa (section 8.3) and our revised discussion of confidence and uncertainty (section 9.2).

We have added a caveat to the last paragraph of the introduction to frame the review for the reader: "…*the emphasis of the article is on the documentation of studies that have used an index approach to climate reconstruction, with critical review and comparison where appropriate. The number of instances where comparative analysis is possible is necessarily restricted by the limited number of studies that have undertaken either different approaches to index development for the same location or identical approaches for different regions*."

The guidelines have been partly modified following my comments, but I still have problems here.

Guideline 8. I do not agree how it is formulated. I think that a general statement advising to sum index series is misleading. Take the case of wet indices. What would happen if you add -3 (very dry period) and 3 (very wet), you will get a 0, indicating a misleading normal value, while what you actually got was two extremes within the same period. The index aggregation should be done with careful assessment, so this guideline needs to be reformulated. This comment also applies to the new text inserted in section 8 after table 1.

**Response** – The reviewer is correct that, when summing the scores for individual months/seasons to produce seasonal/annual totals, negative and positive values cancel one another out. This is perfectly acceptable and is the main way in which years with a '0 index' arise. We appreciate, however, that the process may result in a loss of information.

We have amended the paragraph below Table 1 in section 8.1 to read: "Once monthly index values have been generated, these *are then* summed to produce seasonal or annual classifications where required. Three-month seasonal values can, as a result, fluctuate from -9 to +9 and annual values from -36 to +36 (see Pfister, 1984). *The process of summation may result in positive index values for relatively warmer/wetter months during the year being cancelled out by negative index values for relatively colder/drier months. For example, a year containing a run of extremely dry months followed by a run of extremely wet months may produce a summed index value close to zero – even though the year includes two periods of 'extreme' climate. Careful assessment is therefore required when reporting summed indices to avoid any loss of information, particularly concerning extreme events. The approach used by Nicholson et al. (2012a) for African precipitation series may be helpful here, where individual years were flagged if documentary*

*sources suggested wetter and drier extremes across the year that differed by more than two index classes.*"

Guideline 8 has also been amended to read: "It is advisable to sum-up index series – either in time (i.e. from monthly to seasonal or annual) or in space (i.e. by combining several index series from a climatologically homogeneous region). This approach may well approximate index series to natural climate variability. *Careful assessment is needed, however, to avoid any loss of information during the process of summation, particularly for extreme events (see section 8.1).*"

Guideline 10. The resolution is not the problem when calibrating indices against proxies. The problem is that proxies are an indirect indication of the climate, with their own uncertainties and caveats. They are related through a transfer function with the climate variable and they cannot be taken as the 'truth' for calibration. You can use them for testing and comparing but should not be considered for calibrating, since the index might have a more direct relationship with the climate variable that the proxy itself.

**Response** – Thank you for this. We have modified the sentence to read: "Where instrumental data are not available, overlaps with independent high-resolution palaeoclimate records may be *useful for comparison and testing, noting that palaeoclimate records may have their own biases.*"

Other questions

The statement (lines 713-714 in the track changes manuscript) about precipitation following a gaussian distribution should be corrected. Different from temperature, the precipitation follows different distributions depending on the considered scale, from Gamma (for short periods) to gaussian. The sentence in the text should be clarified.

**Response** – The reviewer is correct. Precipitation does not follow a Gaussian distribution at short time scales. However, when dealing with monthly time scales (which Pfister was), a broadly Gaussian distribution can be assumed. We have clarified the text to read: "In the development of his seven-point scale, Pfister assumed that *monthly* temperature and precipitation followed a Gaussian distribution".

I do not understand the sentence in lines 924-926. "However, even where a period of overlap is lacking, indices from documentary sources can still be used to test reconstructions from proxy data or reconstruction, modelling results and observations". I do not understand the last part of the sentence.

**Response** – We have revised this sentence for clarity to read: "However, even where a period of overlap is lacking, indices from documentary sources can still be used to *cross-check* reconstructions from proxy data (…) or modelling results and observations".

I am wondering why the well-known CLIWOC figure 10 is not referenced to the original papers.

**Response** - We did not reference the original paper for Figure 10 because the image is drafted from data in the open source variant of the CLIWOC database stored at historicalclimatology.com. We have updated the figure caption to clarify this: "Figure 10: Plot of the position of all ships' logbook entries in the CLIWOC database (Degroot and Ottens, 2020). The map is derived from the open source variant of the CLIWOC database (García-Herrera et al., 2005b) held at https://www.historicalclimatology.com.".

One of the longest series derived from documentary sources is the Liu et al 1000 year series of typhoons in S China. This should be included in the text.

Kam-biu Liu, Caiming Shen & Kin-sheun Louie (2001) A 1,000-Year History of Typhoon Landfalls in Guangdong, Southern China, Reconstructed from Chinese Historical Documentary Records,

Annals of the Association of American Geographers, 91:3, 453-464, DOI: 10.1111/0004-5608.00253

**Response** - We did not include this paper in the review as it concerns historical typhoon occurrence/frequency rather than an index-based reconstruction (e.g. of typhoon severity).

**Response to Referee #2**

I recommend to do only several minor changes in the final version of the manuscript (see below). I believe that the whole article and especially the final guideline will represent a significant contribution to the development of historical climatology.

**Response** – Many thanks for this kind comment.

Suggested minor changes:

Line 712: Rather there should be „…calibration and verification… ". However, as the whole paragraph is dealing with the process of index development, the last sentence may be omitted.

**Response** – we have revised the text to read '…calibration and verification…' We have retained the last sentence as it is referring to the next section of the document rather than the following paragraph.

Line 720: "… from the mean of the reference period)"

**Response** – thank you for spotting this – corrected as suggested.

Lines 933–934: please consider formulation that is more correct: "During verification, index values calibrated to physical units (e.g. temperature degrees or precipitation amount) are compared with the instrumental data …"

**Response** – thank you – the text has been changed as suggested.

Line 939: Please consider using e.g. "…not stable through time… " instead of "non-stationary". Several types of "stationarity" (and non-stationarity) may be defined strictly in statistical sense.

**Response** - thank you – text has been amended to read: '…may not be stable through time…'

Line 1024–1025: There should be "… at the risk of incorporating non-homogeneities."

**Response** – thank you – corrected as suggested.

**Response to Referee #3**

The manuscript has improved a lot in the review process. Nevertheless, there are some important questions than remain without solution and I cannot recommend its publication in its current form. One major point is that the authors do not accept the lack of critical analysis in the in the manuscript suggested by RC2 and SC1. In my opinion, to infer the guidelines proposed at the end of the manuscript by the previous sections continues to be impossible.

**Response** – This is a difficult comment to address, as the reviewer provides no specific instances to back up his or her viewpoint. However, please see our response to the opening comment by Reviewer #1, as this addresses Reviewer #3's concerns about the level of critical analysis within the manuscript.

The authors are excluding of the review some indexing methods and documentary sources (see paragraph 197-203 in the manuscript, page 12 of the reply and comments below). For this reason, it is necessary to clarify on which documentary sources and type of reconstructions is focused the review. In the pag.12 of the reply letter the authors said that the narrative sources are the key focus "Space precludes a full coverage of these and comparable indices relating to Europe, so we will emphasise that our key focus is on narrative sources". In this case I suggest including this concept clearly in the title and abstract. Moreover a clear definition of "narrative source" in the introduction is required, and modifies the manuscript according with this definition. I understand that not all the cited sources in the introduction can be considered narrative sources "Information sources include, but are not limited to, annals, chronicles, inscriptions, letters, diaries/journals (including weather diaries), newspapers, financial, legal and administrative documents, ships' logbooks, literature, poems, songs, paintings and pictographic and epigraphic records" and many indices cited in the text are based on both narrative and no narrative sources.

I have the feeling that the authors want to focus the manuscript in situations in which the researcher has many different documentary sources with different origins and with highly variable climate information. In this situation, probably the categorized index +X/-X is a good option. If this is the idea, this must be crystal clear in the manuscript, starting by the title because currently is to general. In this case probably has no sense to include in the review the works focused on weather diaries, logbooks, phenology or chapter acts among others, because in these cases researchers use only one type of documentary sources with very specific climate information. Another option would be to divide the review in works that use many different documentary sources with high variable climate information to reconstruct one series and the works that use only one kind of documentary source with specific climate information.

**Response** – This comment combines several concerns, which we respond to thematically.

1. Narrative evidence/narrative sources

The reviewer requests that we provide clearer definitions of 'narrative sources' and 'narrative evidence'. Although the term is widely used, defining a 'narrative source' is not easy. Documentary sources fall along a spectrum from those containing purely quantitative data to those that include purely narrative evidence.

Rather than attempt a definition, we have checked every mention of 'narrative source' in the manuscript and changed it to 'sources containing narrative evidence', 'documentary sources' or 'narrative evidence' as appropriate. We have also made minor edits to the abstract and opening paragraphs of the manuscript to provide greater clarity. Note that (following Brönnimann et al., 2018) we already identify that narrative evidence is concerned with "descriptions of short-term atmospheric processes and their impacts on environments and societies" (lines 35-36) and echo this in line 41 where we state that narrative descriptions contain "local observations of short-term atmospheric processes and their impacts".

2. Exclusion of indexing methods and documentary sources

The reviewer suggests that we are "are excluding of the review some indexing methods and documentary sources". This comment appears to be related to his/her view of our treatment of historical records of rogation ceremonies. We have hopefully addressed this concern in our response to his/her final substantive comment (see final page of this response). As we explain in the manuscript, we are selective only in the text dealing with Europe, where there is such a large volume of index-based studies that a book would be required to do it justice. Even here, we are very clear that the review only excludes studies that do not include original published series based on primary sources and/or those that do not reconstruct meteorological entities (lines 93-96).

3. Single versus multiple sources

The reviewer suggests that we "want to focus the manuscript in situations in which the researcher has many different documentary sources with different origins and with highly variable climate

information". This isn't quite correct. In the paper, we consider studies where authors have used both single source types (e.g. Brázdil and Kotyza [1995, 2000], Fernández-Fernández et al. [2014]) and multiple source types. We also consider studies that include narrative evidence from weather diaries and from phenological records. It should be remembered that these types of source may contain descriptive accounts of meteorological phenomena as well as quantitative data. In all of these cases it is perfectly feasible to develop categorised indices in the format +X/-X. With this in mind, we would prefer to keep the manuscript organised on a continent-by-continent basis.

We agree, however, that some source types are not suited to the development of '+X/-X' style indices. With this in mind, we have edited the second paragraph in section 10.2, as follows: "Having a standard approach to index-based climate reconstruction would clearly have its benefits. However, we recognise that a 'one size fits all' approach is *neither* appropriate for all climate phenomena *nor for all source types*. The reconstruction of historical wind patterns over the oceans from ships' logbooks *and the identification of precipitation variability through the analysis of descriptions of rogation ceremonies*, for example, already *have* well-developed methodologies and protocols."

We have also amended the text immediately preceding the guidelines: "The guidelines are of greatest relevance to index-based reconstructions of temperature and precipitation *from multiple source types* but also have resonance for other climate phenomena (e.g. winter severity) *and for many single source types (e.g. annals, chronicles, letters, diaries/journals, newspapers).*"

In my opinion this paragraph:
line 197-203: Index series based on historical records of rogation ceremonies – closely linked to precipitation (or a lack thereof) – warrant separate discussion. This source type is particularly valuable for western Mediterranean regions (e.g. Álvarez Vázquez, 1986; Martín-Vide and Vallvé, 1995; Barriendos, 1997; Piervitali and Colacino, 2001; Domínguez-Castro et al., 2008; Barriendos, 2010; Domínguez-Castro et al., 2010; Garnier, 2010; Domínguez-Castro et al., 2012b; Fragoso et al., 2018; Tejedor et al., 2018; Gil-Guirado et al., 2019; Bravo-Paredes et al., 2020). However, as most studies base their indices on the type or cost of ceremonies – or the space within individual documents devoted to describing each ceremony – rather than a meteorological entity, we do not go into further detail.

has some conceptual errors and it is incorrect. The articles cited only have in common that they use rogation as a climate proxy. Some articles generate a precipitation index using pro pluvia and pro serenitate rogations i.e. Álvarez Vázquez, 1986; Martín-Vide and Vallvé, 1995; Barriendos, 1997; Barriendo 2010. Other as Fragoso et al. (2018) work with different climate information, not only with rogation and generate a monthly precipitation index (-1, 0, +1) and a drought record. Gil-Guirado et al. (2019) provides a drought index mainly based in pro pluvia ceremonies and an extreme rainfall index based in some pro serenitate rogations and much other climate information. Bravo-Paredes et al. (2019) analyse the use of pro pluvial rogations in Extremadura as winter NAO proxy. Finally, Piervitali and Colacino (2001), Domínguez-Castro et al. (2008, 2010, 2012b), Garnier (2010) and Tejedor et al. (2018) produce drought index from pro pluvia ceremonies information with different methodologies of indexation. For this reason, in my opinion some articles are correctly cited in section 2.3 but others have no sense in there and must be cited in section 2.5. In addition probably is better to cite the final version of Tejedor et al., 2018/19 accepted in climate of the past (Clim. Past, 15, 1647–1664, 2019, https://doi.org/10.5194/cp-15-1647-2019) instead the Climate of the Past Discussion version.

As I am mentioning, all these works provide indices of "meteorological entity" i.e. precipitation, drought or extreme rainfall events. Moreover some of these works provide original indexation methods far from the (–X,+X) continually cited in this review. In my opinion these original methodologies deserve to be commented and analysed in this review. Dominguez-Castro et al (2008) and Gil-Girado et al (2016) are methodological works, in which different indexing methods are compared. In my opinion this kind of works are necessary to know which methodologies of indexation are better, and not only try to accept and repeat the legacy of previous studies as the authors recommend in the point 4 of the guidelines.

**Response** – We thank the reviewer for this helpful detail. We have edited and expanded the text in the final paragraph of section 2.3 such that it provides much more specific information about rogation as a precipitation proxy:

"*Index series based on historical records of religious rogation ceremonies warrant separate discussion. Rogations are liturgical acts conducted to request either rainfall during a drought (termed pro-pluvia rogations) or an end to excessive or persistent precipitation (pro-serenitate rogations), and were used as an institutional mechanism to address social stress in response to such meteorological extremes (see Martín-Vide and Barriendos, 1995; Barriendos, 2005; Tejedor et al., 2019). Analyses of the occurrence and nature of rogation ceremonies have proven particularly valuable for western Mediterranean regions (most notably the Iberia Peninsula), where they have been used to create precipitation indices spanning the 16th to 19th centuries (e.g. Álvarez Vázquez, 1986; Martín-Vide and Vallvé, 1995; Barriendos, 1997, 2010; Gil-Guirado et al., 2019). In some cases, information about rogation ceremonies has been combined with climate-related narrative evidence to generate precipitation series (e.g. Fragoso et al., 2018). Useful evaluations of different indexing methods are provided by Domínguez-Castro et al. (2008) and Gil-Guirado et al. (2016). For a discussion of the use of rogation ceremonies as a proxy for drought see section 2.5, and for examples of rogation-based reconstructions in South America see section 5.*"

We have added the following paragraph to section 2.5 on drought indices:

"*Drought indices have also been derived for the Western Mediterranean using records of rogation ceremonies, with specific methodologies developed to estimate the length, severity and continuity of drought episodes (see Domínguez-Castro et al., 2008). A number of studies have used evidence of pro-pluvia ceremonies (see section 2.3) as a drought proxy (Piervitali and Colacino, 2001; Domínguez-Castro et al., 2008; Domínguez-Castro et al., 2010; Garnier, 2010; Domínguez-Castro et al., 2012b; Tejedor et al., 2019), sometimes in combination with other narrative evidence (e.g. Fragoso et al., 2018; Gil-Guirado et al., 2019). Readers are referred to Brázdil et al. (2018) for a detailed discussion of the different types of drought indices.*"

And the following sentence to section 2.6:

"*Pro-pluvia rogation ceremonies have been analysed as a proxy for the winter North Atlantic Oscillation between 1824 and 1931 CE in the Extremedura region of Spain (Bravo-Paredes et al., 2020).*"

Finally, we have updated the reference for Tejedor et al. to the accepted 2019 version in *Climate of the Past*.

---

## Author Response (AR3)

**"Climate indices in historical climate reconstructions: A global state-of-the-art"**

**Changes to manuscript in response to review comments – second revision**

**Response to Referee #1**

The effort done by the authors to compile and summarize most of the scientific literature of the manuscript topic is highly valuable. Moreover, the article has improved a lot during the review process. However, I expected a deeper analysis of the indexing methodologies, but probably it is difficult to do all in the same manuscript. Anyway, I think that the manuscript could be published after take in consideration some minor points:

**Response** – We thank the reviewer for these comments. We have addressed all his/her minor points in the manuscript and trust that the paper is now acceptable for publication.

There is an incoherence among table 5 and the text. For example, table 5 indicates that there are no studies of droughts for Americas but in section 5.2 some works are cited:

Line 846 "Mendoza et al. (2007) constructed a similar series of droughts on the Yucatan Peninsula for the 16th to 19th centuries. Garza Merodio (2017) improved this index and extended it back in time (see Hernández and Garza Merodio, 2010), based on the frequency and complexity of rogation ceremonies (16th to 20th centuries)".

Something similar occurs with the Snow/ice also in Americas, the table shows no studies but some works have been cited in section 5.3.

**Response** – The reviewer makes a useful point here. We have revised Table 5 in line with his/her suggestions by adding in scores of 1 under 'Drought' and 'Snow/Ice' for the Americas. We have also checked over the text for other regions and – as a result – added in a further score of 1 under 'Snow/Ice' for the Oceans.

I have some comments to the guidelines:

Point 7 "Where reconstruction must rely on a single observer or record, or on secondary sources, appropriate levels of uncertainty should be noted in the final reconstruction (see 12)". I agree that secondary sources must be used carefully. But I am not sure that reconstruction based on a single observer or record has more uncertainties than reconstruction with more observers. In my opinion, this statement could be in contradiction with this paragraph of the manuscript:

Line 1024-1028 "indices are compiled from a unique documentary source – such as a private diary or diaries (e.g. Brázdil et al., 2008; Adamson, 2015; Domínguez-Castro 1025 et al., 2015), a series of correspondence (e.g. Rodrigo et al., 1998; Nash and Endfield, 2002; Fernández-Fernández et al., 1026 2014) or a series of acts of municipal and ecclesiastical institutions for a location (e.g. Barriendos, 1997; Dominguez-Castro 1027 et al., 2018) – it is easier to identify and correct unexpected bias or homogeneity problems".

**Response** – The reviewer has correctly identified a potential contradiction in the text. Lines 1024-1028 (as quoted) are factually accurate. In Point 7, we were paraphrasing Pfister et al. (2018) but have clearly lost some of the meaning of the original text. We have revised Point 7 to read as follows (new text in italics):

"*If weather in a region is documented within a single contemporary record*, appropriate levels of uncertainty should be noted in the final reconstruction (*see Pfister et al., 2018*)".

Point 8 "It is advisable to sum-up index series – either in time (i.e. from monthly to seasonal or annual) or in space (i.e. by combining several index series from a climatologically homogeneous region). This approach may well approximate index series to natural climate variability. Careful

assessment is needed, however, to avoid any loss of information during the process of summation, particularly for extreme events (see section 8.1)". First, are there any evidences for this affirmation: "This approach may well approximate index series to natural climate variability?" could you provide some references?

**Response** – Having revisited section 8.1 in light of the reviewer's final comment (see below), we have opted to delete the phrase "*This approach may well approximate index series to natural climate variability*" from Point 8.

Secondly, I agree with the authors that sum-up indices produce loss of information, but not only. In my opinion also can produce unexpected bias due to some "problems":
- The possible seasonal bias of the documentary sources. This problem is briefly commented by the authors at line 721, but it should be remembered here, because could be an important problem when sum-up series in time.

**Response** – This is a helpful point. We have added the following sentence to the end of Point 8: "*Potential seasonal biases within documentary sources should also be considered as these will influence annual totals.*"

- In many cases the indexation only generates ordinal data as the authors mention in the manuscript. The ordinal data do not have metric information. Although we label each month numerically as '-2', '-1', '0', +1, +2 the numerals do not indicate equal intervals between levels. This could produce important caveats when we sum months. We do not know if the distant among -2 and -1 is the same than the distant among +1 and +2. Neither we know if this distance among levels is the same in the different months of the year. In my opinion this is a key point in the indexation and could be useful to be commented in section 8.

**Response** – This is a very good point. We have added the following text to paragraph 2 of section 8.1: "*It should be remembered, however, that indexation generates ordinal data, with no guarantee that the intervals between each index level are equal, so that the sum for a specific season or year can only approximate the magnitude of a meteorological phenomenon.*"

---

## Author Response (AR4)

**"Climate indices in historical climate reconstructions: A global state-of-the-art"**

**Response to Editor**

Please find attached the final accepted version of the manuscript plus accompanying files.